# Learning-Augmented Algorithms
# for the Bahncard Problem

**Hailiang Zhao**[1]    **Xueyan Tang**[2]    **Peng Chen**[1]    **Shuiguang Deng**[1]

[1]Zhejiang University    [2]Nanyang Technological University

{hliangzhao,pgchen,dengsg}@zju.edu.cn    asxytang@ntu.edu.sg

## Abstract

In this paper, we study learning-augmented algorithms for the Bahncard problem. The Bahncard problem is a generalization of the ski-rental problem, where a traveler needs to irrevocably and repeatedly decide between a cheap short-term solution and an expensive long-term one with an unknown future. Even though the problem is canonical, only a primal-dual-based learning-augmented algorithm was explicitly designed for it. We develop a new learning-augmented algorithm, named PFSUM, that incorporates both history and short-term future to improve online decision making. We derive the competitive ratio of PFSUM as a function of the prediction error and conduct extensive experiments to show that PFSUM outperforms the primal-dual-based algorithm.

## 1   Introduction

The Bahncard is a railway pass of the German railway company, which provides a discount on all train tickets for a fixed time period of pass validity. When a travel request arises, a traveler can buy the train ticket with the regular price, or purchase a Bahncard first and get entitled to a discount on all train tickets within its valid time. The Bahncard problem is an *online* cost minimization problem, whose objective is to minimize the overall cost of pass and ticket purchases, without knowledge of future travel requests [1]. It reveals a *recurring* renting-or-buying phenomenon, where an online algorithm needs to irrevocably and repeatedly decide between a cheap short-term solution and an expensive long-term one with an unknown future. The performance of an online algorithm is typically measured by the *competitive ratio*, which is the worst-case ratio across all inputs between the costs of the online algorithm and an optimal offline one [2].

Recently, there has been growing interest in using machine-learned predictions to improve online algorithms. One seminal work was done by Lykouris and Vassilvtiskii [3] for the caching problem, in which the online algorithms are allowed to leverage predictions of future inputs to make decisions, but they are not given any guarantee on the prediction accuracy. Since then, the framework of online algorithms with predictions has been extensively studied for a wide range of problems including ski-rental [4–8], scheduling [9, 10], graph optimization [11], covering [12, 13], data structures [14–16], etc. In this framework, the measurements for the online algorithms include *consistency* and *robustness*, where the former refers to the competitive ratio under perfect predictions while the latter refers to the upper bound of the competitive ratio when the predictions can be arbitrarily bad.

The Bahncard problem with machine-learned predictions was recently studied by Bamas et al. [12]. They designed a primal-dual-based algorithm assuming given a prediction of the optimal solution, i.e., an optimal collection of times to purchase Bahncards such that the total cost is minimized. While Bamas et al. [12] presented an elegant framework for seamlessly integrating the primal-dual method with a predicted solution, their approach faces three critical issues. First, their considered scenario is limited, involving slotted time and a fixed ticket price of 1 for any travel request. The discretization is necessary for them to formulate an integer program and enable the application of the primal-dual

method proposed by Buchbinder et al. [17]. Second, their results of consistency and robustness are weak in that they only hold under the condition that the cost of a Bahncard goes towards infinity, in which case the optimal solution is to never buy any Bahncard. Last, their algorithm demands a complete solution as input advice, which implies a predicted *complete* sequence of travel requests over an arbitrarily long timespan, which is impractical for real employment.

In this paper, we study the Bahncard problem with machine-learned predictions, adopting a different technical perspective. We address the general scenario where travel requests may arise at any time with diverse ticket prices. We develop an algorithm, named PFSUM, which takes short-term predictions on future trips as inputs. We derive its competitive ratio under any prediction error that is naturally defined as the difference between the predicted and true values. To present our results, we introduce several notations. The Bahncard problem is instantiated by three parameters and denoted by $\mathrm{BP}(C, \beta, T)$, meaning that a Bahncard costs $C$, reduces any (regular) ticket price $p$ to $\beta p$ for some $0 \leq \beta < 1$, and is valid for a time period of $T$. When $\beta = 0$ and $T \to \infty$, the Bahncard problem reduces to the well-known ski-rental problem. Following the definition in [3, 4], we take the competitive ratio of a learning-augmented online algorithm ALG as a function $\mathrm{CR}_{\mathsf{ALG}}(\eta)$ of the prediction error $\eta$. ALG is $\delta$-*consistent* if $\mathrm{CR}_{\mathsf{ALG}}(0) = \delta$, and $\vartheta$-*robust* if $\mathrm{CR}_{\mathsf{ALG}}(\eta) \leq \vartheta$ for all $\eta$.

At any time $t$ when a travel request arises and there is no a valid Bahncard, a prediction on the total (regular) ticket price of all travel requests in the upcoming interval $[t, t + T)$ is made. Incorporating this prediction, PFSUM purchases a Bahncard at time $t$ when (i) the total ticket price in the past interval $(t - T, t]$ is at least $\gamma$ and (ii) the predicted total ticket price in $[t, t + T)$ is also at least $\gamma$, where $\gamma := C/(1 - \beta)$. Denoting by $\eta$ the maximum prediction error, we derive that

$$\mathrm{CR}_{\mathsf{PFSUM}}(\eta) = \begin{cases} \frac{2\gamma + (2-\beta)\eta}{(1+\beta)\gamma + \beta\eta} & 0 \leq \eta \leq \gamma, \\ \frac{(3-\beta)\gamma + \eta}{(1+\beta)\gamma + \beta\eta} & \eta > \gamma. \end{cases} \tag{1}$$

The result shows that PFSUM is $2/(1 + \beta)$-consistent and $1/\beta$-robust, and its competitive ratio degrades smoothly as the prediction error increases. We also share our experience in the design of PFSUM with some interesting observations.

## 2 Related Work

**The Bahncard problem.** Fleischer [1] was the first to study the Bahncard problem. By extending the optimal 2-competitive break-even algorithm for ski-rental, he proposed an optimal $(2 - \beta)$-competitive deterministic algorithm named SUM for $\mathrm{BP}(C, \beta, T)$. He also proposed a randomized algorithm named RAND for $\mathrm{BP}(C, \beta, \infty)$, and proved that it is $e/(e - 1 + \beta)$-competitive. He conjectured that RAND keeps the same competitiveness for $T < \infty$. Karlin et al. [18] designed a randomized online algorithm for TCP acknowledgement and applied it to $\mathrm{BP}(C, \beta, T)$ with the conjecture settled positively. In addition, the Bahncard problem was extended to several realistic problems in computer systems such as bandwidth cost minimization [19], cloud instance reservation [20], and data migration [21]. Notably, Wang et al. [20] designed algorithms incorporating short-term predictions about the future for reserving virtual machine instances in clouds, which brings inspiration to our algorithm design.

**Learning-augmented algorithms.** Learning-augmented algorithms aim to leverage machine-learned predictions to improve the performance in both theory and practice [22]. Algorithms using possibly imperfect predictions have found applications in numerous important problems [4–7, 9–11, 16, 23–27]. See `https://algorithms-with-predictions.github.io/` for a comprehensive collection of the literature.

To our knowledge, the only learning-augmented algorithms specifically designed for the Bahncard problem were developed by Bamas et al. [12] and Drygala et al. [28]. Bamas et al. [12] proposed an algorithm, named PDLA, which is $\lambda/(1 - \beta + \lambda\beta) \cdot (e^\lambda - \beta)/(e^\lambda - 1)$-consistent and $(e^\lambda - \beta)/(e^\lambda - 1)$-robust when $C \to \infty$, where $\lambda \in (0, 1]$ is a hyper-parameter of the algorithm. Their method, as previously discussed, is limited to scenarios with slotted time and uniform ticket prices for all travel requests. Our approach significantly differs from [12]. First, we address the general scenario where travel requests can arise at any time and feature varied ticket prices. Second, our algorithm utilizes short-term predictions provided at the times when travel requests are made, in contrast to their method which requires predicting a complete solution. We experimentally compare our algorithm with that of [12] in Section 5. Drygala et al. [28] focused on how many predictions are required to gather enough

information to output a near-optimal Bahncard purchasing schedule, assuming all predictions are correct. The learning-augmented algorithm proposed takes a suggested sequence of buying times as input and assumes a slotted time setting, thereby sharing similar drawbacks to [12].

Im et al. [26] explored a somewhat related TCP acknowledgement problem with machine-learned predictions. They introduced a new prediction error measure designed to assess how much the optimal objective value changes as the difference between actual requests and predicted requests varies. A detailed comparison between our work and [26] is provided in Appendix A.1 of [29].

## 3 Preliminaries

We adhere to the notations coined by Fleischer [1]. $\text{BP}(C, \beta, T)$ with $C > 0, T > 0$, and $0 \leq \beta < 1$ is a request-answer game between an online algorithm $\mathsf{ALG}$ and an *adversary* (see, e.g., [30]). The adversary presents a finite sequence of travel requests $\sigma = \sigma_1 \sigma_2 \cdots$, where each $\sigma_i$ is a tuple $(t_i, p_i)$ that contains the travel time $t_i \geq 0$ and the regular ticket price $p_i \geq 0$. The travel requests are presented in chronological order: $0 \leq t_1 < t_2 < \cdots$.

$\mathsf{ALG}$ needs to react to each travel request $\sigma_i$. If $\mathsf{ALG}$ does not have a valid Bahncard, it can opt to buy the ticket with the regular price $p_i$, or first purchase a Bahncard which costs $C$, and then pay the ticket price with a $\beta$-discount, i.e., $\beta p_i$. A Bahncard purchased at time $t$ is valid during the time interval $[t, t + T)$. Note that a travel request arising at time $t + T$ does not benefit from the Bahncard purchased at time $t$. We say $\sigma_i$ is a *reduced request* of $\mathsf{ALG}$ if $\mathsf{ALG}$ has a valid Bahncard at time $t_i$. Otherwise, $\sigma_i$ is a *regular request* of $\mathsf{ALG}$. We use $\mathsf{ALG}(\sigma_i)$ to denote $\mathsf{ALG}$'s cost on $\sigma_i$:

$$\mathsf{ALG}(\sigma_i) = \begin{cases} \beta p_i & \mathsf{ALG} \text{ has a valid Bahncard at } t_i, \\ p_i & \text{otherwise.} \end{cases}$$

We denote by $\mathsf{ALG}(\sigma)$ the total cost of $\mathsf{ALG}$ for reacting to all the travel requests in $\sigma$. The competitive ratio of $\mathsf{ALG}$ is defined by $\mathsf{CR}_{\mathsf{ALG}} := \max_\sigma \mathsf{ALG}(\sigma)/\mathsf{OPT}(\sigma)$, where $\mathsf{OPT}$ is an optimal offline algorithm for $\text{BP}(C, \beta, T)$. We use $\mathsf{ALG}(\sigma; \mathcal{I})$ to denote the partial cost incurred during a time interval $\mathcal{I}$: $\mathsf{ALG}(\sigma; \mathcal{I}) = C \cdot x + \sum_{i:t_i \in \mathcal{I}} \mathsf{ALG}(\sigma_i)$, where $x$ is the number of Bahncards purchased by $\mathsf{ALG}$ in $\mathcal{I}$. Additionally, we use $c(\sigma; \mathcal{I})$ to denote the total regular cost in $\mathcal{I}$: $c(\sigma; \mathcal{I}) := \sum_{i:t_i \in \mathcal{I}} p_i$. Given a time length $l$, we define the *l-recent-cost* of $\sigma$ at time $t$ as $c(\sigma; (t - l, t])$. Similarly, we define the *l-future-cost* of $\sigma$ at time $t$ as $c(\sigma; [t, t + l))$. In our design, we assume that when a travel request $(t, p)$ arises, a short-term prediction of the total regular cost in the time interval $[t, t + T)$ can be made. To represent prediction errors, we use $\hat{c}(\sigma; [t, t + T))$ to denote the predicted total regular cost in $[t, t + T)$. Sometimes we are concerned about the regular requests of an algorithm $\mathsf{ALG}$ in a recent time interval. Thus, we further define the *regular l-recent-cost* of $\mathsf{ALG}$ on $\sigma$ at time $t$ as

$$\mathsf{ALG}^r\big(\sigma; (t - l, t]\big) := \sum_{\substack{i:\sigma_i \text{ is a regular request of } \mathsf{ALG} \text{ in } (t-l,t]}} p_i.$$

Without loss of generality, we assume that an online algorithm $\mathsf{ALG}$ or an optimal offline algorithm $\mathsf{OPT}$ considers purchasing Bahncards at the times of regular requests *only*. The rationale is that the purchase of a Bahncard at any other time can always be delayed to the next regular request without increasing the total cost.

**Lemma 3.1.** *[1] For any time $t$, if $c\big(\sigma; [t, t + T)\big) \geq \gamma := C/(1 - \beta)$, $\mathsf{OPT}$ has at least one reduced request in $[t, t + T)$. The same holds for the time interval $(t, t + T]$.*

$\gamma$ is known as the break-even point, i.e., the threshold to purchase a Bahncard at time $t$ for minimizing the cost incurred in the time interval $[t, t + T)$.

**Corollary 3.2.** *At any time $t$, if the $T$-future-cost $c\big(\sigma; [t, t + T)\big) < \gamma$, $\mathsf{OPT}$ does not purchase a Bahncard at $t$.*

An optimal deterministic online algorithm for the Bahncard problem is $\mathsf{SUM}$, which is $(2 - \beta)$-competitive [1]. $\mathsf{SUM}$ purchases a Bahncard at a regular request $(t, p)$ whenever its regular $T$-recent-cost at time $t$ is at least $\gamma$, i.e., $\mathsf{SUM}^r\big(\sigma; (t - T, t]\big) \geq \gamma$.

# 4 Learning-Augmented Algorithms for the Bahncard Problem

## 4.1 Initial Attempt

The SUM algorithm developed in [1] purchases a Bahncard based on the cost incurred in the past only. It is likely that there is no further travel request in the valid time of the Bahncard besides the request at the purchasing time. Our initial attempt is an adaptation of the $A_\gamma^w$ algorithm proposed in [20]. Aiming to save cost, the $A_\gamma^w$ algorithm shifts the cost consideration for Bahncard purchasing towards the future with the help of predictions. That is, a Bahncard is purchased based on a combination of the actual cost incurred in the past and the predicted cost in the future. We name the adaptation of $A_\gamma^w$ to the Bahncard problem as $\mathsf{SUM}_w$.[1] Specifically, at each regular request $(t, p)$, $\mathsf{SUM}_w$ predicts the total regular cost in a prediction window $(t, t + w]$ where $w$ $(0 < w < T)$ is the length of the prediction window. $\mathsf{SUM}_w$ purchases a Bahncard at a regular request $(t, p)$ whenever the sum of the regular $(T - w)$-recent-cost at $t$ and the predicted total regular cost in $(t, t + w]$ is at least $\gamma$, i.e.,

$$\mathsf{SUM}_w^r\big(\sigma; (t + w - T, t]\big) + \hat{c}\big(\sigma; (t, t + w]\big) \geq \gamma. \tag{2}$$

Note that $\mathsf{SUM}_w$ reduces to $\mathsf{SUM}$ when $w = 0$.

Unfortunately, $\mathsf{SUM}_w$ is not a good algorithm for using machine-learned predictions. We find that its consistency is at least $(3 - \beta)/(1 + \beta)$, which is even larger than $\mathsf{SUM}$'s competitive ratio of $2 - \beta$ since $\beta < 1$. We construct a travel request sequence to show the consistency result in Appendix A.2 of [29]. Moreover, $\mathsf{SUM}_w$ does not have any bounded robustness. Consider another example where only one travel request $(t, p)$ arises with $p \to 0$, but the predictor yields $\hat{c}\big(\sigma; (t, t + w]\big) \geq \gamma$. Then, $\mathsf{SUM}_w$ purchases a Bahncard at $(t, p)$. Thus, we have $\mathsf{SUM}_w(\sigma)/\mathsf{OPT}(\sigma) = (C + \beta p)/p \to \infty$.

$\mathsf{SUM}_w$ fails because if a Bahncard is purchased at time $t$, it is possible that most of the ticket cost in the interval $(t + w - T, t + w]$ is incurred before $t$. Consequently, only a small fraction of the ticket cost is incurred from $t$ onward and can benefit from the Bahncard purchased. As a result, $\mathsf{SUM}_w$ suffers from the same deficiency as $\mathsf{SUM}$. We remark that it is not helpful to set the prediction window length $w$ to $T$ or even larger values, because the travel request arising at (or beyond) time $t + T$ (if any) is not covered by the Bahncard purchased at time $t$.

## 4.2 Second Attempt

What we have learned from $\mathsf{SUM}_w$ is that the Bahncard purchase condition should not be based on the total ticket cost in a past time interval and a future prediction window. Thus, our second algorithm FSUM (Future SUM) is designed to purchase a Bahncard at a regular request $(t, p)$ whenever the predicted $T$-future-cost at time $t$ is at least $\gamma$, i.e.,

$$\hat{c}\big(\sigma; [t, t + T)\big) \geq \gamma. \tag{3}$$

Note that $\mathsf{FSUM} \neq \mathsf{SUM}_T$ because the Bahncard purchase condition of $\mathsf{SUM}_T$ is $\hat{c}\big(\sigma; (t, t + T]\big) \geq \gamma$.

**Theorem 4.1.** *FSUM is $2/(1 + \beta)$-consistent.*

The proof is given in Appendix A.3 of [29]. FSUM's consistency is generally better than SUM's competitive ratio since $2/(1 + \beta) < 2 - \beta$ always holds for $0 < \beta < 1$. However, similar to $\mathsf{SUM}_w$, FSUM does not have any bounded robustness. Consider again the example where only one travel request $(t, p)$ arises with $p \to 0$, but the predictor yields $\hat{c}(\sigma; [t, t + T)) \geq \gamma$. Then, FSUM purchases a Bahncard at $(t, p)$ and $\mathsf{FSUM}(\sigma)/\mathsf{OPT}(\sigma) = (C + \beta p)/p \to \infty$.

## 4.3 PFSUM Algorithm

FSUM fails to achieve any bounded robustness because it completely ignores the historical information in the Bahncard purchase condition. Thus, the worst case is that the actual ticket cost in the prediction window is close to 0, while the predictor forecasts that it exceeds $\gamma$, in which case hardly anything benefits from the Bahncard purchased. On the other hand, we note that SUM achieves a decent competitive ratio because a Bahncard is purchased only when the regular $T$-recent-cost is

---

[1]Wang et al. [20] studied a restricted setting where time is slotted into hours and purchases are performed only at the beginning of slots because virtual machine instances in clouds are billed in an hourly manner.

at least $\gamma$, so that the Bahncard cost can be charged to the regular $T$-recent-cost in the competitive analysis. Motivated by this observation, we introduce a new algorithm PFSUM (Past and Future SUM), in which the Bahncard purchase condition incorporates the ticket costs in both a past time interval and a future prediction window, but uses them *separately* rather than taking their sum. PFSUM purchases a Bahncard at a regular request $(t, p)$ whenever (i) the $T$-recent-cost at $t$ is at least $\gamma$, i.e., $c(\sigma; (t - T, t]) \geq \gamma$, and (ii) the predicted $T$-future-cost at $t$ is also at least $\gamma$, i.e., $\hat{c}(\sigma; [t, t + T)) \geq \gamma$. Note that the first condition is different from the condition for SUM to purchase a Bahncard: PFSUM considers the $T$-recent-cost, but SUM considers only the regular $T$-recent-cost. The rationale is that by definition, the regular $T$-recent-cost is not covered by any Bahncard, so requiring it to be at least $\gamma$ for Bahncard purchase would make the algorithm less effective in saving cost. Compared with FSUM, though PFSUM has an additional condition for purchasing the Bahncard, it is somewhat surprising that PFSUM achieves the same consistency of $2/(1 + \beta)$ as FSUM, as shown below.

Given a travel request sequence $\sigma$, we use $\mu_1 < \cdots < \mu_m$ to denote the times when PFSUM purchases Bahncards. Accordingly, the timespan can be divided into epochs $E_j := [\mu_j, \mu_{j+1})$ for $0 \leq j \leq m$, where we define $\mu_0 = 0$ and $\mu_{m+1} = \infty$. Each epoch $E_j$ (except $E_0$) starts with an *on* phase $[\mu_j, \mu_j + T)$ (the valid time of the Bahncard purchased by PFSUM), followed by an *off* phase $[\mu_j + T, \mu_{j+1})$ (in which there is no valid Bahncard by PFSUM). Epoch $E_0$ has an off phase only. We define $\eta$ as the maximum prediction error among all the predictions used by PFSUM:

$$\eta := \max_{(t,p) \text{ is a regular request}} \left| \hat{c}(\sigma; [t, t + T)) - c(\sigma; [t, t + T)) \right|. \tag{4}$$

Then, for any travel request $(t, p)$ in an off phase, we have

$$c(\sigma; [t, t + T)) - \eta \leq \hat{c}(\sigma; [t, t + T)) \leq c(\sigma; [t, t + T)) + \eta. \tag{5}$$

The following lemmas will be used frequently in the competitive analysis of PFSUM.

**Lemma 4.2.** *The total regular cost in an on phase is at least $\gamma - \eta$.*

*Proof.* If $c(\sigma; [\mu_j, \mu_j + T)) < \gamma - \eta$ for some $j$, it follows from (5) that $\hat{c}(\sigma; [\mu_j, \mu_j + T)) \leq c(\sigma; [\mu_j, \mu_j + T)) + \eta < \gamma - \eta + \eta = \gamma$, which means that PFSUM would not purchase a Bahncard at time $\mu_j$, leading to a contradiction. Thus, we must have $c(\sigma; [\mu_j, \mu_j + T)) \geq \gamma - \eta$. $\square$

**Lemma 4.3.** *In an off phase, the total regular cost in any time interval $[t, t + l)$ of length $l \leq T$ is less than $2\gamma + \eta$.*

*Proof.* Assume on the contrary that $c(\sigma; [t, t + l)) \geq 2\gamma + \eta$. We take the earliest travel request $(t', p)$ in $[t, t + l)$ such that $c(\sigma; [t, t']) \geq \gamma$. This implies $c(\sigma; [t, t')) < \gamma$. Then, we have $c(\sigma; [t', t + l)) = c(\sigma; [t, t + l)) - c(\sigma; [t, t')) > 2\gamma + \eta - \gamma = \gamma + \eta$. Hence, $c(\sigma; (t' - T, t']) \geq c(\sigma; [t, t']) \geq \gamma$ and $c(\sigma; [t', t' + T)) \geq c(\sigma; [t', t + l)) > \gamma + \eta$. By (5), the latter further leads to $\hat{c}(\sigma; [t', t' + T)) \geq c(\sigma; [t', t' + T)) - \eta > \gamma$, which means that PFSUM should purchase a Bahncard at time $t'$, contradicting that $t' \in [t, t + l)$ is in the off phase. Hence, $c(\sigma; [t, t + T)) < 2\gamma + \eta$ must hold. $\square$

**Lemma 4.4.** *Suppose a time interval $[t, t + T)$ overlaps with an off phase. Among the total regular cost in $[t, t + T)$, let $s_2$, $s_3$ and $s_4$ denote those in the preceding on phase, the off phase, and the succeeding on phase respectively (see Figure 1). If $0 \leq \eta \leq \gamma$, $s_2 \leq \gamma$ and $s_4 \leq \gamma$, then the total regular cost in $[t, t + T)$ is no more than $2\gamma + \eta$, i.e., $s_2 + s_3 + s_4 \leq 2\gamma + \eta$.*

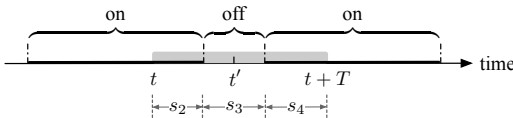

Figure 1: Illustration for Lemma 4.4. The shaded rectangle is the valid time of a Bahncard purchased by OPT.

*Proof.* Assume on the contrary that $s_2 + s_3 + s_4 > 2\gamma + \eta$. Then, $s_3 > 0$ and hence there is at least one travel request during the off phase. We take the earliest travel request $(t', p)$ in the off phase, such that $c(\sigma; [t, t']) \geq \gamma$. With a similar analysis to Lemma 4.3, we can derive that PFSUM should purchase a Bahncard at time $t'$, contradicting that $t'$ is in the off phase. $\square$

We adopt a divide-and-conquer approach to analyze PFSUM. Specifically, we focus on the time intervals in which at least one of PFSUM and OPT has a valid Bahncard, and analyze the cost ratio between PFSUM and OPT in these intervals. Consider a maximal contiguous time interval throughout which at least one of PFSUM and OPT has a valid Bahncard. As shown in Figure 2, there are 6 different patterns. If OPT and PFSUM purchase a Bahncard at the same time, the time interval is exactly an on phase (Pattern I), and the cost ratio in it is 1 (Proposition 4.5). Otherwise, there are 5 different cases: the time interval does not overlap with any on phase (Pattern II); and the time interval overlaps with at least one on phase – it can start at some time in an on or off phase and end at some time in an on or off phase, giving rise to four cases (Patterns III to VI). In Patterns II to VI, we assume that none of the involved Bahncards purchased by OPT are bought at the same time as any Bahncards purchased by PFSUM.

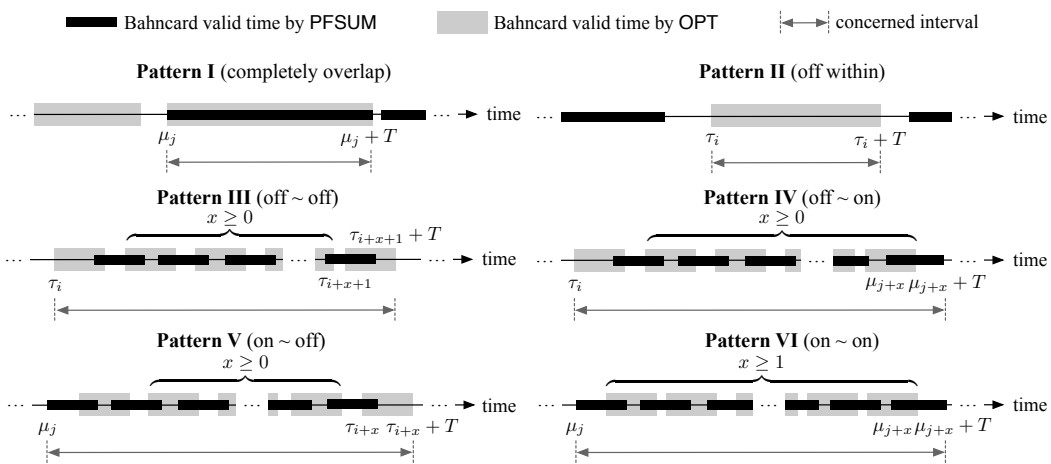

Figure 2: All the 6 patterns of concerned time intervals in which either PFSUM or OPT has a Bahncard. In Patterns III to VI, $x$ is the number of Bahncards purchased by OPT in an on phase and expiring in the next on phase. $x$ can be any non-negative integer.

**Proposition 4.5.** (Pattern I) *If OPT purchases a Bahncard at time $\tau_i$ at the beginning of epoch $E_j$, i.e., $\tau_i = \mu_j$, then*

$$\frac{PFSUM(\sigma; [\mu_j, \mu_j + T))}{OPT(\sigma; [\mu_j, \mu_j + T))} = 1. \tag{6}$$

Apparently, if the cost ratios between PFSUM and OPT of Patterns II to VI are all capped by the same bound, the competitive ratio of PFSUM is given by this bound. Unfortunately, this is *not* exactly true. In what follows, we show that the cost ratios of Patterns II to V can be capped by the same bound (Propositions 4.6 and 4.7), which is the competitive ratio of PFSUM that we would like to prove. For Pattern VI, we show that its cost ratio is capped by the same bound if a particular condition holds, where we refer to such Pattern VI as *augmented* Pattern VI (Proposition 4.8). For non-augmented Pattern VI, we show that it must be accompanied by Patterns I to IV in the sense that a time interval of non-augmented Pattern VI must be preceded (not necessarily immediately) by a time interval of Pattern I, II, III or IV. We prove that the cost ratio of non-augmented Pattern VI combined with such Pattern I, II, III or IV is capped by the same aforesaid bound (Propositions 4.9 and 4.10). This then concludes that PFSUM's competitive ratio is given by this bound.

**Proposition 4.6.** (Pattern II) *If OPT purchases a Bahncard at time $\tau_i$ in the off phase of an epoch $E_j$ and the Bahncard expires in the same off phase, i.e., $\mu_j + T \le \tau_i < \tau_i + T < \mu_{j+1}$, then*

$$\frac{PFSUM(\sigma; [\tau_i, \tau_i + T))}{OPT(\sigma; [\tau_i, \tau_i + T))} < \frac{2\gamma + \eta}{(1 + \beta)\gamma + \beta\eta}. \tag{7}$$

*Proof.* Let $x = c(\sigma; [\tau_i, \tau_i + T))$. Based on the definition of Pattern II, $OPT(\sigma; [\tau_i, \tau_i + T)) = C + \beta x$ (OPT buys a card at $\tau_i$) and $PFSUM(\sigma; [\tau_i, \tau_i + T)) = x$ (PFSUM does not buy cards during any off phase). Hence, the cost ratio is $\frac{x}{C + \beta x}$, which increases with $x$ since $\beta < 1$. By Lemma 4.3, $x < 2\gamma + \eta$. Thus, the cost ratio is bounded by $\frac{2\gamma + \eta}{C + \beta(2\gamma + \eta)} = \frac{2\gamma + \eta}{(1 - \beta)\gamma + \beta(2\gamma + \eta)} = \frac{2\gamma + \eta}{(1 + \beta)\gamma + \beta\eta}$. $\square$

The proof technique of the following propositions for Patterns III to VI is to divide the time interval concerned into sub-intervals, where each sub-interval starts and ends at the time when OPT or PFSUM purchases a Bahncard or a Bahncard purchased expires. Then, for $0 \leq \eta \leq \gamma$ and $\eta > \gamma$, we respectively derive the upper bound of the cost ratio based on Lemmas 4.2 to 4.4. All the bounds in Propositions 4.7 to 4.8 are tight (achievable). Detailed proofs are given in Appendixes A.4 to A.5 of [29].

**Proposition 4.7.** (Patterns III to V) *The cost ratio in the time interval of Patterns III, IV, and V is bounded by*

$$\begin{cases} \frac{2\gamma + (2-\beta)\eta}{(1+\beta)\gamma + \beta\eta} & 0 \leq \eta \leq \gamma, \\ \frac{(3-\beta)\gamma + \eta}{(1+\beta)\gamma + \beta\eta} & \eta > \gamma. \end{cases} \tag{8}$$

We refer to Pattern VI as *augmented* Pattern VI if the total regular cost in the last on phase involved is at least $\gamma$. Proposition 4.8 shows that the cost ratio of augmented Pattern VI is capped by the same bound as Patterns III to V.

**Proposition 4.8.** (Augmented Pattern VI) *Denote by $\mu_j$ and $\mu_{j+x}$ respectively the first and last Bahncards purchased by PFSUM in Pattern VI. If the total regular cost in the on phase of $E_{j+x}$ is at least $\gamma$, the cost ratio in the time interval of Pattern VI, i.e., $[\mu_j, \mu_{j+x} + T)$, is bounded by (8).*

We remark that the cost ratio of general Pattern VI cannot be capped by the same bound as Patterns III to V. This is because in Pattern VI, PFSUM purchases one more Bahncard than OPT and hence has a higher cost of card purchases, whereas PFSUM purchases less or equal numbers of Bahncards compared to OPT in Patterns III to V. Thus, an additional condition is necessary in Proposition 4.8.

Next, we examine non-augmented Pattern VI. Note that the time intervals of Patterns V and VI cannot exist in isolation. By the definition of PFSUM, when a Bahncard is purchased at time $\mu_j$, the total regular cost in the preceding interval $(\mu_j - T, \mu_j]$ is at least $\gamma$ (referred to as feature $\diamond$). Consequently, by Lemma 3.1, OPT must purchase a Bahncard whose valid time overlaps with $(\mu_j - T, \mu_j]$. To deal with non-augmented Pattern VI that starts at time $\mu_j$ for some $j$, we backtrack from $\mu_j$ to find out what happens earlier. Our target is to identify all possible patterns that might precede Pattern VI.

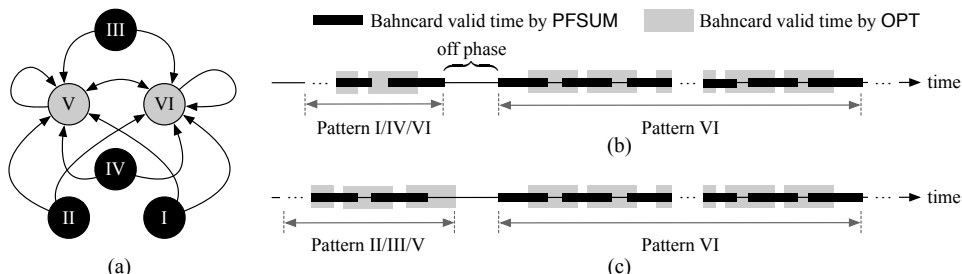

Figure 3: Pattern graph.

Note that the time interval $(\mu_j - T, \mu_j]$ definitely intersects with the off phase of epoch $E_{j-1}$ and may also intersect with the on phase of $E_{j-1}$ (if the off phase of $E_{j-1}$ is shorter than $T$). Therefore, it is possible for all Patterns I to VI to precede Pattern VI (see Figures 3(b) and (c) for illustrations). If Pattern I, IV or VI precedes Pattern VI, there is an off phase in between, in which neither PFSUM nor OPT holds a valid Bahncard. Figure 3(a) presents a *pattern graph* to illustrate all possible concatenations of patterns preceding Pattern VI. In this graph, a node represents a pattern, and an edge from node $i$ to node $j$ means pattern $i$ can precede pattern $j$. Since every Pattern V or VI must be preceded by some pattern, the backtracking will always encounter a time interval of Pattern I, II, III or IV. We stop backtracking at the first Pattern I, II, III or IV encountered.

We use $p_1 \oplus p_2$ to denote the composite of pattern $p_1$ followed by pattern $p_2$; use $p^y$ to denote a sequence comprising $y$ consecutive instances of pattern $p$; and use $\{p_1, \ldots, p_n\} \oplus p_j$ to represent all possible composite patterns of the form $p_i \oplus p_j$ for each $i = 1, \ldots, n$. Then, the patterns encountered in the backtracking can be represented by

$$\{\text{I}, \text{II}, \text{III}, \text{IV}\} \oplus \{\text{V}, \text{VI}\}^y \oplus \text{VI}, \tag{9}$$

where $y$ can be any non-negative integers. Following the feature $\diamond$, for each VI in $\{V, VI\}^y$ of (9), the total regular cost in the last on phase of Pattern VI and the following off phase is at least $\gamma$. It is easy to see that the cost ratio between PFSUM and OPT in the time interval of such Pattern VI is no larger than that of augmented Pattern VI and hence the upper bound given in Proposition 4.8. For each V in $\{V, VI\}^y$ of (9), the cost ratio of Pattern V is capped by the same bound based on Proposition 4.7. In the following, we prove that the cost ratio of non-augmented Pattern VI combined with Pattern I, II, III or IV at the beginning of (9) is also capped by the same bound.

**Proposition 4.9.** *The cost ratio in the combination of a time interval of Pattern VI and a time interval of Pattern II or III is bounded by* (8).

*Proof Sketch.* Note that in Patterns II and III, PFSUM purchases one less Bahncard than OPT. Recall that in Pattern VI, PFSUM purchases one more Bahncard than OPT. Hence, when considering Pattern VI with Pattern II or III together, PFSUM purchases the same number of Bahncards as OPT. Therefore, we can use the same technique as Proposition 4.7 to cap the cost ratio by the same bound. See Appendix A.6 of [29] for details. □

**Proposition 4.10.** *The cost ratio in the combination of a time interval of Pattern VI and a time interval of Pattern I or IV encountered in backtracking is bounded by* (8).

*Proof Sketch.* Consider Pattern IV. When analyzing the cost ratio of Pattern IV in Proposition 4.7, it is assumed that the total regular cost in the last on phase is at least $\gamma - \eta$. In IV of (9), following the feature $\diamond$, the total cost of travel requests in the last on phase of Pattern IV and the following off phase is at least $\gamma$. Thus, there is an additional cost of at least $\eta$. Therefore, we can "migrate" an additional cost of $\eta$ to the non-augmented Pattern VI at the end of (9) and make the latter become an augmented Pattern VI so that the result of Proposition 4.8 can be applied. Meanwhile, the proof of Proposition 4.7 still applies to the Pattern IV even if the cost $\eta$ is removed from the last on phase. Hence, after migrating the cost of $\eta$, the Patterns IV and VI involved have the same upper bound of cost ratio given in (8). The proof for Pattern I is similar. See Appendix A.7 of [29] for details. □

By the above analysis and Propositions 4.6 to 4.10, we have:

**Theorem 4.11.** *PFSUM has a competitive ratio of*

$$CR_{PFSUM}(\eta) = \begin{cases} \frac{2\gamma + (2-\beta)\eta}{(1+\beta)\gamma + \beta\eta} & 0 \le \eta \le \gamma, \\ \frac{(3-\beta)\gamma + \eta}{(1+\beta)\gamma + \beta\eta} & \eta > \gamma. \end{cases} \tag{10}$$

By letting $\eta = 0$, it is easy to see that the consistency of PFSUM is $2/(1 + \beta)$. Note that the competitive ratio of (10) is a continuous function of the prediction error $\eta$. It increases from $2/(1+\beta)$ to $(4-\beta)/(1+2\beta)$ as $\eta$ increases from 0 to $\gamma$, and further increases from $(4-\beta)/(1+2\beta)$ towards $1/\beta$ as $\eta$ increases from $\gamma$ towards infinity. Hence, PFSUM is $1/\beta$-robust.

## 5 Experiments

We conduct extensive experiments to compare PFSUM with SUM [1], PDLA [12], $SUM_w$ (Section 4.1), FSUM (Section 4.2), and SRL (Ski-Rental-based Learning algorithm), where SRL is adapted from Algorithm 2 proposed in [4] for ski-rental. SRL performs as follows. Let $\lambda \in (0, 1]$ be a hyper-parameter, SRL purchases a Bahncard at a regular request $(t, p)$ if and only if there exists a time $t' \in (t - T, t]$ that satisfies one of the following conditions: (i) the predicted $T$-future-cost at time $t'$ is no less than $\gamma$, and the total regular cost in $[t', t]$ is greater than $\lambda\gamma$; or (ii) the predicted $T$-future-cost at time $t'$ is less than $\gamma$, and the total regular cost in $[t', t]$ is greater than $\gamma/\lambda$.

We test $SUM_w$ with $w = T/2$, and test SRL and PDLA with $\lambda$ set to $0.2, 0.5,$ and $1$. To accommodate SRL and PDLA, we discretize time over a sufficiently long timespan, closely approximating a continuous time scenario. The experimental results across diverse input instances demonstrate the superior performance of PFSUM.

**Input instances.** Referring to the experimental setup of [31], we consider two main types of traveler profiles: *commuters*, and *occasional travelers*. We set the timespan at 2000 days, during which commuters travel every day. For occasional travelers, there is a time gap between travel requests. We model the inter-request time of occasional travelers using an exponential distribution with a mean of 2. We consider only one travel request per day. This is because multiple travel requests occurring

on the same day can be effectively treated as a single request with combined ticket price, since a Bahncard's validity either covers an entire day or does not cover any part of the day.

For each request sequence generated, we investigate three types of ticket price distributions: a Normal distribution with a mean of 50 and a variance of 25, a Uniform distribution centered around a mean of 50, and a Pareto distribution called the Lomax distribution with a shape parameter of 2 and a scale parameter of 50. Our choice of the Pareto distribution stems from the observation that prices of travel requests in the real world often exhibit a heavy tail, which suggests that a minority of travel requests typically account for a significant portion of total cost. Such heavy-tailed distributions are often hypothesized to adhere to some form of Pareto distributions [32, 33].

**Noisy prediction.** The predictions are generated by adding noise to the original instance, following the methodology used by Bamas et al. [12]. Specifically, we introduce a perturbation probability $p$. For each day in a given instance, with probability $p$, the travel request on that day is removed, if it exists. Meanwhile, with probability $p$, we add random noise, sampled from the same distribution used for generating ticket prices, to the price of the travel request, or simply add a travel request if no request exists on that day. These two operations are executed independently. We vary the perturbation probability from 0 to 1 in the experiments. Intuitively, the prediction error increases with the perturbation probability. The predictions needed by SRL, SUM$_w$, FSUM, and PFSUM are derived from the total regular cost over the specified time period in the perturbed instance. For PDLA, the predicted solution is obtained by executing the offline optimal algorithm on the perturbed instance.

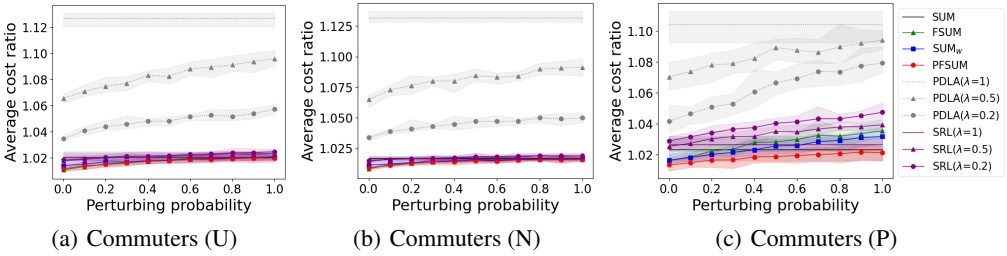

(a) Commuters (U)  (b) Commuters (N)  (c) Commuters (P)

Figure 4: The cost ratios for commuters ($\beta = 0.8$, $T = 10$, $C = 100$). "U", "N" and "P" represent Uniform, Normal and Pareto ticket price distributions respectively.

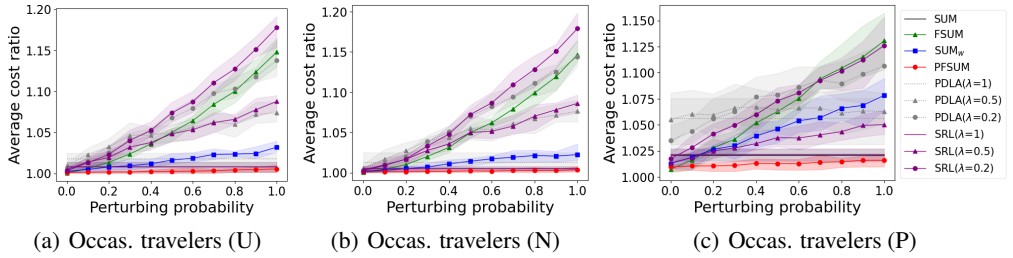

(a) Occas. travelers (U)  (b) Occas. travelers (N)  (c) Occas. travelers (P)

Figure 5: The cost ratios for occasional travelers ($\beta = 0.8$, $T = 10$, $C = 100$). "U", "N" and "P" represent Uniform, Normal and Pareto ticket price distributions respectively.

**Results and discussion.** For all types of input instances, the algorithms exhibit similar relative performance. We present in Figures 4 to 7 the results obtained for commuters and occasional travelers when setting $\beta = 0.8$ (or $0.6$), $T = 10$ (or $5$), $C = 100$, and different ticket price distributions, including the Uniform distribution, the Normal distribution, and the Pareto distribution. Experimental results for other settings are given in Appendix A.8 of [29]. Each curve in the figure represents the average cost ratio between an algorithm and OPT over 100 experiment runs. The shaded area represents the 95% confidence interval of the corresponding curve.

We make the following observations from the experimental results. (1) When predictions are good (perturbation probability is small), PFSUM, FSUM, SUM$_w$, and SRL all perform better than SUM. This shows that predictions on the ticket prices of short-term future trips are useful for reducing the total cost. With perfect predictions (perturbation probability = 0), in most cases, either PFSUM or

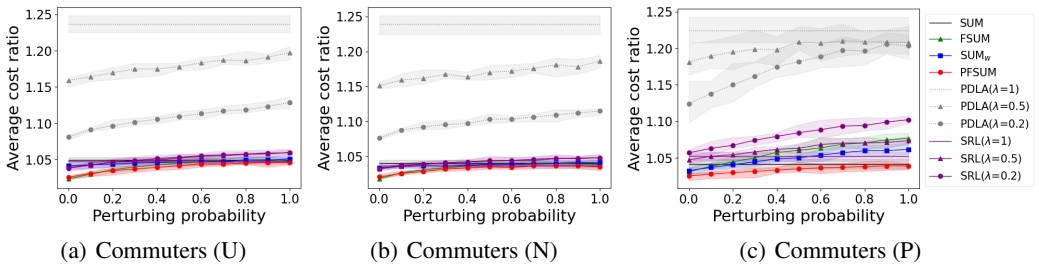

Figure 6: The cost ratios for commuters ($\beta = 0.6$, $T = 5$, $C = 100$). "U", "N" and "P" represent Uniform, Normal and Pareto ticket price distributions respectively.

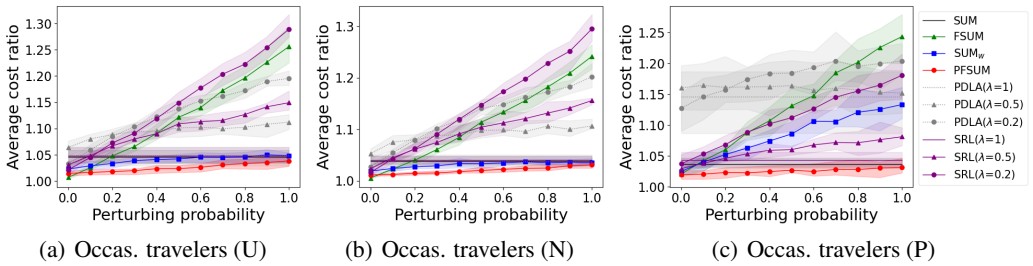

Figure 7: The cost ratios for occasional travelers ($\beta = 0.6$, $T = 5$, $C = 100$). "U", "N" and "P" represent Uniform, Normal and Pareto ticket price distributions respectively.

FSUM produces the lowest cost ratio among all algorithms. This empirically confirms that PFSUM and FSUM have better consistency than SUM and SUM$_w$. (2) PFSUM outperforms all the other algorithms for a wide range of perturbation probability, demonstrating that PFSUM makes wise use of predictions. In general, PFSUM has the narrowest confidence intervals, indicating that its results are the most stable among all algorithms. (3) For the Pareto distribution, as the prediction error increases, PFSUM demonstrates its robustness through a gradual and smooth rise in the cost ratio to OPT. Even when the perturbation probability is large, PFSUM hardly performs worse than SUM. In contrast, the performance of SUM$_w$ and FSUM deteriorates rapidly with increasing perturbation probability, showing their poor robustness. (4) The primal-dual-based algorithm PDLA generally has much higher cost ratios than PFSUM and even the conventional online algorithm SUM. This shows that it is important to take advantage of the characteristics of the Bahncard (it is valid for a fixed time period only and provides the same discount to all the tickets purchased therein) when designing learning-augmented algorithms. (5) PFSUM consistently outperforms SRL, particularly in scenarios with significant prediction errors. While the performance of SRL improves with increasing $\lambda$ under high prediction errors, as a larger $\lambda$ leads to reduced reliance on the predictions and enhances the algorithm's robustness, its overall performance remains inferior to that of SUM.

## 6    Discussions and Conclusions

We have developed a new learning-augmented algorithm called PFSUM for the Bahncard problem. PFSUM makes predictions on the short-term future only, aligning closely with the temporal nature of the Bahncard problem and the practice of real-world ML predictions. We present a comprehensive analysis of PFSUM's competitive ratio under arbitrary prediction errors by a divide-and-conquer approach. Experimental results demonstrate significant performance advantages of PFSUM.

Different from prior works, we do not predict a full sequence of requests from the outset, and predict just a sum of near-future requests (cost) when needed. In some online problems with temporal aspects, compared with individual requests, the aggregated arrival pattern across a group of requests is more predictable (see, e.g., [34]). Thus, our approach can enhance the learnability of ML predictions. We believe that our analysis and result will be useful to various applications of the Bahncard problem as well as other problems with recurring renting-or-buying phenomena.

## Acknowledgments and Disclosure of Funding

This work was supported in part by the Singapore Ministry of Education under Academic Research Fund Tier 2 Award MOE-T2EP20122-0007, the National Natural Science Foundation of China under Grants 62125206 and U20A20173, the National Key Research and Development Program of China under Grant 2022YFB4500100, and the Key Research Project of Zhejiang Province under Grant 2022C01145. Hailiang Zhao's work is supported by Zhejiang University Education Foundation Qizhen Scholar Foundation.

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

# A  Appendix

## A.1  A Detailed Comparison between Our Work and [26]

Im et al. [26] studied a problem somewhat related to the Bahncard problem, focusing on TCP acknowledgement enhanced with machine-learned predictions. In the TCP acknowledgment problem, a sequence of requests unfolds over time, each requiring acknowledgment, and a single acknowledgment can address all pending requests simultaneously. The objective here is to optimize the combined total of request delay and acknowledgement cost (strike a balance between minimizing the delays in addressing requests and the costs of acknowledgments). Im et al. [26] introduced a novel prediction error measure that estimates how much the optimal objective value changes when the difference between the actual numbers of requests and the predictions varies. Im et al. [26] took into account the temporal aspects of requests, akin to a previous work [9]. The new error measure enables their algorithm to achieve near-optimal consistency and robustness simultaneously. Our investigation into the Bahncard problem differs significantly from theirs in terms of techniques employed. The main differences are summarized as follows.

- In [26], predictions are made for a *full* sequence of TCP acknowledgment requests. The prediction necessitates a discrete formulation, where time is partitioned into slots of equal length. In contrast, our approach refrains from discretization and requires only *short-term* predictions within a future interval of length $T$, i.e., the length of the valid time period of a Bahncard. In our work, a prediction on the short-term future is made only when a travel request arises and there is no valid Bahncard, aligning more closely with real-world ML predictions.

- Im et al. [26] designed an error measure that is built on the change of the optimal objective value when the algorithm input varies from $\min\{p_t, \hat{p}_t\}$ to $\max\{p_t, \hat{p}_t\}$, where $p_t$ is the actual number of requests arising in time slot $t$ and $\hat{p}_t$ is the predicted number. By contrast, our error measure estimates the quality of predictions directly. Harnessing the novel error measure, Im et al. [26] formulated an *intricate* algorithm, simultaneously ensuring near-optimal consistency and robustness. In contrast, we design a *simple* and *easily implementable* algorithm using short-term predictions and demonstrates its significant performance advantages in experimental evaluations.

## A.2  Consistency of $\mathsf{SUM}_w$ is at least $(3 - \beta)/(1 + \beta)$

As shown in Figure 8, consider a travel request sequence $\sigma$ of five requests: $(t_1, \epsilon)$, $(t_2, \gamma - \epsilon)$, $(t_3, \gamma - 2\epsilon)$, $(t_4, \epsilon)$, and $(t_5, \epsilon)$, where $t_1 < t_2 \leq t_1 + w < t_4 + w - T < t_1 + T < t_3 < t_4 < t_2 + T < t_3 + w < t_5 \leq t_4 + w$. By the algorithm definition, $\mathsf{SUM}_w$ purchases two Bahncards at times $t_1$ and $t_4$ respectively, while $\mathsf{OPT}$ purchases a Bahncard at time $t_2$. When $\epsilon \to 0$, we have

$$\frac{\mathsf{SUM}_w(\sigma)}{\mathsf{OPT}(\sigma)} = \frac{2C + \beta(\gamma + 2\epsilon) + (\gamma - 2\epsilon)}{C + \beta(2\gamma - 2\epsilon) + 2\epsilon} \to \frac{2C + \beta\gamma + \gamma}{C + 2\beta\gamma} = \frac{3 - \beta}{1 + \beta}.$$

Figure 8: An example where the shaded rectangle (resp. bold line) is the valid time of a Bahncard purchased by $\mathsf{SUM}_w$ (resp. $\mathsf{OPT}$). $\mathsf{SUM}_w$ purchases a Bahncard at $t_1$ because $t_2 \leq t_1 + w$ and $\epsilon + (\gamma - \epsilon) \geq \gamma$. $\mathsf{SUM}_w$ does not purchase a Bahncard at $t_3$ since $t_3 + w < t_5$ and $\mathsf{SUM}_w^r(\sigma; (t_3 + w - T, t_3]) + \hat{c}(\sigma; (t_3, t_3 + w]) = (\gamma - 2\epsilon) + \epsilon < \gamma$. $\mathsf{SUM}_w$ purchases a Bahncard at $t_4$ because $t_3, t_4, t_5 \in (t_4 + w - T, t_4 + w]$ and the total ticket cost is $\gamma$.

## A.3  Proof of Theorem 4.1

For analyzing consistency, we have $\hat{c}(\sigma; [t, t + T)) \equiv c(\sigma; [t, t + T))$ holds for any regular request $(t, p)$. Given a travel request sequence $\sigma$, we denote by $\mu_1 < \cdots < \mu_m$ the times when $\mathsf{FSUM}$ purchases Bahncards. Then, the timespan can be divided into epochs $E_j := [\mu_j, \mu_{j+1})$ for $0 \leq j \leq m$, where we define $\mu_0 = 0$ and $\mu_{m+1} = \infty$. Due to perfect predictions, each epoch $E_j$ (except $E_0$)

starts with an *on* phase $[\mu_j, \mu_j + T)$ in which the total regular cost is at least $\gamma$, followed by an *off* phase $[\mu_j + T, \mu_{j+1})$. Epoch $E_0$ has an off phase only. By the definition of FSUM, for any time $t$ in an off phase when a travel request arises, we have

$$\hat{c}(\sigma; [t, t+T)) = c(\sigma; [t, t+T)) < \gamma. \tag{11}$$

Note that when both FSUM and OPT do not have a valid Bahncard, they pay the same cost. Thus, we focus on only the time intervals in which at least one of FSUM and OPT has a valid Bahncard. We examine the Bahncards purchased by OPT relative to the Bahncards purchased by FSUM. Clearly, OPT purchases at most one Bahncard in any on phase since the length of an on phase is $T$, and OPT does not purchase any Bahncard in any off phase by (11) and Corollary 3.2. For any epoch $E_j$, if OPT purchases its $i$-th Bahncard at time $\tau_i \in [\mu_j, \mu_j + T)$, there are two cases to consider.

Figure 9: Case I. The shaded rectangle is the valid time of a Bahncard purchased by OPT.

**Case I.** The Bahncard expires in the following off phase $[\mu_j + T, \mu_{j+1})$. As shown in Figure 9, for ease of notation, we use $s_1, ..., s_5$ to denote the total regular costs in the time intervals $[\mu_j, \max\{\mu_j, \tau_{i-1} + T\})$, $[\max\{\mu_j, \tau_{i-1} + T\}, \tau_i)$, $[\tau_i, \mu_j + T)$, $[\mu_j + T, \tau_i + T)$, and $[\tau_i + T, \mu_{j+1})$ respectively. Here $\tau_{i-1}$ is the time when OPT purchases its $(i-1)$-th Bahncard. If $i = 1$, we define $\tau_{i-1} = -\infty$. Note that some of the aforesaid intervals can be empty so that the corresponding total regular cost is 0. Since the total regular cost in an on phase is at least $\gamma$, we have

$$s_1 + s_2 + s_3 \geq \gamma. \tag{12}$$

By Lemma 3.1, we have $s_3 + s_4 \geq \gamma$. If no travel request arises during the time interval $[\mu_j + T, \tau_i + T)$, we have $s_4 = 0$. Otherwise, we denote by $t' \in [\mu_j + T, \tau_i + T)$ the time when the first travel request arises in this interval. Since $t'$ is in an off phase, it follows from (11) and $\tau_i < t'$ that

$$s_4 = c(\sigma; [\mu_j + T, \tau_i + T)) = c(\sigma; [t', \tau_i + T)) \leq c(\sigma; [t', t' + T)) < \gamma. \tag{13}$$

Hence, we always have

$$s_4 < \gamma. \tag{14}$$

The cost ratio in epoch $E_j$ is given by

$$
\begin{aligned}
\frac{\mathsf{FSUM}(\sigma; E_j)}{\mathsf{OPT}(\sigma; E_j)} &= \frac{C + \beta(s_1 + s_2 + s_3) + s_4 + s_5}{C + \beta(s_1 + s_3 + s_4) + s_2 + s_5} \\
&< \frac{C + \beta(s_1 + s_2 + s_3) + \gamma + s_5}{C + \beta(s_1 + s_3 + \gamma) + s_2 + s_5} && \text{(14) \& } \beta < 1 \\
&\leq \frac{C + \beta(s_1 + s_2 + s_3) + \gamma + s_5}{C + \beta(s_1 + s_2 + s_3 + \gamma) + s_5} && s_2 \geq 0 \text{ \& } \beta < 1 \\
&\leq \frac{C + \beta(s_1 + s_2 + s_3) + \gamma}{C + \beta(s_1 + s_2 + s_3 + \gamma)} && s_5 \geq 0 \\
&\leq \frac{C + \beta\gamma + \gamma}{C + 2\beta\gamma} && \text{(12)} \\
&= \frac{2}{1 + \beta}. && \gamma = \frac{C}{1 - \beta} \quad \text{(15)}
\end{aligned}
$$

**Case II.** The Bahncard expires in the on phase of the next epoch, i.e., $[\mu_{j+1}, \mu_{j+1} + T)$.

**Case II-A.** OPT does not purchase any Bahncard in $[\tau_i + T, \mu_{j+1} + T)$. As shown in Figure 10, we use $s_1, ..., s_7$ to denote the total regular costs in the time intervals $[\mu_j, \max\{\mu_j, \tau_{i-1} + T\})$, $[\max\{\mu_j, \tau_{i-1} + T\}, \tau_i)$, $[\tau_i, \mu_j + T)$, $[\mu_j + T, \mu_{j+1})$, $[\mu_{j+1}, \tau_i + T)$, $[\tau_i + T, \mu_{j+1} + T)$, and

$[\mu_{j+1} + T, \mu_{j+2})$, respectively. Again $\tau_{i-1}$ is the time when OPT purchases its $(i-1)$-th Bahncard. If $i = 1$, we define $\tau_{i-1} = -\infty$. Since the total regular cost in an on phase is at least $\gamma$, we have

$$s_1 + s_2 + s_3 \geq \gamma, \tag{16}$$
$$s_5 + s_6 \geq \gamma. \tag{17}$$

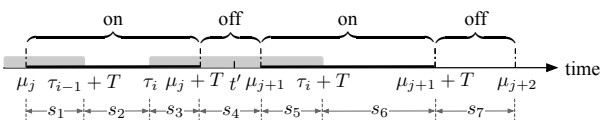

Figure 10: Case II-A. The shaded rectangle is the valid time of a Bahncard purchased by OPT.

If no travel request arises during the time interval $[\mu_j + T, \mu_{j+1})$, we have $s_4 = 0 \leq s_6$. Otherwise, we denote by $t' \in [\mu_j + T, \mu_{j+1})$ the time when the first travel request arises in this interval. It follows from (11) and $\tau_i < t'$ that

$$s_4 + s_5 = c\big(\sigma; [\mu_j + T, \tau_i + T)\big) = c\big(\sigma; [t', \tau_i + T)\big) \leq c\big(\sigma; [t', t' + T)\big) < \gamma. \tag{18}$$

Together with (17), we get $s_4 < s_6$. Therefore, we always have

$$s_4 \leq s_6, \tag{19}$$

and

$$s_4 < \gamma. \tag{20}$$

The cost ratio in epochs $E_j$ and $E_{j+1}$ is given by

$$
\begin{aligned}
\frac{\mathsf{FSUM}(\sigma; E_j \cup E_{j+1})}{\mathsf{OPT}(\sigma; E_j \cup E_{j+1})} &= \frac{2C + \beta(s_1 + s_2 + s_3 + s_5 + s_6) + s_4 + s_7}{C + \beta(s_1 + s_3 + s_4 + s_5) + s_2 + s_6 + s_7} \\
&\leq \frac{2C + \beta(s_1 + s_2 + s_3 + s_5 + s_6) + s_4 + s_7}{C + \beta(s_1 + s_3 + s_5 + s_6) + s_2 + s_4 + s_7} && \text{(19) \& } \beta < 1 \\
&\leq \frac{2C + \beta(s_1 + s_2 + s_3 + s_5 + s_6) + s_4 + s_7}{C + \beta(s_1 + s_2 + s_3 + s_5 + s_6) + s_4 + s_7} && s_2 \geq 0 \ \& \ \beta < 1 \\
&\leq \frac{2C + \beta(s_1 + s_2 + s_3 + s_5 + s_6)}{C + \beta(s_1 + s_2 + s_3 + s_5 + s_6)} && s_4 \geq 0 \ \& \ s_7 \geq 0 \\
&\leq \frac{2C + 2\beta\gamma}{C + 2\beta\gamma} && \text{(16) \& (17)} \\
&= \frac{2}{1 + \beta}. && \gamma = \frac{C}{1 - \beta} \qquad \text{(21)}
\end{aligned}
$$

In addition, the cost ratio in epoch $E_j$ is given by

$$
\begin{aligned}
\frac{\mathsf{FSUM}(\sigma; E_j)}{\mathsf{OPT}(\sigma; E_j)} &= \frac{C + \beta(s_1 + s_2 + s_3) + s_4}{C + \beta(s_1 + s_3 + s_4) + s_2} \\
&\leq \frac{C + \beta(s_1 + s_2 + s_3) + s_4}{C + \beta(s_1 + s_2 + s_3 + s_4)} && s_2 \geq 0 \ \& \ \beta < 1 \\
&\leq \frac{C + \beta\gamma + s_4}{C + \beta\gamma + \beta s_4} && \text{(16)} \\
&< \frac{C + \beta\gamma + \gamma}{C + \beta\gamma + \beta\gamma} && \text{(20) \& } \beta < 1 \\
&= \frac{2}{1 + \beta}. && \gamma = \frac{C}{1 - \beta} \qquad \text{(22)}
\end{aligned}
$$

**Case II-B.** After time $\tau_i + T$, OPT purchases another $x$ Bahncards at times $\tau_{i+1}, ..., \tau_{i+x}$ respectively ($x \geq 1$) as shown in Figure 11. For each $k = 1, ..., x$, the purchasing time $\tau_{i+k}$ falls in the on phase

$[\mu_{j+k}, \mu_{j+k} + T)$, and the expiry time $\tau_{i+k} + T$ falls in the next on phase $[\mu_{j+k+1}, \mu_{j+k+1} + T)$. OPT does not purchase any new Bahncard in $[\tau_{i+x} + T, \mu_{j+x+1} + T)$. Essentially, the pattern in epoch $E_j$ of Case II-A repeats $x$ times, followed by the pattern of Case II-A. By (22), $\mathsf{FSUM}(\sigma; E_{j+k})/\mathsf{OPT}(\sigma; E_{j+k}) \leq 2/(1+\beta)$ for each $k = 0, ..., x-1$. By (21), $\mathsf{FSUM}(\sigma; E_{j+x} \cup E_{j+x+1})/\mathsf{OPT}(\sigma; E_{j+x} \cup E_{j+x+1}) \leq 2/(1+\beta)$. Hence,

$$\frac{\mathsf{FSUM}(\sigma; E_j \cup \cdots \cup E_{j+x+1})}{\mathsf{OPT}(\sigma; E_j \cup \cdots \cup E_{j+x+1})} \leq \frac{2}{1+\beta}. \tag{23}$$

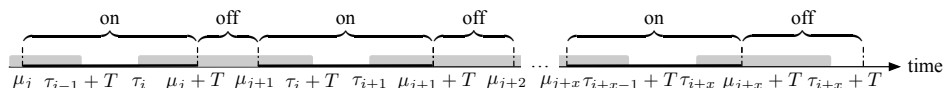

Figure 11: Case II-B. The shaded rectangle is the valid time of a Bahncard purchased by OPT.

**Case II-C.** After time $\tau_i + T$, OPT purchases another $x$ Bahncards at times $\tau_{i+1}, ..., \tau_{i+x}$ respectively ($x \geq 1$) as shown in Figure 12. For each $k = 1, ..., x-1$, the purchasing time $\tau_{i+k}$ falls in the on phase $[\mu_{j+k}, \mu_{j+k}+T)$, and the expiry time $\tau_{i+k}+T$ falls in the next on phase $[\mu_{j+k+1}, \mu_{j+k+1}+T)$. The purchasing time $\tau_{i+x}$ falls in the on phase $[\mu_{j+x}, \mu_{j+x}+T)$, and the expiry time $\tau_{i+x}+T$ falls in the following off phase $[\mu_{j+x}+T, \mu_{j+x+1})$. Essentially, the pattern in epoch $E_j$ of Case II-A repeats $x$ times, followed by the pattern of Case I. By (22), $\mathsf{FSUM}(\sigma; E_{j+k})/\mathsf{OPT}(\sigma; E_{j+k}) \leq 2/(1+\beta)$ for each $k = 0, ..., x-1$. By (15), $\mathsf{FSUM}(\sigma; E_{j+x})/\mathsf{OPT}(\sigma; E_{j+x}) \leq 2/(1+\beta)$. Hence,

$$\frac{\mathsf{FSUM}(\sigma; E_j \cup \cdots \cup E_{j+x})}{\mathsf{OPT}(\sigma; E_j \cup \cdots \cup E_{j+x})} \leq \frac{2}{1+\beta}. \tag{24}$$

Figure 12: Case II-C. The shaded rectangle is the valid time of a Bahncard purchased by OPT.

If OPT does not purchase any Bahncard in the on phase of $E_j$, it must have purchased a Bahncard in the on phase of $E_{j-1}$, and the Bahncard expires in the on phase of $E_j$ due to Lemma 3.1. Otherwise, purchasing another Bahncard at the beginning of $E_j$ can never increase the total cost, because the total regular cost in the on phase of $E_j$ is at least $\gamma$. Thus, $E_{j-1}$ and $E_j$ must fall into Case II-A or be the last two epochs of Case II-B, which have been analyzed above.

The theorem follows from (15), (21), (23) and (24).

## A.4 Proof of Proposition 4.7

### A.4.1 Proof for Pattern III

**Proposition Restated.** If OPT purchases $x + 2$ Bahncards ($x \geq 0$) in successive on phases starting from $E_j$, where (i) the first Bahncard has its purchase time $\tau_i$ falling in the off phase of $E_{j-1}$ and its expiry time $\tau_i + T$ falling in the on phase of $E_j$, (ii) for each $k = 1, ..., x$, the $(k+1)$-th Bahncard has its purchase time $\tau_{i+k}$ falling in the on phase of $E_{j+k-1}$ and its expiry time $\tau_{i+k} + T$ falling in the on phase of $E_{j+k}$, and (iii) the $(x+2)$-th Bahncard has its purchase time $\tau_{i+x+1}$ falling in the on phase of $E_{j+x}$ and its expiry time $\tau_{i+x+1} + T$ falling in the off phase of $E_{j+x}$, then

$$\frac{\mathsf{PFSUM}\big(\sigma; [\tau_i, \tau_{i+x+1}+T)\big)}{\mathsf{OPT}\big(\sigma; [\tau_i, \tau_{i+x+1}+T)\big)} \leq \begin{cases} \frac{2\gamma + (2-\beta)\eta}{(1+\beta)\gamma + \beta\eta} & 0 \leq \eta \leq \gamma, \\ \frac{(3-\beta)\gamma + \eta}{(1+\beta)\gamma + \beta\eta} & \eta > \gamma, \end{cases} \tag{25}$$

where the upper bound is tight (achievable) when $x \to \infty$.

**Proof.** As shown in Figure 13, we divide $[\tau_i, \tau_{i+x+1}+T)$ into $4x+5$ time intervals, where each time interval starts and ends with the time when OPT or PFSUM purchases a Bahncard or a Bahncard purchased by OPT or PFSUM expires. Let these intervals be indexed by $-1, 0, 1, 2, ..., 4x+3$, and

let $s_{-1}, s_0, s_1, s_2, ..., s_{4x+3}$ denote the total regular costs in these intervals respectively. By definition, it is easy to see that for each $k = 0, ..., x-1$, the $(4k+3)$-th time interval, i.e., $[\mu_{j+k} + T, \mu_{j+k+1})$ (which is an off phase), is shorter than $T$ since it is within the valid time of a Bahncard purchased by OPT.

Figure 13: Illustration for Pattern III. The shaded rectangle is the valid time of a Bahncard purchased by OPT.

First, we observe that the cost ratio $\mathsf{PFSUM}\big(\sigma; [\tau_i, \tau_{i+x+1} + T)\big)/\mathsf{OPT}\big(\sigma; [\tau_i, \tau_{i+x+1} + T)\big)$ is less than $1/\beta$, as shown below.

$$
\frac{1}{\beta} - \frac{\mathsf{PFSUM}\big(\sigma; [\tau_i, \tau_{i+x+1} + T)\big)}{\mathsf{OPT}\big(\sigma; [\tau_i, \tau_{i+x+1} + T)\big)}
$$

$$
= \frac{(x+2)C + \beta\Big[(s_{-1} + s_0) + \sum_{k=0}^{x-1}\big(s_{4k+2} + s_{4k+3} + s_{4k+4}\big) + (s_{4x+2} + s_{4x+3})\Big]}{\beta \cdot \mathsf{OPT}\big(\sigma; [\tau_i, \tau_{i+x+1} + T)\big)}
$$

$$
+ \frac{\sum_{k=0}^{x} s_{4k+1} - \beta\Big[(x+1)C + \beta\sum_{k=0}^{x}\big(s_{4k} + s_{4k+1} + s_{4k+2}\big) + \sum_{k=-1}^{x} s_{4k+3}\Big]}{\beta \cdot \mathsf{OPT}\big(\sigma; [\tau_i, \tau_{i+x+1} + T)\big)}
$$

$$
= \frac{(x+2)C + \beta\sum_{k=0}^{x} s_{4k} + \sum_{k=0}^{x} s_{4k+1} + \beta\sum_{k=0}^{x} s_{4k+2} + \beta\sum_{k=-1}^{x} s_{4k+3}}{\beta \cdot \mathsf{OPT}\big(\sigma; [\tau_i, \tau_{i+x+1} + T)\big)}
$$

$$
- \frac{\beta(x+1)C + \beta^2\sum_{k=0}^{x}\big(s_{4k} + s_{4k+1} + s_{4k+2}\big) + \beta\sum_{k=-1}^{x} s_{4k+3}}{\beta \cdot \mathsf{OPT}\big(\sigma; [\tau_i, \tau_{i+x+1} + T)\big)}
$$

$$
= \frac{[(x+2) - \beta(x+1)]C + \beta(1-\beta)\sum_{k=0}^{x}\big(s_{4k} + s_{4k+2}\big) + (1-\beta^2)\sum_{k=0}^{x} s_{4k+1}}{\beta \cdot \mathsf{OPT}\big(\sigma; [\tau_i, \tau_{i+x+1} + T)\big)}
$$

$$
> 0. \quad \text{(since } 0 \le \beta < 1\text{)}
$$

Thus, the following inequality always holds:

$$
\frac{\mathsf{PFSUM}\big(\sigma; [\tau_i, \tau_{i+x+1} + T)\big)}{\mathsf{OPT}\big(\sigma; [\tau_i, \tau_{i+x+1} + T)\big)} < \frac{1}{\beta}. \tag{26}
$$

Next, we analyze the upper bound of the cost ratio. There are two cases to consider.

**Case I.** $0 \le \eta \le \gamma$. By Corollary 3.2, for each $k = -1, 0, ..., x$, the $T$-future-cost at $\tau_{i+k+1}$ is at least $\gamma$, i.e.,

$$
\begin{cases}
s_{-1} + s_0 \ge \gamma, \\
s_{4k+2} + s_{4k+3} + s_{4k+4} \ge \gamma & \text{for each } k = 0, ..., x-1, \\
s_{4x+2} + s_{4x+3} \ge \gamma.
\end{cases} \tag{27}
$$

By Lemma 4.2, for each $k = 0, ..., x$, we have

$$
s_{4k} + s_{4k+1} + s_{4k+2} \ge \gamma - \eta. \tag{28}
$$

Note that all the travel requests in the $(4k+2)$-th (for $k = 0, ..., x$) and the $(4k+4)$-th (for $k = -1, ..., x-1$) time intervals are reduced requests of both PFSUM and OPT. Thus, to maximize the cost ratio in $[\tau_i, \tau_{i+x+1} + T)$, we should minimize these $s_{4k+2}$'s and $s_{4k+4}$'s. If they are greater

than $\gamma$, the cost ratio can be increased by decreasing $s_{4k+2}$ or $s_{4k+4}$ to $\gamma$ without violating (27) and (28). Thus, for the purpose of deriving an upper bound on the cost ratio, we can assume that

$$\begin{cases} s_{4k+2} \leq \gamma & \text{for each } k = 0, ..., x, \\ s_{4k+4} \leq \gamma & \text{for each } k = -1, ..., x - 1. \end{cases} \tag{29}$$

It follows from (29) and Lemma 4.4 that

$$\begin{cases} s_{4k+3} + s_{4k+4} \leq 2\gamma + \eta & \text{for } k = -1, \\ s_{4k+2} + s_{4k+3} + s_{4k+4} \leq 2\gamma + \eta & \text{for each } k = 0, ..., x - 1, \\ s_{4k+2} + s_{4k+3} \leq 2\gamma + \eta & \text{for } k = x. \end{cases} \tag{30}$$

As a result, we have

$$\frac{\mathsf{PFSUM}\big(\sigma; [\tau_i, \tau_{i+x+1} + T)\big)}{\mathsf{OPT}\big(\sigma; [\tau_i, \tau_{i+x+1} + T)\big)}$$

$$= \frac{(x+1)C + \beta \sum_{k=0}^{x}\big(s_{4k} + s_{4k+1} + s_{4k+2}\big) + \sum_{k=-1}^{x} s_{4k+3}}{(x+2)C + \beta\Big[(s_{-1} + s_0) + \sum_{k=0}^{x-1}\big(s_{4k+2} + s_{4k+3} + s_{4k+4}\big) + (s_{4x+2} + s_{4x+3})\Big] + \sum_{k=0}^{x} s_{4k+1}}$$

$$= \frac{(x+1)C + \Big[(s_{-1} + s_0) + \sum_{k=0}^{x-1}\big(s_{4k+2} + s_{4k+3} + s_{4k+4}\big) + (s_{4x+2} + s_{4x+3})\Big]}{(x+2)C + \beta\Big[(s_{-1} + s_0) + \sum_{k=0}^{x-1}\big(s_{4k+2} + s_{4k+3} + s_{4k+4}\big) + (s_{4x+2} + s_{4x+3})\Big] + \sum_{k=0}^{x} s_{4k+1}}$$

$$+ \frac{\beta \sum_{k=0}^{x} s_{4k+1} - (1-\beta)\sum_{k=0}^{x}\big(s_{4k} + s_{4k+2}\big)}{(x+2)C + \beta\Big[(s_{-1} + s_0) + \sum_{k=0}^{x-1}\big(s_{4k+2} + s_{4k+3} + s_{4k+4}\big) + (s_{4x+2} + s_{4x+3})\Big] + \sum_{k=0}^{x} s_{4k+1}}$$

$$\leq \frac{(x+1)C + (x+2)(2\gamma + \eta) + \beta \sum_{k=0}^{x} s_{4k+1} - (1-\beta)\sum_{k=0}^{x}\big(s_{4k} + s_{4k+2}\big)}{(x+2)C + \beta(x+2)(2\gamma + \eta) + \sum_{k=0}^{x} s_{4k+1}} \quad \text{(by (26)) and (30))}$$

$$= \frac{(x+1)C + (x+2)(2\gamma + \eta) + \sum_{k=0}^{x} s_{4k+1} - (1-\beta)\sum_{k=0}^{x}\big(s_{4k} + s_{4k+1} + s_{4k+2}\big)}{(x+2)C + \beta(x+2)(2\gamma + \eta) + \sum_{k=0}^{x} s_{4k+1}}$$

$$\leq \frac{(x+1)C + (x+2)(2\gamma + \eta) + \sum_{k=0}^{x} s_{4k+1} - (1-\beta)(x+1)(\gamma - \eta)}{(x+2)C + \beta(x+2)(2\gamma + \eta) + \sum_{k=0}^{x} s_{4k+1}} \quad \text{(by (28))}$$

$$\leq \frac{(x+1)C + (x+2)(2\gamma + \eta) - (1-\beta)(x+1)(\gamma - \eta)}{(x+2)C + \beta(x+2)(2\gamma + \eta)} \quad \left(\text{since } \sum_{k=0}^{x} s_{4k+1} \geq 0\right)$$

$$= \frac{x\big(C + (\gamma + 2\eta) + \beta(\gamma - \eta)\big) + \big(C + 3(\gamma + \eta) + \beta(\gamma - \eta)\big)}{x\big(C + \beta(2\gamma + \eta)\big) + \big(2C + \beta(4\gamma + 2\eta)\big)}$$

$$= \frac{x\big(2\gamma + (2 - \beta)\eta\big) + \big(4\gamma + (3 - \beta)\eta\big)}{x\big((1 + \beta)\gamma + \beta\eta\big) + \big((2 + 2\beta)\gamma + 2\beta\eta\big)}$$

$$\leq \frac{2\gamma + (2 - \beta)\eta}{(1 + \beta)\gamma + \beta\eta}. \quad \text{(when } x \to \infty) \tag{31}$$

**Case II.** $\eta > \gamma$. By Lemma 4.3, for each $k = -1, 0, ..., x$, the total regular cost in the $(4k + 3)$-th time interval (which is in an off phase and has length at most $T$) is less than $2\gamma + \eta$:

$$s_{4k+3} < 2\gamma + \eta. \tag{32}$$

On the other hand, the total regular cost in any time interval in an on phase is non-negative. As a result, we have

$$\frac{\mathsf{PFSUM}\big(\sigma; [\tau_i, \tau_{i+x+1} + T)\big)}{\mathsf{OPT}\big(\sigma; [\tau_i, \tau_{i+x+1} + T)\big)} < \frac{(x+1)C + (x+2)(2\gamma + \eta)}{(x+2)C + \beta(x+2)(2\gamma + \eta)} \quad \text{(by (26)) and (32))}$$

$$= \frac{x\big(C + (2\gamma + \eta)\big) + \big(C + (4\gamma + 2\eta)\big)}{x\big(C + \beta(2\gamma + \eta)\big) + \big(2C + \beta(4\gamma + 2\eta)\big)}$$

$$\leq \frac{C + (2\gamma + \eta)}{C + \beta(2\gamma + \eta)} \quad \text{(when } x \to \infty)$$

$$= \frac{(3 - \beta)\gamma + \eta}{(1 + \beta)\gamma + \beta\eta}. \tag{33}$$

The result follows from (31) and (33).

A tight example is given in Figure 14, where $x \to \infty$. The upper bound is achieved when $0 \leq \eta \leq \gamma$, for each $k = -1, ..., x - 1$, $s_{4k+3} = \gamma + 2\eta$, $s_{4x+3} \to 2\gamma + \eta$; for each $k = 0, ..., x$, $s_{4k} = \gamma - \eta$, $s_{4k+1} = s_{4k+2} = 0$, or when $\eta > \gamma$, for each $k = -1, ..., x$, $s_{4k+3} = 2\gamma + \eta$ and for each $k = 0, ..., x$, $s_{4k} = s_{4k+1} = s_{4k+2} = 0$.

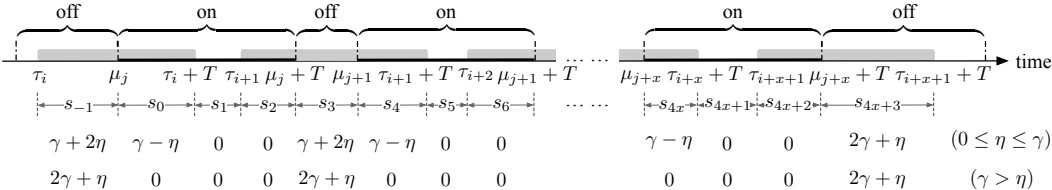

Figure 14: A tight example for Pattern IV. The shaded rectangle is the valid time of a Bahncard purchased by OPT.

### A.4.2 Proof for Pattern IV

**Proposition Restated.** If OPT purchases $x + 1$ Bahncards ($x \geq 0$) in successive on phases starting from $E_j$, where (i) the first Bahncard has its purchase time $\tau_i$ falling in the off phase of $E_{j-1}$ and its expiry time $\tau_i + T$ falling in the on phase of $E_j$, and (ii) for each $k = 1, ..., x$, the $(k + 1)$-th Bahncard has its purchase time $\tau_{i+k}$ falling in the on phase of $E_{j+k-1}$ and its expiry time $\tau_{i+k} + T$ falling in the on phase of $E_{j+k}$, and (iii) OPT does not purchase any new Bahncard in the on phase of $E_{j+x}$, then

$$\frac{\mathsf{PFSUM}\big(\sigma; [\tau_i, \mu_{j+x} + T)\big)}{\mathsf{OPT}\big(\sigma; [\tau_i, \mu_{j+x} + T)\big)} \leq \begin{cases} \frac{2\gamma + (2-\beta)\eta}{(1+\beta)\gamma + \beta\eta} & 0 \leq \eta \leq \gamma, \\ \frac{(3-\beta)\gamma + \eta}{(1+\beta)\gamma + \beta\eta} & \eta > \gamma, \end{cases} \tag{34}$$

where the upper bound is tight (achievable) for any $x$.

**Proof.** As shown in Figure 15, we divide $[\tau_i, \mu_{j+x} + T)$ into $4x + 3$ time intervals, where each time interval starts and ends with the time when OPT or PFSUM purchases a Bahncard or a Bahncard purchased by OPT or PFSUM expires. Let these intervals be indexed by $-1, 0, 1, 2, ..., 4x + 1$, and let $s_{-1}, s_0, s_1, s_2, ..., s_{4x+1}$ denote the total regular costs in these intervals respectively. By definition, it is easy to see that for each $k = 0, ..., x - 1$, the $(4k + 3)$-th time interval, i.e., $[\mu_{j+k} + T, \mu_{j+k+1})$ (which is an off phase), is shorter than $T$ since it is within the valid time of a Bahncard purchased by OPT.

Figure 15: Illustration for Pattern IV. The shaded rectangle is the valid time of a Bahncard purchased by OPT.

First, we observe that the cost ratio $\mathsf{PFSUM}\big(\sigma; [\tau_i, \mu_{j+x} + T)\big)/\mathsf{OPT}\big(\sigma; [\tau_i, \mu_{j+x} + T)\big)$ is less than $1/\beta$, as shown below.

$$\frac{1}{\beta} - \frac{\mathsf{PFSUM}\big(\sigma; [\tau_i, \mu_{j+x} + T)\big)}{\mathsf{OPT}\big(\sigma; [\tau_i, \mu_{j+x} + T)\big)}$$

$$= \frac{(x+1)C + \beta\Big[(s_{-1}+s_0) + \sum_{k=0}^{x-1}\big(s_{4k+2}+s_{4k+3}+s_{4k+4}\big)\Big] + \sum_{k=0}^{x} s_{4k+1}}{\beta \cdot \mathsf{OPT}\big(\sigma; [\tau_i, \mu_{j+x}+T)\big)}$$

$$- \frac{\beta\Big[(x+1)C + \beta\big[\sum_{k=0}^{x-1}\big(s_{4k}+s_{4k+1}+s_{4k+2}\big) + \big(s_{4x}+s_{4x+1}\big)\big] + \sum_{k=-1}^{x-1} s_{4k+3}\Big]}{\beta \cdot \mathsf{OPT}\big(\sigma; [\tau_i, \mu_{j+x}+T)\big)}$$

$$= \frac{(x+1)C + \beta\Big[\sum_{k=0}^{x} s_{4k} + \sum_{k=0}^{x-1} s_{4k+2} + \sum_{k=-1}^{x-1} s_{4k+3}\Big] + \sum_{k=0}^{x} s_{4k+1}}{\beta \cdot \mathsf{OPT}\big(\sigma; [\tau_i, \mu_{j+x}+T)\big)}$$

$$- \frac{\beta(x+1)C + \beta^2\Big[\sum_{k=0}^{x} s_{4k} + \sum_{k=0}^{x} s_{4k+1} + \sum_{k=0}^{x-1} s_{4k+2}\Big] + \beta\sum_{k=-1}^{x-1} s_{4k+3}}{\beta \cdot \mathsf{OPT}\big(\sigma; [\tau_i, \mu_{j+x}+T)\big)}$$

$$= \frac{(1-\beta)(x+1)C + \beta(1-\beta)\big(\sum_{k=0}^{x} s_{4k} + \sum_{k=0}^{x-1} s_{4k+2}\big) + (1-\beta^2)\sum_{k=0}^{x} s_{4k+1}}{\beta \cdot \mathsf{OPT}\big(\sigma; [\tau_i, \mu_{j+x}+T)\big)}$$

$$> 0. \quad \text{(since } 0 \le \beta < 1)$$

Thus, the following inequality always holds:

$$\frac{\mathsf{PFSUM}\big(\sigma; [\tau_i, \mu_{j+x}+T)\big)}{\mathsf{OPT}\big(\sigma; [\tau_i, \mu_{j+x}+T)\big)} < \frac{1}{\beta}. \tag{35}$$

Next, we analyze the upper bound of the cost ratio. There are two cases to consider.

**Case I.** $0 \le \eta \le \gamma$. By Corollary 3.2, for each $k = -1, 0, ..., x$, the $T$-future-cost at $\tau_{i+k+1}$ is at least $\gamma$, i.e.,

$$\begin{cases} s_{-1} + s_0 \ge \gamma, \\ s_{4k+2} + s_{4k+3} + s_{4k+4} \ge \gamma \quad \text{for each } k = 0, ..., x-1. \end{cases} \tag{36}$$

By Lemma 4.2, we have

$$\begin{cases} s_{4k} + s_{4k+1} + s_{4k+2} \ge \gamma - \eta \quad \text{for each } k = 0, ..., x-1, \\ s_{4x} + s_{4x+1} \ge \gamma - \eta. \end{cases} \tag{37}$$

Note that all the travel requests in the $(4k+2)$-th ($k = 0, ..., x-1$) and the $(4k+4)$-th ($k = -1, ..., x-1$) time intervals are reduced requests of both PFSUM and OPT. Thus, to maximize the cost ratio in $[\tau_i, \mu_{j+x}+T)$, we should minimize $s_{4k+2}$ and $s_{4k+4}$. If they are greater than $\gamma$, the cost ratio can be increased by decreasing $s_{4k+2}$ or $s_{4k+4}$ to $\gamma$ without violating (36) and (37). Thus, for the purpose of deriving an upper bound on the cost ratio, we can assume that

$$\begin{cases} s_{4k+2} \le \gamma \quad \text{for each } k = 0, ..., x-1, \\ s_{4k+4} \le \gamma \quad \text{for each } k = -1, ..., x-1. \end{cases} \tag{38}$$

It follows from (38), and Lemma 4.4 that

$$\begin{cases} s_{4k+3} + s_{4k+4} \le 2\gamma + \eta \quad\quad\quad \text{for } k = -1, \\ s_{4k+2} + s_{4k+3} + s_{4k+4} \le 2\gamma + \eta \quad \text{for each } k = 0, ..., x-1. \end{cases} \tag{39}$$

As a result, we have

$$\frac{\mathsf{PFSUM}\big(\sigma; [\tau_i, \mu_{j+x}+T)\big)}{\mathsf{OPT}\big(\sigma; [\tau_i, \mu_{j+x}+T)\big)}$$

$$= \frac{(x+1)C + \beta\Big[\sum_{k=0}^{x-1}\big(s_{4k}+s_{4k+1}+s_{4k+2}\big) + \big(s_{4x}+s_{4x+1}\big)\Big] + \sum_{k=-1}^{x-1} s_{4k+3}}{(x+1)C + \beta\Big[(s_{-1}+s_0) + \sum_{k=0}^{x-1}\big(s_{4k+2}+s_{4k+3}+s_{4k+4}\big)\Big] + \sum_{k=0}^{x} s_{4k+1}}$$

$$= \frac{(x+1)C + (s_{-1}+s_0) + \sum_{k=0}^{x-1}\big(s_{4k+2}+s_{4k+3}+s_{4k+4}\big) + \beta\sum_{k=0}^{x} s_{4k+1}}{(x+1)C + \beta\Big[(s_{-1}+s_0) + \sum_{k=0}^{x-1}\big(s_{4k+2}+s_{4k+3}+s_{4k+4}\big)\Big] + \sum_{k=0}^{x} s_{4k+1}}$$

$$-\frac{(1-\beta)\left[\sum_{k=0}^{x-1}\left(s_{4k}+s_{4k+2}\right)+s_{4x}\right]}{(x+1)C+\beta\left[(s_{-1}+s_0)+\sum_{k=0}^{x-1}\left(s_{4k+2}+s_{4k+3}+s_{4k+4}\right)\right]+\sum_{k=0}^{x}s_{4k+1}}$$

$$\leq\frac{(x+1)C+(x+1)(2\gamma+\eta)+\beta\sum_{k=0}^{x}s_{4k+1}-(1-\beta)\left[\sum_{k=0}^{x-1}\left(s_{4k}+s_{4k+2}\right)+s_{4x}\right]}{(x+1)C+\beta(x+1)(2\gamma+\eta)+\sum_{k=0}^{x}s_{4k+1}}$$

$$\text{(by (35) and (39))}$$

$$=\frac{(x+1)(C+2\gamma+\eta)+\sum_{k=0}^{x}s_{4k+1}-(1-\beta)\left[\sum_{k=0}^{x-1}\left(s_{4k}+s_{4k+1}+s_{4k+2}\right)+(s_{4x}+s_{4x+1})\right]}{(x+1)C+\beta(x+1)(2\gamma+\eta)+\sum_{k=0}^{x}s_{4k+1}}$$

$$\leq\frac{(x+1)C+(x+1)(2\gamma+\eta)+\sum_{k=0}^{x}s_{4k+1}-(1-\beta)(x+1)(\gamma-\eta)}{(x+1)C+\beta(x+1)(2\gamma+\eta)+\sum_{k=0}^{x}s_{4k+1}}\quad\text{(by (37))}$$

$$\leq\frac{(x+1)C+(x+1)(2\gamma+\eta)-(1-\beta)(x+1)(\gamma-\eta)}{(x+1)C+\beta(x+1)(2\gamma+\eta)}\quad\left(\text{since }\sum_{k=0}^{x}s_{4k+1}\geq0\right)$$

$$=\frac{2\gamma+(2-\beta)\eta}{(1+\beta)\gamma+\beta\eta}.\tag{40}$$

**Case II.** $\eta>\gamma$. By Lemma 4.3, for each $k=-1,0,...,x$, the total regular cost in the $(4k+3)$-th time interval (which is in an off phase and has length at most $T$) is less than $2\gamma+\eta$:

$$s_{4k+3}<2\gamma+\eta.\tag{41}$$

On the other hand, the total regular cost in any time interval in an on phase is non-negative. As a result, we have

$$\frac{\mathsf{PFSUM}\left(\sigma;[\tau_i,\mu_{j+x}+T)\right)}{\mathsf{OPT}\left(\sigma;[\tau_i,\mu_{j+x}+T)\right)}\leq\frac{(x+1)C+(x+1)(2\gamma+\eta)}{(x+1)C+\beta(x+1)(2\gamma+\eta)}\quad\text{by (35) and (41)}$$

$$=\frac{(3-\beta)\gamma+\eta}{(1+\beta)\gamma+\beta\eta}.\tag{42}$$

The result follows from (40) and (42).

A tight example is given in Figure 16, where $x=0$. The upper bound is achieved when $0\leq\eta\leq\gamma$, $s_{-1}\to\gamma+2\eta$, $s_0=\gamma-\eta$, and $s_1=0$, or when $\eta>\gamma$, $s_{-1}\to2\gamma+\eta$, and $s_0=s_1=0$.

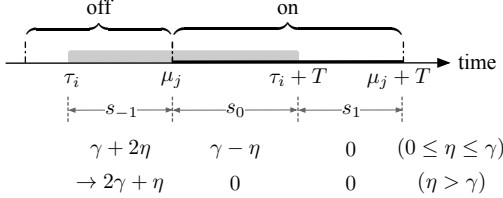

Figure 16: A tight example for Pattern IV. The shaded rectangle is the valid time of a Bahncard purchased by OPT.

### A.4.3 Proof for Pattern V

**Proposition Restated.** If (i) there is no Bahncard purchased by OPT expiring in the on phase of $E_j$, (ii) OPT purchases $x+1$ Bahncards ($x\geq0$) in successive on phases starting from $E_j$, where (a) for each $k=0,...,x-1$, the $(k+1)$-th Bahncard has its purchase time $\tau_{i+k}$ falling in the on phase of $E_{j+k}$ and its expiry time $\tau_{i+k}+T$ falling in the on phase of $E_{j+k+1}$, and (b) the $(x+1)$-th Bahncard has its purchase time $\tau_{i+x}$ falling in the on phase of $E_{j+x}$ and its expiry time $\tau_{i+x}+T$ falling in the off phase of $E_{j+x}$, then

$$\frac{\mathsf{PFSUM}(\sigma;[\mu_j,\tau_{i+x}+T))}{\mathsf{OPT}(\sigma;[\mu_j,\tau_{i+x}+T))}\leq\begin{cases}\frac{2\gamma+(2-\beta)\eta}{(1+\beta)\gamma+\beta\eta}&0\leq\eta\leq\gamma,\\\frac{(3-\beta)\gamma+\eta}{(1+\beta)\gamma+\beta\eta}&\eta>\gamma,\end{cases}\tag{43}$$

where the upper bound is tight (achievable) for any $x$.

**Proof.** As shown in Figure 17, we divide $[\mu_j, \tau_{i+x} + T)$ into $4x + 3$ time intervals, where each time interval starts and ends with the time when OPT or PFSUM purchases a Bahncard or a Bahncard purchased by OPT or PFSUM expires. Let these intervals be indexed by $1, 2, ..., 4x + 3$, and let $s_1, s_2, ..., s_{4x+3}$ denote the total regular costs in these intervals respectively. By definition, it is easy to see that for each $k = 0, ..., x - 1$, the $(4k + 3)$-th time interval, i.e., $[\mu_{j+k} + T, \mu_{j+k+1})$ (which is an off phase), is shorter than $T$ since it is within the valid time of a Bahncard purchased by OPT.

Next, we analyze the upper bound of the cost ratio. There are two cases to consider.

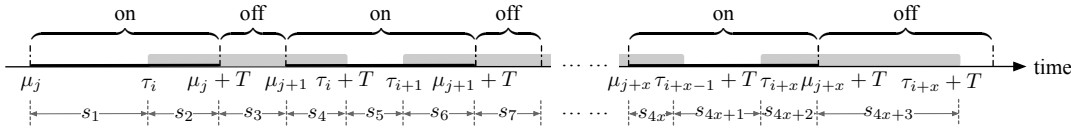

Figure 17: Illustration for Pattern V. The shaded rectangle is the valid time of a Bahncard purchased by OPT.

First, we observe that the cost ratio $\mathsf{PFSUM}(\sigma; [\mu_j, \tau_{i+x} + T))/\mathsf{OPT}(\sigma; [\mu_j, \tau_{i+x} + T))$ is less than $1/\beta$, as shown below.

$$
\frac{1}{\beta} - \frac{\mathsf{PFSUM}(\sigma; [\mu_j, \tau_{i+x} + T))}{\mathsf{OPT}(\sigma; [\mu_j, \tau_{i+x} + T))}
$$

$$
= \frac{(x+1)C + \beta\left[\sum_{k=0}^{x-1}\left(s_{4k+2} + s_{4k+3} + s_{4k+4}\right) + \left(s_{4x+2} + s_{4x+3}\right)\right] + \sum_{k=0}^{x} s_{4k+1}}{\beta \cdot \mathsf{OPT}(\sigma; [\mu_j, \tau_{i+x} + T))}
$$

$$
- \frac{\beta\left[(x+1)C + \beta\left[\sum_{k=0}^{x-1}\left(s_{4k+1} + s_{4k+2} + s_{4k+4}\right) + \left(s_{4x+1} + s_{4x+2}\right)\right] + \sum_{k=0}^{x} s_{4k+3}\right]}{\beta \cdot \mathsf{OPT}(\sigma; [\mu_j, \tau_{i+x} + T))}
$$

$$
= \frac{(x+1)C + \beta\left[\sum_{k=0}^{x} s_{4k+2} + \sum_{k=0}^{x} s_{4k+3} + \sum_{k=0}^{x-1} s_{4k+4}\right] + \sum_{k=0}^{x} s_{4k+1}}{\beta \cdot \mathsf{OPT}(\sigma; [\mu_j, \tau_{i+x} + T))}
$$

$$
- \frac{\beta(x+1)C + \beta^2\left[\sum_{k=0}^{x} s_{4k+1} + \sum_{k=0}^{x} s_{4k+2} + \sum_{k=0}^{x-1} s_{4k+4}\right] + \beta \sum_{k=0}^{x} s_{4k+3}}{\beta \cdot \mathsf{OPT}(\sigma; [\mu_j, \tau_{i+x} + T))}
$$

$$
= \frac{(1-\beta)(x+1)C + \beta(1-\beta)\left(\sum_{k=0}^{x} s_{4k+2} + \sum_{k=0}^{x-1} s_{4k+4}\right) + (1-\beta^2)\sum_{k=0}^{x} s_{4k+1}}{\beta \cdot \mathsf{OPT}(\sigma; [\mu_j, \tau_{i+x} + T))}
$$

$$
> 0. \quad \text{(since } 0 \leq \beta < 1\text{)}
$$

Thus, the following inequality always holds:

$$
\frac{\mathsf{PFSUM}(\sigma; [\mu_j, \tau_{i+x} + T))}{\mathsf{OPT}(\sigma; [\mu_j, \tau_{i+x} + T))} < \frac{1}{\beta}. \tag{44}
$$

**Case I.** $0 \leq \eta \leq \gamma$. By Corollary 3.2, for each $k = 0, ..., x$, the $T$-future-cost at $\tau_{i+k}$ is at least $\gamma$, i.e.,

$$
\begin{cases} s_{4k+2} + s_{4k+3} + s_{4k+4} \geq \gamma & \text{for each } k = 0, ..., x - 1, \\ s_{4x+2} + s_{4x+3} \geq \gamma. \end{cases} \tag{45}
$$

By Lemma 4.2, we have

$$
\begin{cases} s_1 + s_2 \geq \gamma - \eta, \\ s_{4k} + s_{4k+1} + s_{4k+2} \geq \gamma - \eta & \text{for each } k = 1, ..., x. \end{cases} \tag{46}
$$

Note that all the travel requests in the $(4k + 2)$-th (for $k = 0, ..., x$) and the $(4k + 4)$-th (for $k = 0, ..., x - 1$) time intervals are reduced requests of both PFSUM and OPT. Thus, to maximize the cost ratio in $[\mu_j, \tau_{i+x} + T)$, we should minimize these $s_{4k+2}$'s and $s_{4k+4}$'s. If they are greater than $\gamma$, the cost ratio can be increased by decreasing $s_{4k+2}$ or $s_{4k+4}$ to $\gamma$ without violating (45) and (46). Thus, for the purpose of deriving an upper bound on the cost ratio, we can assume that

$$\begin{cases} s_{4k+2} \leq \gamma & \text{for each } k = 0, ..., x, \\ s_{4k+4} \leq \gamma & \text{for each } k = 0, ..., x - 1. \end{cases} \tag{47}$$

It follows from (47) and Lemma 4.4 that

$$\begin{cases} s_{4k+2} + s_{4k+3} + s_{4k+4} \leq 2\gamma + \eta & \text{for each } k = 0, ..., x - 1, \\ s_{4k+2} + s_{4k+3} \leq 2\gamma + \eta & \text{for } k = x. \end{cases} \tag{48}$$

As a result, we have

$$\frac{\mathsf{PFSUM}\big(\sigma; [\mu_j, \tau_{i+x} + T)\big)}{\mathsf{OPT}\big(\sigma; [\mu_j, \tau_{i+x} + T)\big)}$$

$$= \frac{(x+1)C + \beta\Big[\sum_{k=0}^{x-1}\big(s_{4k+1} + s_{4k+2} + s_{4k+4}\big) + \big(s_{4x+1} + s_{4x+2}\big)\Big] + \sum_{k=0}^{x} s_{4k+3}}{(x+1)C + \beta\Big[\sum_{k=0}^{x-1}\big(s_{4k+2} + s_{4k+3} + s_{4k+4}\big) + \big(s_{4x+2} + s_{4x+3}\big)\Big] + \sum_{k=0}^{x} s_{4k+1}}$$

$$= \frac{(x+1)C + \sum_{k=0}^{x-1}\big(s_{4k+2} + s_{4k+3} + s_{4k+4}\big) + \big(s_{4x+2} + s_{4x+3}\big) + \beta\sum_{k=0}^{x} s_{4k+1}}{(x+1)C + \beta\Big[\sum_{k=0}^{x-1}\big(s_{4k+2} + s_{4k+3} + s_{4k+4}\big) + \big(s_{4x+2} + s_{4x+3}\big)\Big] + \sum_{k=0}^{x} s_{4k+1}}$$

$$- \frac{(1-\beta)\Big[\sum_{k=0}^{x-1}\big(s_{4k+2} + s_{4k+4}\big) + s_{4x+2}\Big]}{(x+1)C + \beta\Big[\sum_{k=0}^{x-1}\big(s_{4k+2} + s_{4k+3} + s_{4k+4}\big) + \big(s_{4x+2} + s_{4x+3}\big)\Big] + \sum_{k=0}^{x} s_{4k+1}}$$

$$\leq \frac{(x+1)C + (x+1)(2\gamma + \eta) + \beta\sum_{k=0}^{x} s_{4k+1} - (1-\beta)\Big[\sum_{k=0}^{x-1}\big(s_{4k+2} + s_{4k+4}\big) + s_{4x+2}\Big]}{(x+1)C + \beta(x+1)(2\gamma + \eta) + \sum_{k=0}^{x} s_{4k+1}}$$

(by (44) and (48))

$$= \frac{(x+1)C + (x+1)(2\gamma + \eta) + \sum_{k=0}^{x} s_{4k+1} - (1-\beta)\Big[(s_1 + s_2) + \sum_{k=1}^{x}\big(s_{4k} + s_{4k+1} + s_{4k+2}\big)\Big]}{(x+1)C + \beta(x+1)(2\gamma + \eta) + \sum_{k=0}^{x} s_{4k+1}}$$

$$\leq \frac{(x+1)C + (x+1)(2\gamma + \eta) + \sum_{k=0}^{x} s_{4k+1} - (1-\beta)(x+1)(\gamma - \eta)}{(x+1)C + \beta(x+1)(2\gamma + \eta) + \sum_{k=0}^{x} s_{4k+1}} \quad \text{(by (46))}$$

$$\leq \frac{(x+1)C + (x+1)(2\gamma + \eta) - (1-\beta)(x+1)(\gamma - \eta)}{(x+1)C + \beta(x+1)(2\gamma + \eta)} \quad \Big(\text{since } \sum_{k=0}^{x} s_{4k+1} \geq 0\Big)$$

$$= \frac{2\gamma + (2-\beta)\eta}{(1+\beta)\gamma + \beta\eta}. \tag{49}$$

**Case II.** $\eta > \gamma$. By Lemma 4.3, for each $k = 0, ..., x$, the total regular cost in the $(4k + 3)$-th time interval (which is in an off phase and has length at most $T$) is less than $2\gamma + \eta$:

$$s_{4k+3} < 2\gamma + \eta. \tag{50}$$

On the other hand, the total regular cost in any time interval in an on phase is non-negative. As a result, we have

$$\frac{\mathsf{FSUM}\big(\sigma; [\mu_j, \tau_{i+x} + T)\big)}{\mathsf{OPT}\big(\sigma; [\mu_j, \tau_{i+x} + T)\big)} < \frac{(x+1)C + (x+1)(2\gamma + \eta)}{(x+1)C + \beta(x+1)(2\gamma + \eta)} \quad \text{by (44) and (50)}$$

$$= \frac{(3-\beta)\gamma + \eta}{(1+\beta)\gamma + \beta\eta}. \tag{51}$$

The result follows from (49) and (51).

A tight example is given in Figure 18, where $x = 0$. The upper bound is achieved when $0 \leq \eta \leq \gamma$, $s_1 = 0$, $s_2 = \gamma - \eta$, and $s_3 \to \gamma + 2\eta$, or when $\eta > \gamma$, $s_1 = s_2 = 0$, and $s_3 \to 2\gamma + \eta$.

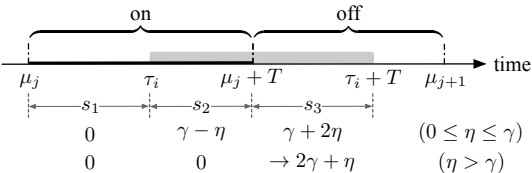

Figure 18: A tight example for Pattern V. The shaded rectangle is the valid time of a Bahncard purchased by OPT.

## A.5 Proof of Proposition 4.8 (For Augmented Pattern VI)

**Proposition Restated.** If (i) there is no Bahncard purchased by OPT expiring in the on phase of $E_j$, (ii) OPT purchases $x$ Bahncards ($x \geq 1$) in successive on phases starting from $E_j$, where for each $k = 0, ..., x - 1$, the $(k + 1)$-th Bahncard has its purchase time $\tau_{i+k}$ falling in the on phase of $E_{j+k}$ and its expiry time $\tau_{i+k} + T$ falling in the on phase of $E_{j+k+1}$, and (iii) OPT does not purchase any new Bahncard in the on phase of $E_{j+x}$, and (iv) the total regular cost in the on phase of $E_{j+x}$ is at least $\gamma$, then

$$\frac{\mathsf{PFSUM}(\sigma; [\mu_j, \mu_{j+x} + T))}{\mathsf{OPT}(\sigma; [\mu_j, \mu_{j+x} + T))} \leq \begin{cases} \frac{2\gamma + (2-\beta)\eta}{(1+\beta)\gamma + \beta\eta} & 0 \leq \eta \leq \gamma, \\ \frac{(3-\beta)\gamma + \eta}{(1+\beta)\gamma + \beta\eta} & \eta > \gamma, \end{cases} \tag{52}$$

where the upper bound is tight (achievable) for any $x$.

**Proof.** As shown in Figure 19, we divide $[\mu_j, \mu_{j+x} + T)$ into $4x + 1$ time intervals, where each time interval starts and ends with the time when OPT or PFSUM purchases a Bahncard or a Bahncard purchased by OPT or PFSUM expires. Let these intervals be indexed by $1, 2, ..., 4x + 1$, and let $s_1, s_2, ..., s_{4x+1}$ denote the total regular costs in these intervals respectively. By definition, it is easy to see that for each $k = 0, ..., x - 1$, the $(4k + 3)$-th time interval, i.e., $[\mu_{j+k} + T, \mu_{j+k+1})$ (which is an off phase), is shorter than $T$ since it is within the valid time of a Bahncard purchased by OPT.

Figure 19: Illustration for Proposition 4.8. The shaded rectangle is the valid time of a Bahncard purchased by OPT.

First, we observe that the cost ratio $\mathsf{PFSUM}(\sigma; [\mu_j, \mu_{j+x} + T))/\mathsf{OPT}(\sigma; [\mu_j, \mu_{j+x} + T)) < 1/\beta$, as shown below:

$$\frac{1}{\beta} - \frac{\mathsf{PFSUM}(\sigma; [\mu_j, \mu_{j+x} + T))}{\mathsf{OPT}(\sigma; [\mu_j, \mu_{j+x} + T))}$$

$$= \frac{xC + \beta\left[\sum_{k=0}^{x-1}\left(s_{4k+2} + s_{4k+3} + s_{4k+4}\right)\right] + \sum_{k=0}^{x} s_{4k+1}}{\beta \cdot \left[\mathsf{OPT}(\sigma; [\mu_j, \mu_{j+x} + T))\right]}$$

$$- \frac{\beta\left[(x+1)C + \beta\left[\sum_{k=0}^{x-1}\left(s_{4k+1} + s_{4k+2} + s_{4k+4}\right) + s_{4x+1}\right] + \sum_{k=0}^{x-1} s_{4k+3}\right]}{\beta \cdot \left[\mathsf{OPT}(\sigma; [\mu_j, \mu_{j+x} + T))\right]}$$

$$= \frac{(1-\beta)xC - \beta C + \beta(1-\beta)\sum_{k=0}^{x-1} s_{4k+2} + \beta(1-\beta)\sum_{k=0}^{x-1} s_{4k+4} + (1-\beta^2)\sum_{k=0}^{x} s_{4k+1}}{\beta \cdot \left[\mathsf{OPT}(\sigma; [\mu_j, \mu_{j+x} + T))\right]}$$

$$> \frac{(1-\beta)xC - \beta C + \beta(1-\beta)\left(\sum_{k=0}^{x-1} s_{4k+2} + \sum_{k=0}^{x-2} s_{4k+4}\right)}{\beta \cdot \left[\mathsf{OPT}(\sigma; [\mu_j, \mu_{j+x} + T))\right]}$$

$$+ \frac{(1-\beta^2)\sum_{k=0}^{x-1} s_{4k+1} + \beta(1-\beta)(s_{4x} + s_{4x+1})}{\beta \cdot \left[\mathsf{OPT}\big(\sigma; [\mu_j, \mu_{j+x} + T)\big)\right]} \quad \text{(since } 0 \leq \beta < 1\text{)}$$

$$\geq \frac{(1-\beta)xC - \beta C + \beta(1-\beta)\big(\sum_{k=0}^{x-1} s_{4k+2} + \sum_{k=0}^{x-2} s_{4k+4}\big)}{\beta \cdot \left[\mathsf{OPT}\big(\sigma; [\mu_j, \mu_{j+x} + T)\big)\right]}$$

$$+ \frac{(1-\beta^2)\sum_{k=0}^{x-1} s_{4k+1} + \beta(1-\beta)\gamma}{\beta \cdot \left[\mathsf{OPT}\big(\sigma; [\mu_j, \mu_{j+x} + T)\big)\right]} \quad \text{(since } s_{4x} + s_{4x+1} \geq \gamma\text{)}$$

$$= \frac{(1-\beta)xC + \beta(1-\beta)\big(\sum_{k=0}^{x-1} s_{4k+2} + \sum_{k=0}^{x-2} s_{4k+4}\big) + (1-\beta^2)\sum_{k=0}^{x-1} s_{4k+1}}{\beta \cdot \left[\mathsf{OPT}\big(\sigma; [\mu_j, \mu_{j+x} + T)\big)\right]}$$

$$> 0. \tag{53}$$

Thus, the following inequality always holds:

$$\frac{\mathsf{PFSUM}\big(\sigma; [\mu_j, \mu_{j+x} + T)\big)}{\mathsf{OPT}\big(\sigma; [\mu_j, \mu_{j+x} + T)\big)} < \frac{1}{\beta}. \tag{54}$$

There are two cases to consider.

**Case I.** $0 \leq \eta \leq \gamma$. By Corollary 3.2, for each $k = 0, ..., x-1$, the $T$-future-cost at $\tau_{i+k}$ is at least $\gamma$, i.e.,

$$s_{4k+2} + s_{4k+3} + s_{4k+4} \geq \gamma. \tag{55}$$

On the other hand, by Lemma 4.2 and the (iv)-th condition of the proposition, we have

$$\begin{cases} s_1 + s_2 \geq \gamma - \eta, \\ s_{4k} + s_{4k+1} + s_{4k+2} \geq \gamma - \eta \quad \text{for each } k = 1, ..., x-1, \\ s_{4x} + s_{4x+1} \geq \gamma. \end{cases} \tag{56}$$

Note that for each $k = 0, ..., x-1$, all the travel requests in the $(4k+2)$-th and the $(4k+4)$-th time intervals are reduced requests of both $\mathsf{PFSUM}$ and $\mathsf{OPT}$. Thus, to maximize the cost ratio in $[\mu_j, \mu_{j+x} + T)$, we should minimize $s_{4k+2}$ and $s_{4k+4}$. If they are greater than $\gamma$, the cost ratio can be increased by decreasing $s_{4k+2}$ or $s_{4k+4}$ to $\gamma$ without violating (55) and (56) (except for $s_{4x}$). For $s_{4x}$, if it is greater than $\gamma$, the cost ratio can be increased by decreasing it to $\gamma$ without violating (55) and (56). Thus, for the purpose of deriving an upper bound on the cost ratio, we can assume that

$$\begin{cases} s_{4k+2} \leq \gamma & \text{for each } k = 0, ..., x-1, \\ s_{4k+4} \leq \gamma & \text{for each } k = 0, ..., x-1. \end{cases} \tag{57}$$

It follows from (57) and Lemma 4.4 that, for each $k = 0, ..., x-2$,

$$s_{4k+2} + s_{4k+3} + s_{4k+4} \leq 2\gamma + \eta. \tag{58}$$

For $k = x-1$, we still have

$$s_{4k+2} + s_{4k+3} + s_{4k+4} \leq 2\gamma + \eta. \tag{59}$$

Otherwise, there must exist a time $t$ in the off phase of $E_{j+x-1}$ such that $\mathsf{PFSUM}$ purchases a Bahncard, leading to a contradiction.

As a result,

$$\frac{\mathsf{PFSUM}\big(\sigma; [\mu_j, \mu_{j+x} + T)\big)}{\mathsf{OPT}\big(\sigma; [\mu_j, \mu_{j+x} + T)\big)}$$

$$= \frac{(x+1)C + \beta\left[\sum_{k=0}^{x-1}\big(s_{4k+1} + s_{4k+2} + s_{4k+4}\big) + s_{4x+1}\right] + \sum_{k=0}^{x-1} s_{4k+3}}{xC + \beta\left[\sum_{k=0}^{x-1}\big(s_{4k+2} + s_{4k+3} + s_{4k+4}\big)\right] + \sum_{k=0}^{x} s_{4k+1}}$$

$$= \frac{(x+1)C + \beta \sum_{k=0}^{x} s_{4k+1} + \sum_{k=0}^{x-1} \left(s_{4k+2} + s_{4k+3} + s_{4k+4}\right) - (1-\beta)\sum_{k=0}^{x-1} \left(s_{4k+2} + s_{4k+4}\right)}{xC + \beta \left[\sum_{k=0}^{x-1} \left(s_{4k+2} + s_{4k+3} + s_{4k+4}\right)\right] + \sum_{k=0}^{x} s_{4k+1}}$$

$$= \frac{(x+1)C + \sum_{k=0}^{x} s_{4k+1} + \sum_{k=0}^{x-1} \left(s_{4k+2} + s_{4k+3} + s_{4k+4}\right)}{xC + \beta \left[\sum_{k=0}^{x-1} \left(s_{4k+2} + s_{4k+3} + s_{4k+4}\right)\right] + \sum_{k=0}^{x} s_{4k+1}}$$
$$- \frac{(1-\beta)\left[\sum_{k=0}^{x-1} \left(s_{4k+1} + s_{4k+2} + s_{4k+4}\right) + s_{4x+1}\right]}{xC + \beta \left[\sum_{k=0}^{x-1} \left(s_{4k+2} + s_{4k+3} + s_{4k+4}\right)\right] + \sum_{k=0}^{x} s_{4k+1}}$$

$$= \frac{(x+1)C + \sum_{k=0}^{x} s_{4k+1} + \sum_{k=0}^{x-1} \left(s_{4k+2} + s_{4k+3} + s_{4k+4}\right)}{xC + \beta \left[\sum_{k=0}^{x-1} \left(s_{4k+2} + s_{4k+3} + s_{4k+4}\right)\right] + \sum_{k=0}^{x} s_{4k+1}}$$
$$- \frac{(1-\beta)\left[(s_1 + s_2) + \sum_{k=1}^{x-1} \left(s_{4k} + s_{4k+1} + s_{4k+2}\right) + (s_{4x} + s_{4x+1})\right]}{xC + \beta \left[\sum_{k=0}^{x-1} \left(s_{4k+2} + s_{4k+3} + s_{4k+4}\right)\right] + \sum_{k=0}^{x} s_{4k+1}}$$

$$\leq \frac{(x+1)C + \sum_{k=0}^{x} s_{4k+1} + \sum_{k=0}^{x-1} \left(s_{4k+2} + s_{4k+3} + s_{4k+4}\right) - (1-\beta)\left[x(\gamma - \eta) + \gamma\right]}{xC + \beta \sum_{k=0}^{x-1} \left(s_{4k+2} + s_{4k+3} + s_{4k+4}\right) + \sum_{k=0}^{x} s_{4k+1}} \quad \text{(by (56))}$$

$$\leq \frac{(x+1)C + \sum_{k=0}^{x-1} \left(s_{4k+2} + s_{4k+3} + s_{4k+4}\right) - (1-\beta)\left[x(\gamma - \eta) + \gamma\right]}{xC + \beta \sum_{k=0}^{x-1} \left(s_{4k+2} + s_{4k+3} + s_{4k+4}\right)} \quad \left(\text{since } \sum_{k=0}^{x} s_{4k+1} \geq 0\right)$$

$$\leq \frac{(x+1)C + x(2\gamma + \eta) - (1-\beta)\left[x(\gamma - \eta) + \gamma\right]}{xC + \beta x(2\gamma + \eta)} \quad \text{(by (54), (58), and (59))}$$

$$= \frac{(x+1)(1-\beta)\gamma + x(2\gamma + \eta) - (1-\beta)\left[x(\gamma - \eta) + \gamma\right]}{x(1-\beta)\gamma + \beta x(2\gamma + \eta)} \quad \text{(since } C = (1-\beta)\gamma)$$

$$= \frac{2\gamma + (2 - 2\beta)\eta}{(1+\beta)\gamma + \beta\eta}. \tag{60}$$

**Case II.** $\eta > \gamma$. In this case, the total regular cost in any time interval in an on phase can be zero (except for the on phase of $E_{j+x}$):

$$\begin{cases} s_1 + s_2 \geq 0, \\ s_{4k} + s_{4k+1} + s_{4k+2} \geq 0 \quad \text{for each } k = 1, ..., x-1, \\ s_{4x} + s_{4x+1} \geq \gamma. \end{cases} \tag{61}$$

On the other hand, by Lemma 4.3, for each $k = 0, ..., x-2$, the total regular cost in the $(4k+3)$-th time interval (which is in an off phase and has length at most $T$) is less than $2\gamma + \eta$:

$$s_{4k+3} < 2\gamma + \eta. \tag{62}$$

For $k = x - 1$, we have

$$s_{4x-1} < \gamma + \eta. \tag{63}$$

Otherwise, there must exist a time $t$ in the off phase of $E_{j+x-1}$ such that PFSUM purchases a Bahncard, leading to a contradiction.

As a result,

$$\frac{\mathsf{PFSUM}\left(\sigma; [\mu_j, \mu_{j+x} + T)\right)}{\mathsf{OPT}\left(\sigma; [\mu_j, \mu_{j+x} + T)\right)} < \frac{(x+1)C + (x-1)(2\gamma + \eta) + (\gamma + \eta) + \beta\gamma}{xC + \beta x(2\gamma + \eta)}$$
$$\text{(by (61), (62), and (63))}$$
$$= \frac{(3-\beta)\gamma + \eta}{(1+\beta)\gamma + \beta\eta}. \tag{64}$$

The result follows from (60) and (64).

A tight example for Proposition 4.8 is given in Figure 20, where $x = 1$. The upper bound is achieved when $0 \leq \eta \leq \gamma$, $s_1 = s_5 = 0$, $s_2 = \gamma - \eta$, $s_3 \to 2\eta$, and $s_4 = \gamma$, or when $\eta > \gamma$, $s_1 = s_2 = s_5 = 0$, $s_3 \to \gamma + \eta$, and $s_4 = \gamma$.

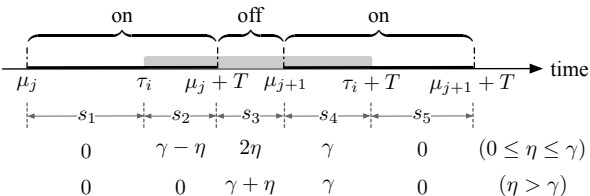

|  | on | off | on |  |  |
|---|---|---|---|---|---|
|  | 0 | $\gamma - \eta$ | $2\eta$ | $\gamma$ | 0 | $(0 \le \eta \le \gamma)$ |
|  | 0 | 0 | $\gamma + \eta$ | $\gamma$ | 0 | $(\eta > \gamma)$ |

Figure 20: A tight example for Proposition 4.8. The shaded rectangle is the valid time of a Bahncard purchased by OPT.

## A.6 Proof of Proposition 4.9

### A.6.1 Non-Augmented Pattern VI Combined with Pattern II

**Proposition Restated.** If (i) OPT purchases a Bahncard at time $\tau_k$ in the off phase of some epoch $E_l$ which expires in the same off phase, (ii) there is no Bahncard purchased by OPT expiring in the on phase of $E_j$ ($j > l$), (iii) OPT purchases $x$ Bahncards ($x \ge 1$) in successive on phases starting from $E_j$, where for each $k = 0, ..., x-1$, the $(k+1)$-th Bahncard has its purchasing time $\tau_{i+k}$ falling in the on phase of $E_{j+k}$ and its expiry time $\tau_{i+k} + T$ falling in the on phase of $E_{j+k+1}$, and (iv) OPT does not purchase any new Bahncard in the on phase of $E_{j+x}$, then

$$\frac{\mathsf{PFSUM}\big(\sigma; [\tau_k, \tau_k + T) \cup [\mu_j, \mu_{j+x} + T)\big)}{\mathsf{OPT}\big(\sigma; [\tau_k, \tau_k + T) \cup [\mu_j, \mu_{j+x} + T)\big)} \le \begin{cases} \frac{2\gamma + (2-\beta)\eta}{(1+\beta)\gamma + \beta\eta} & 0 \le \eta \le \gamma, \\ \frac{(3-\beta)\gamma + \eta}{(1+\beta)\gamma + \beta\eta} & \eta > \gamma, \end{cases} \tag{65}$$

where the upper bound is tight (achievable) for any $x$.

**Proof.** As shown in Figure 21, we divide $[\tau_k, \tau_k + T) \cup [\mu_j, \mu_{j+x} + T)$ into $4x + 2$ time intervals, where each time interval starts and ends with the time when OPT or PFSUM purchases a Bahncard or a Bahncard purchased by OPT or PFSUM expires. Let these intervals be indexed by $-1, 1, ..., 4x+1$, and let $s_{-1}, s_1, ..., s_{4x+1}$ denote the total regular costs in these intervals respectively. By definition, it is easy to see that for each $k = 0, ..., x-1$, the $(4k+3)$-th time interval, i.e., $[\mu_{j+k} + T, \mu_{j+k+1})$ (which is an off phase), is shorter than $T$ since it is within the valid time of a Bahncard purchased by OPT.

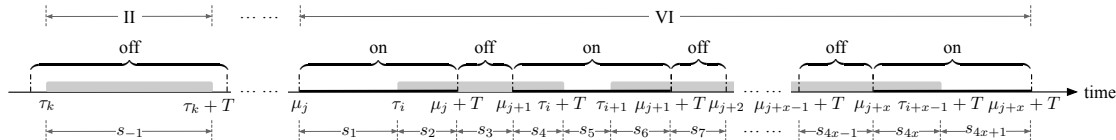

Figure 21: Illustration for Appendix A.6.1. The shaded rectangle is the valid time of a Bahncard purchased by OPT.

First, we observe that the cost ratio $\mathsf{PFSUM}\big(\sigma; [\tau_k, \tau_k + T) \cup [\mu_j, \mu_{j+x} + T)\big)/\mathsf{OPT}\big(\sigma; [\tau_k, \tau_k + T) \cup [\mu_j, \mu_{j+x} + T)\big)$ is less than $1/\beta$, as shown below:

$$\frac{1}{\beta} - \frac{\mathsf{PFSUM}\big(\sigma; [\tau_k, \tau_k + T) \cup [\mu_j, \mu_{j+x} + T)\big)}{\mathsf{OPT}\big(\sigma; [\tau_k, \tau_k + T) \cup [\mu_j, \mu_{j+x} + T)\big)}$$

$$= \frac{(x+1)C + \beta\Big[s_{-1} + \sum_{k=0}^{x-1}(s_{4k+2} + s_{4k+3} + s_{4k+4})\Big] + \sum_{k=0}^{x} s_{4k+1}}{\beta \cdot \mathsf{OPT}\big(\sigma; [\tau_k, \tau_k + T) \cup [\mu_j, \mu_{j+x} + T)\big)}$$

$$- \frac{\beta\Big[(x+1)C + \beta\Big[\sum_{k=0}^{x-1}(s_{4k+1} + s_{4k+2} + s_{4k+4}) + s_{4x+1}\Big] + s_{-1} + \sum_{k=0}^{x-1} s_{4k+3}\Big]}{\beta \cdot \mathsf{OPT}\big(\sigma; [\tau_k, \tau_k + T) \cup [\mu_j, \mu_{j+x} + T)\big)}$$

$$= \frac{(1-\beta)(x+1)C + \beta(1-\beta)\sum_{k=0}^{x-1}(s_{4k+2} + s_{4k+4}) + (1-\beta^2)\sum_{k=0}^{x} s_{4k+1}}{\beta \cdot \mathsf{OPT}\big(\sigma; [\tau_k, \tau_k + T) \cup [\mu_j, \mu_{j+x} + T)\big)}$$

$$> 0. \quad \text{(since } 0 \le \beta < 1)$$

Thus, the following inequality always holds:

$$\frac{\mathsf{PFSUM}\big(\sigma; [\tau_k, \tau_k + T) \cup [\mu_j, \mu_{j+x} + T)\big)}{\mathsf{OPT}\big(\sigma; [\tau_k, \tau_k + T) \cup [\mu_j, \mu_{j+x} + T)\big)} < \frac{1}{\beta}. \tag{66}$$

Next, we analyze the upper bound of the cost ratio. Note that by Lemma 4.3, we have

$$s_{-1} < 2\gamma + \eta. \tag{67}$$

There are two cases to consider.

**Case I.** $0 \le \eta \le \gamma$. By Corollary 3.2, for each $k = 0, ..., x - 1$, the $T$-future-cost at $\tau_{i+k}$ is at least $\gamma$, i.e.,

$$s_{4k+2} + s_{4k+3} + s_{4k+4} \ge \gamma. \tag{68}$$

By Lemma 4.2, we have

$$\begin{cases} s_1 + s_2 \ge \gamma - \eta, \\ s_{4k} + s_{4k+1} + s_{4k+2} \ge \gamma - \eta & \text{for each } k = 1, ..., x - 1, \\ s_{4x} + s_{4x+1} \ge \gamma - \eta. \end{cases} \tag{69}$$

Note that for each $k = 0, ..., x - 1$, all the travel requests in the $(4k + 2)$-th and the $(4k + 4)$-th time intervals are reduced requests of both $\mathsf{PFSUM}$ and $\mathsf{OPT}$. Thus, to maximize the cost ratio in $[\tau_k, \tau_k + T) \cup [\mu_j, \mu_{j+x} + T)$, we should minimize $s_{4k+2}$ and $s_{4k+4}$. If they are greater than $\gamma$, the cost ratio can be increased by decreasing $s_{4k+2}$ or $s_{4k+4}$ to $\gamma$ without violating (68) and (69). Thus, for the purpose of deriving an upper bound on the cost ratio, we can assume that

$$s_{4k+2} \le \gamma, \tag{70}$$
$$s_{4k+4} \le \gamma. \tag{71}$$

It follows from (70), (71) and Lemma 4.4 that, for each $k = 0, ..., x - 1$,

$$s_{4k+2} + s_{4k+3} + s_{4k+4} \le 2\gamma + \eta. \tag{72}$$

As a result, we have

$$\frac{\mathsf{PFSUM}\big(\sigma; [\tau_k, \tau_k + T) \cup [\mu_j, \mu_{j+x} + T)\big)}{\mathsf{OPT}\big(\sigma; [\tau_k, \tau_k + T) \cup [\mu_j, \mu_{j+x} + T)\big)}$$

$$= \frac{(x+1)C + \beta\Big[\sum_{k=0}^{x-1} \big(s_{4k+1} + s_{4k+2} + s_{4k+4}\big) + s_{4x+1}\Big] + s_{-1} + \sum_{k=0}^{x-1} s_{4k+3}}{(x+1)C + \beta\Big[s_{-1} + \sum_{k=0}^{x-1} \big(s_{4k+2} + s_{4k+3} + s_{4k+4}\big)\Big] + \sum_{k=0}^{x} s_{4k+1}}$$

$$< \frac{(x+1)C + (2\gamma + \eta) + \beta\Big[\sum_{k=0}^{x-1} \big(s_{4k+1} + s_{4k+2} + s_{4k+4}\big) + s_{4x+1}\Big] + \sum_{k=0}^{x-1} s_{4k+3}}{(x+1)C + \beta\Big[2\gamma + \eta + \sum_{k=0}^{x-1} \big(s_{4k+2} + s_{4k+3} + s_{4k+4}\big)\Big] + \sum_{k=0}^{x} s_{4k+1}} \quad \text{(by (67))}$$

$$= \frac{(x+1)C + (2\gamma + \eta) + \beta \sum_{k=0}^{x} s_{4k+1} + \sum_{k=0}^{x-1} \big(s_{4k+2} + s_{4k+3} + s_{4k+4}\big) - (1 - \beta) \sum_{k=0}^{x-1} \big(s_{4k+2} + s_{4k+4}\big)}{(x+1)C + \beta\Big[2\gamma + \eta + \sum_{k=0}^{x-1} \big(s_{4k+2} + s_{4k+3} + s_{4k+4}\big)\Big] + \sum_{k=0}^{x} s_{4k+1}}$$

$$\le \frac{(x+1)C + \beta \sum_{k=0}^{x} s_{4k+1} + (x+1)(2\gamma + \eta) - (1 - \beta) \sum_{k=0}^{x-1} \big(s_{4k+2} + s_{4k+4}\big)}{(x+1)C + \beta(x+1)(2\gamma + \eta) + \sum_{k=0}^{x} s_{4k+1}} \quad \text{(by (66) and (72))}$$

$$= \frac{(x+1)C + \sum_{k=0}^{x} s_{4k+1} + (x+1)(2\gamma + \eta) - (1 - \beta)\Big[\sum_{k=0}^{x-1} \big(s_{4k+1} + s_{4k+2} + s_{4k+4}\big) + s_{4x+1}\Big]}{(x+1)C + \beta(x+1)(2\gamma + \eta) + \sum_{k=0}^{x} s_{4k+1}}$$

$$= \frac{(x+1)C + \sum_{k=0}^{x} s_{4k+1} + (x+1)(2\gamma + \eta)}{(x+1)C + \beta(x+1)(2\gamma + \eta) + \sum_{k=0}^{x} s_{4k+1}}$$

$$- \frac{(1-\beta)\Big[(s_1+s_2)+\sum_{k=1}^{x-1}\big(s_{4k}+s_{4k+1}+s_{4k+2}\big)+(s_{4x}+s_{4x+1})\Big]}{(x+1)C+\beta(x+1)(2\gamma+\eta)+\sum_{k=0}^{x}s_{4k+1}}$$

$$\leq \frac{(x+1)C+\sum_{k=0}^{x}s_{4k+1}+(x+1)(2\gamma+\eta)-(1-\beta)(x+1)(\gamma-\eta)}{(x+1)C+\beta(x+1)(2\gamma+\eta)+\sum_{k=0}^{x}s_{4k+1}} \quad \text{(by (69))}$$

$$\leq \frac{(x+1)C+(x+1)(2\gamma+\eta)-(1-\beta)(x+1)(\gamma-\eta)}{(x+1)C+\beta(x+1)(2\gamma+\eta)} \quad \Big(\text{since } \sum_{k=0}^{x}s_{4k+1}\geq 0\Big)$$

$$= \frac{C+(2\gamma+\eta)-(1-\beta)(\gamma-\eta)}{C+\beta(2\gamma+\eta)}$$

$$= \frac{2\gamma+(2-\beta)\eta}{(1+\beta)\gamma+\beta\eta}. \tag{73}$$

**Case II.** $\eta > \gamma$. By Lemma 4.3, for each $k=0,...,x-1$, the total regular cost in the $(4k+3)$-th time interval (which is in an off phase and has length at most $T$) is less than $2\gamma+\eta$:

$$s_{4k+3} < 2\gamma+\eta. \tag{74}$$

On the other hand, the total regular cost in any time interval in an on phase is non-negative. As a result, we have

$$\frac{\mathsf{PFSUM}\big(\sigma;[\tau_k,\tau_k+T)\cup[\mu_j,\mu_{j+x}+T)\big)}{\mathsf{OPT}\big(\sigma;[\tau_k,\tau_k+T)\cup[\mu_j,\mu_{j+x}+T)\big)} < \frac{(x+1)C+(x+1)(2\gamma+\eta)}{(x+1)C+\beta(x+1)(2\gamma+\eta)}$$
$$\text{(by (66), (67), and (74))}$$
$$= \frac{(3-\beta)\gamma+\eta}{(1+\beta)\gamma+\beta\eta}. \tag{75}$$

The result follows from (73) and (75).

### A.6.2 Non-Augmented Pattern VI Combined with Pattern III

**Proposition Restated.** If $\mathsf{OPT}$ purchases $x+2$ Bahncards ($x\geq 0$) starting from the off phase of $E_{l-1}$, where (i) the first Bahncard has its purchase time $\tau_k$ falling in the off phase of $E_{l-1}$ and its expiry time $\tau_k+T$ falling in the on phase of $E_l$, (ii) for each $p=1,...,x$, the $(p+1)$-th Bahncard has its purchase time $\tau_{k+p}$ falling in the on phase of $E_{l+p-1}$ and its expiry time $\tau_{k+p}+T$ falling in the on phase of $E_{l+p}$, (iii) the $(x+2)$-th Bahncard has its purchase time $\tau_{k+x+1}$ falling in the on phase of $E_{l+x}$ and its expiry time $\tau_{k+x+1}+T$ falling in the off phase of $E_{l+x}$, and for some $j\geq l$ and $i\geq k$, $\mathsf{OPT}$ purchases $y$ Bahncards ($y\geq 0$) in successive on phases starting from $E_{j+x+1}$, where for each $p=0,...,y-1$, the $(p+1)$-th Bahncard has its purchasing time $\tau_{i+x+p+2}$ falling in the on phase of $E_{j+x+p+1}$ and its expiry time $\tau_{i+x+p+2}+T$ falling in the on phase of $E_{j+x+p+2}$, and $(v)$ $\mathsf{OPT}$ does not purchase any new Bahncard in the on phase of $E_{j+x+y+1}$, then

$$\frac{\mathsf{PFSUM}\big(\sigma;[\tau_k,\tau_{k+x+1}+T)\cup[\mu_{j+x+1},\mu_{j+x+y+1}+T)\big)}{\mathsf{OPT}\big(\sigma;[\tau_k,\tau_{k+x+1}+T)\cup[\mu_{j+x+1},\mu_{j+x+y+1}+T)\big)} \leq \begin{cases} \frac{2\gamma+(2-\beta)\eta}{(1+\beta)\gamma+\beta\eta} & 0\leq\eta\leq\gamma, \\ \frac{(3-\beta)\gamma+\eta}{(1+\beta)\gamma+\beta\eta} & \eta>\gamma. \end{cases} \tag{76}$$

**Proof.** As shown in Figure 22, we divide $[\tau_k,\tau_{k+x+1}+T)\cup[\mu_{j+x+1},\mu_{j+x+y+1}+T)$ into $4(x+y)+6$ time intervals, where each time intervals starts and ends with the time when $\mathsf{OPT}$ or $\mathsf{PFSUM}$ purchases a Bahncard or a Bahncard purchased by $\mathsf{OPT}$ or $\mathsf{PFSUM}$ expires. The first $4x+5$ time intervals, from time $\tau_i$ to $\tau_{i+x+1}+T$, are concerned time duration of Pattern III. The total regular costs in these intervals are denoted by $s_{-1},s_0,s_1,...,s_{4x+3}$, respectively. $[\mu_{j+x+1},\mu_{j+x+y+1}+T)$ is the concerned interval of Pattern VI, which is divided into $4y+1$ time intervals. The total regular costs in these intervals are denoted by $q_1,...,q_{4y+1}$, respectively.

For ease of notation, we introduce $\theta_k := s_{4k+2}+s_{4k+3}+s_{4k+4}$, and $\delta_k := q_{4k+2}+q_{4k+3}+q_{4k+4}$. First, we observe that the cost ratio $\mathsf{PFSUM}\big(\sigma;[\tau_k,\tau_{k+x+1}+T)\cup[\mu_{j+x+1},\mu_{j+x+y+1}+T)\big)/\mathsf{OPT}\big(\sigma;[\tau_k,\tau_{k+x+1}+T)\cup[\mu_{j+x+1},\mu_{j+x+y+1}+T)\big)$ is less than $1/\beta$, as shown below:

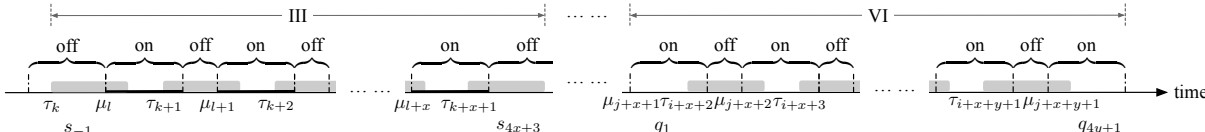

Figure 22: Illustration for Appendix A.6.2. The shaded rectangle is the valid time of a Bahncard purchased by OPT.

$$\frac{1}{\beta} - \frac{\mathsf{PFSUM}\big(\sigma; [\tau_k, \tau_{k+x+1} + T) \cup [\mu_{j+x+1}, \mu_{j+x+y+1} + T)\big)}{\mathsf{OPT}\big(\sigma; [\tau_k, \tau_{k+x+1} + T) \cup [\mu_{j+x+1}, \mu_{j+x+y+1} + T)\big)}$$

$$= \frac{(x+2)C + \beta\Big[s_{-1} + s_0 + s_{4x+2} + s_{4x+3} + \sum_{k=0}^{x-1}\theta_k\Big]}{\beta \cdot \mathsf{OPT}\big(\sigma; [\tau_k, \tau_{k+x+1} + T) \cup [\mu_{j+x+1}, \mu_{j+x+y+1} + T)\big)}$$

$$+ \frac{\sum_{k=0}^{x} s_{4k+1} + s_{4x+4} + yC + \beta\sum_{k=0}^{y-1}\delta_k + \sum_{k=0}^{y} q_{4k+1}}{\beta \cdot \mathsf{OPT}\big(\sigma; [\tau_k, \tau_{k+x+1} + T) \cup [\mu_{j+x+1}, \mu_{j+x+y+1} + T)\big)}$$

$$- \beta \cdot \frac{(x+1)C + \beta\sum_{k=0}^{x}\big(s_{4k} + s_{4k+1} + s_{4k+2}\big) + \sum_{k=-1}^{x} s_{4k+3} + s_{4x+4}}{\beta \cdot \mathsf{OPT}\big(\sigma; [\tau_k, \tau_{k+x+1} + T) \cup [\mu_{j+x+1}, \mu_{j+x+y+1} + T)\big)}$$

$$- \beta \cdot \frac{(y+1)C + \beta\Big[\sum_{k=0}^{y-1}\big(q_{4k+1} + q_{4k+2} + q_{4k+4}\big) + q_{4y+1}\Big] + \sum_{k=0}^{y-1} q_{4k+3}}{\beta \cdot \mathsf{OPT}\big(\sigma; [\tau_k, \tau_{k+x+1} + T) \cup [\mu_{j+x+1}, \mu_{j+x+y+1} + T)\big)}$$

$$= \frac{(1-\beta)(x+y+2)C + \beta(1-\beta)\big(\sum_{k=0}^{x} s_{4k+2} + \sum_{k=0}^{x} s_{4k}\big) + (1-\beta^2)\sum_{k=0}^{x} s_{4k+1}}{\beta \cdot \mathsf{OPT}\big(\sigma; [\tau_k, \tau_{k+x+1} + T) \cup [\mu_{j+x+1}, \mu_{j+x+y+1} + T)\big)}$$

$$+ \frac{\beta(1-\beta)\big(\sum_{k=0}^{y-1} q_{4k+2} + \sum_{k=0}^{y-1} q_{4k+4}\big) + (1-\beta^2)\sum_{k=0}^{y} q_{4k+1}}{\beta \cdot \mathsf{OPT}\big(\sigma; [\tau_k, \tau_{k+x+1} + T) \cup [\mu_{j+x+1}, \mu_{j+x+y+1} + T)\big)}$$

$$> \frac{(1-\beta)(x+y+2)C}{\beta \cdot \mathsf{OPT}\big(\sigma; [\tau_k, \tau_{k+x+1} + T) \cup [\mu_{j+x+1}, \mu_{j+x+y+1} + T)\big)} \quad \text{(since } 0 \le \beta < 1, C > 0)$$

$$> 0.$$

Thus, the following inequality always holds:

$$\frac{\mathsf{PFSUM}\big(\sigma; [\tau_k, \tau_{k+x+1} + T) \cup [\mu_{j+x+1}, \mu_{j+x+y+1} + T)\big)}{\mathsf{OPT}\big(\sigma; [\tau_k, \tau_{k+x+1} + T) \cup [\mu_{j+x+1}, \mu_{j+x+y+1} + T)\big)} < \frac{1}{\beta}. \tag{77}$$

**Case I.** $0 \le \eta \le \gamma$. By Corollary 3.2, the $T$-future-cost at $\tau_{i+k}$ is at least $\gamma$, i.e.,

$$\begin{cases} s_{-5} + s_{-4} \ge \gamma, \\ s_{-2} + s_{-1} \ge \gamma, \\ s_{4k+2} + s_{4k+3} + s_{4k+4} \ge \gamma & \text{for each } k = 0, ..., x-1, \\ q_{4k+2} + q_{4k+3} + q_{4k+4} \ge \gamma & \text{for each } k = 0, ..., y-1. \end{cases} \tag{78}$$

By Lemma 4.2, we have

$$\begin{cases} s_{4k} + s_{4k+1} + s_{4k+2} \ge \gamma - \eta & \text{for each } k = 0, 1, ..., x. \\ q_1 + q_2 \ge \gamma - \eta \\ q_{4y} + q_{4y+1} \ge \gamma - \eta \\ q_{4k} + q_{4k+1} + q_{4k+2} \ge \gamma - \eta & \text{for each } k = 1, 2, ..., y-1. \end{cases} \tag{79}$$

Note that for Pattern III, all the travel requests in the $(4k+2)$-th (for $k = 0, ..., x$) and the $(4k+4)$-th (for $k = -1, ..., x-1$) time intervals are reduced requests of both PFSUM and OPT, similar for Pattern VI. Thus, to maximize the cost ratio in $[\tau_k, \tau_{k+x+1} + T) \cup [\mu_{j+x+1}, \mu_{j+x+y+1} + T)$, we

should minimize these $s_{4k+2}$'s, $s_{4k+4}$'s, $q_{4k+2}$'s, and $q_{4k+4}$'s. If they are greater than $\gamma$, the cost ratio can be increased by decreasing $s_{4k+2}$ or $s_{4k+4}$ to $\gamma$ without violating (78) and (79). Thus, for the purpose of deriving an upper bound on the cost ratio, we can assume that

$$
\begin{cases}
s_{4k+2} \le \gamma & \text{for each } k = 0, ..., x, \\
s_{4k+4} \le \gamma & \text{for each } k = -1, ..., x-1, \\
q_{4k+2} \le \gamma & \text{for each } k = 0, ..., y-1, \\
q_{4k+4} \le \gamma & \text{for each } k = 0, ..., y-1.
\end{cases}
\tag{80}
$$

It follows from (80) and Lemma 4.4 that,

$$
\begin{cases}
s_{-1} + s_0 \le 2\gamma + \eta, \\
s_{4x+2} + s_{4x+3} \le 2\gamma + \eta, \\
\theta_k = s_{4k+2} + s_{4k+3} + s_{4k+4} \le 2\gamma + \eta & \text{for each } k = 0, ..., x-1, \\
\delta_k = q_{4k+2} + q_{4k+3} + q_{4k+4} \le 2\gamma + \eta & \text{for each } k = 0, ..., y-1.
\end{cases}
\tag{81}
$$

As a result, we have

$$
\frac{\mathsf{PFSUM}\big(\sigma; [\tau_k, \tau_{k+x+1} + T) \cup [\mu_{j+x+1}, \mu_{j+x+y+1} + T)\big)}{\mathsf{OPT}\big(\sigma; [\tau_k, \tau_{k+x+1} + T) \cup [\mu_{j+x+1}, \mu_{j+x+y+1} + T)\big)}
$$

$$
= \frac{(x+1)C + s_{-1} + s_0 + s_{4x+2} + s_{4x+3} + \sum_{k=0}^{x-1}\theta_k + \sum_{k=0}^{x} s_{4k+1} - (1-\beta)\sum_{k=0}^{x}(s_{4k} + s_{4k+1} + s_{4k+2})}{(x+2)C + \beta\Big[s_{-1} + s_0 + s_{4x+2} + s_{4x+3} + \sum_{k=0}^{x-1}\theta_k\Big] + \sum_{k=0}^{x} s_{4k+1} + yC + \beta\sum_{k=0}^{y-1}\delta_k + \sum_{k=0}^{y} q_{4k+1}}
$$

$$
+ \frac{(y+1)C + \sum_{k=0}^{y} q_{4k+1} + \sum_{k=0}^{y-1}\delta_k - (1-\beta)\Big[q_1 + q_2 + q_{4y} + q_{4y+1} + \sum_{k=1}^{y-1}(q_{4k} + q_{4k+1} + q_{4k+2})\Big]}{(x+2)C + \beta\Big[s_{-1} + s_0 + s_{4x+2} + s_{4x+3} + \sum_{k=0}^{x-1}\theta_k\Big] + \sum_{k=0}^{x} s_{4k+1} + yC + \beta\sum_{k=0}^{y-1}\delta_k + \sum_{k=0}^{y} q_{4k+1}}
$$

$$
\le \frac{(x+1)(1-\beta)\eta + s_{-1} + s_0 + s_{4x+2} + s_{4x+3} + \sum_{k=0}^{x} s_{4k+1} + \sum_{k=0}^{x-1}\theta_k}{(x+2)C + \beta\Big[s_{-1} + s_0 + s_{4x+2} + s_{4x+3} + \sum_{k=0}^{x} s_{4k+1} + \sum_{k=0}^{x-1}\theta_k\Big] + yC + \beta\sum_{k=0}^{y-1}\delta_k + \sum_{k=0}^{y} q_{4k+1}}
$$

$$
+ \frac{(y+1)(1-\beta)\eta + \sum_{k=0}^{y-1}\delta_k + \sum_{k=0}^{y} q_{4k+1}}{(x+2)C + \beta\Big[s_{-1} + s_0 + s_{4x+2} + s_{4x+3} + \sum_{k=0}^{x} s_{4k+1} + \sum_{k=0}^{x-1}\theta_k\Big] + yC + \beta\sum_{k=0}^{y-1}\delta_k + \sum_{k=0}^{y} q_{4k+1}}
$$

$$
\text{(by (79))}
$$

$$
\le \frac{(x+1)(1-\beta)\eta + s_{-1} + s_0 + s_{4x+2} + s_{4x+3} + \sum_{k=0}^{x-1}\theta_k + (y+1)(1-\beta)\eta + \sum_{k=0}^{y-1}\delta_k}{(x+2)C + \beta(s_{-1} + s_0 + s_{4x+2} + s_{4x+3} + \sum_{k=0}^{x-1}\theta_k) + yC + \beta\sum_{k=0}^{y-1}\delta_k}
$$

$$
\Big(\text{since } \sum_{k=0}^{x} s_{4k+1} \ge 0, \sum_{k=0}^{y} q_{4k+1} \ge 0\Big)
$$

$$
= \frac{(x+y+2)(1-\beta)\eta + (s_{-1} + s_0 + s_{4x+2} + s_{4x+3} + \sum_{k=0}^{x-1}\theta_k) + \sum_{k=0}^{y-1}\delta_k}{(x+y+2)C + \beta\Big[s_{-1} + s_0 + s_{4x+2} + s_{4x+3} + \sum_{k=0}^{x-1}\theta_k\Big] + \beta y\sum_{k=0}^{y-1}\delta_k}
$$

$$
\le \frac{(x+y+2)(1-\beta)\eta + (x+y+2)(2\gamma+\eta)}{(x+y+2)C + \beta(x+y+2)(2\gamma+\eta)} \quad \text{(by (77) and (81))}
$$

$$
= \frac{(1-\beta)\eta + 2\gamma + \eta}{C + \beta(2\gamma+\eta)}
$$

$$
= \frac{2\gamma + (2-\beta)\eta}{(1+\beta)\gamma + \beta\eta}
\tag{82}
$$

**Case II.** $\eta > \gamma$. By Lemma 4.3, for Pattern III, the total regular cost in the $(4k+3)$-th (for $k = -1, ..., x$) time interval (which is in an off phase and has length at most $T$) is less than $2\gamma + \eta$, similar for Pattern VI. Thus we have

$$
\begin{cases}
s_{4k+3} < 2\gamma + \eta & \text{for each } k = -1, ..., x, \\
q_{4k+3} < 2\gamma + \eta & \text{for each } k = 0, 1, ..., y-1.
\end{cases}
\tag{83}
$$

On the other hand, the total regular cost in any time interval in an on phase is non-negative, so it can be minimized to zero. As a result, we have

$$
\begin{aligned}
&\frac{\mathsf{PFSUM}\big(\sigma;[\tau_k,\tau_{k+x+1}+T)\cup[\mu_{j+x+1},\mu_{j+x+y+1}+T)\big)}{\mathsf{OPT}\big(\sigma;[\tau_k,\tau_{k+x+1}+T)\cup[\mu_{j+x+1},\mu_{j+x+y+1}+T)\big)}\\
&\qquad\le\frac{(x+1)C+\sum_{k=-1}^{x}s_{4k+3}+(y+1)C+\sum_{k=0}^{y-1}q_{4k+3}}{(x+2)C+\beta\Big[s_{-1}+s_{4x+3}+\sum_{k=0}^{x-1}s_{4k+3}\Big]+yC+\beta\sum_{k=0}^{y-1}q_{4k+3}}\\
&\qquad<\frac{(x+1)C+(x+2)(2\gamma+\eta)+(y+1)C+y(2\gamma+\eta)}{(x+2)C+\beta(x+2)(2\gamma+\eta)+yC+\beta y(2\gamma+\eta)}\quad\text{(by (77) and (83))}\\
&\qquad=\frac{(x+y+2)C+(x+y+2)(2\gamma+\eta)}{(x+y+2)C+\beta(x+y+2)(2\gamma+\eta)}\\
&\qquad=\frac{(3-\beta)\gamma+\eta}{(1+\beta)\gamma+\beta\eta}
\end{aligned}
\tag{84}
$$

The result follows from (82) and (84).

### A.7 Proof of Proposition 4.10

#### A.7.1 Non-Augmented Pattern VI Combined with Pattern I

**Proposition Restated.** If (i) $\mathsf{OPT}$ purchases a Bahncard at time $\tau_k$ at the beginning of $E_l$, i.e., $\tau_k=\mu_l$, (ii) there is no Bahncard purchased by $\mathsf{OPT}$ expiring in the on phase of $E_j$ ($l<j$), (iii) $\mathsf{OPT}$ purchases $x$ Bahncards ($x\ge 1$) in successive on phases starting from $E_j$, where for each $k=0,...,x-1$, the $(k+1)$-th Bahncard has its purchasing time $\tau_{i+k}$ falling in the on phase of $E_{j+k}$ and its expiry time $\tau_{i+k}+T$ falling in the on phase of $E_{j+k+1}$, and (iv) $\mathsf{OPT}$ does not purchase any new Bahncard in the on phase of $E_{j+x}$, then

$$
\frac{\mathsf{PFSUM}\big(\sigma;[\tau_k,\tau_k+T)\cup[\mu_j,\mu_{j+x}+T)\big)}{\mathsf{OPT}\big(\sigma;[\tau_k,\tau_k+T)\cup[\mu_j,\mu_{j+x}+T)\big)}\le
\begin{cases}
\frac{2\gamma+(2-\beta)\eta}{(1+\beta)\gamma+\beta\eta} & 0\le\eta\le\gamma,\\[2mm]
\frac{(3-\beta)\gamma+\eta}{(1+\beta)\gamma+\beta\eta} & \eta>\gamma,
\end{cases}
\tag{85}
$$

where the upper bound is irrelevant with $x$.

**Proof.** As shown in Figure 23, we divide $[\tau_k,\tau_k+T)\cup[\mu_j,\mu_{j+x}+T)$ into $4x+2$ time intervals, where each time interval starts and ends with the time when $\mathsf{OPT}$ or $\mathsf{PFSUM}$ purchases a Bahncard or a Bahncard purchased by $\mathsf{OPT}$ or $\mathsf{PFSUM}$ expires. Let these intervals be indexed by $-1,1,...,4x+1$, and let $s_{-1},s_1,...,s_{4x+1}$ denote the total regular costs in these intervals respectively. By definition, it is easy to see that for each $k=0,...,x-1$, the $(4k+3)$-th time interval, i.e., $[\mu_{j+k}+T,\mu_{j+k+1})$ (which is an off phase), is shorter than $T$ since it is within the valid time of a Bahncard purchased by $\mathsf{OPT}$.

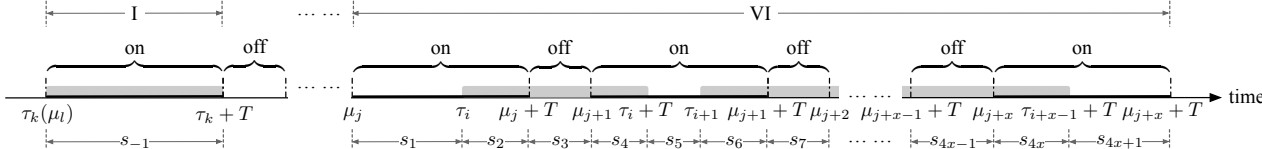

Figure 23: Illustration for Appendix A.7.1. The shaded rectangle is the valid time of a Bahncard purchased by $\mathsf{OPT}$.

First, we observe that the cost ratio $\mathsf{PFSUM}\big(\sigma;[\tau_k,\tau_k+T)\cup[\mu_j,\mu_{j+x}+T)\big)/\mathsf{OPT}\big(\sigma;[\tau_k,\tau_k+T)\cup[\mu_j,\mu_{j+x}+T)\big)$ is less than $1/\beta$, as shown below:

$$
\frac{1}{\beta}-\frac{\mathsf{PFSUM}\big(\sigma;[\tau_k,\tau_k+T)\cup[\mu_j,\mu_{j+x}+T)\big)}{\mathsf{OPT}\big(\sigma;[\tau_k,\tau_k+T)\cup[\mu_j,\mu_{j+x}+T)\big)}
$$

$$= \frac{(x+1)C + \beta\Big[s_{-1} + \sum_{k=0}^{x-1}(s_{4k+2} + s_{4k+3} + s_{4k+4})\Big] + \sum_{k=0}^{x} s_{4k+1}}{\beta \cdot \mathsf{OPT}\big(\sigma; [\tau_k, \tau_k + T) \cup [\mu_j, \mu_{j+x} + T)\big)}$$

$$- \frac{\beta\Big[(x+2)C + \beta\Big[s_{-1} + \sum_{k=0}^{x-1}(s_{4k+1} + s_{4k+2} + s_{4k+4}) + s_{4x+1}\Big] + \sum_{k=0}^{x-1} s_{4k+3}\Big]}{\beta \cdot \mathsf{OPT}\big(\sigma; [\tau_k, \tau_k + T) \cup [\mu_j, \mu_{j+x} + T)\big)}$$

$$= \frac{(1-\beta)(x+1)C - \beta C + \beta(1-\beta)\Big[s_{-1} + \sum_{k=0}^{x-1}(s_{4k+2} + s_{4k+4})\Big] + (1-\beta^2)\sum_{k=0}^{x} s_{4k+1}}{\beta \cdot \mathsf{OPT}\big(\sigma; [\tau_k, \tau_k + T) \cup [\mu_j, \mu_{j+x} + T)\big)}$$

$$\geq \frac{(1-\beta)(x+1)C + \beta(1-\beta)\Big[\sum_{k=0}^{x-1}(s_{4k+2} + s_{4k+4})\Big] + (1-\beta^2)\sum_{k=0}^{x} s_{4k+1}}{\beta \cdot \mathsf{OPT}\big(\sigma; [\tau_k, \tau_k + T) \cup [\mu_j, \mu_{j+x} + T)\big)} \quad \text{(since } s_{-1} \geq \gamma\text{)}$$

$$> 0. \quad \text{(since } 0 \leq \beta < 1\text{)}$$

Thus, the following inequality always holds:

$$\frac{\mathsf{PFSUM}\big(\sigma; [\tau_k, \tau_k + T) \cup [\mu_j, \mu_{j+x} + T)\big)}{\mathsf{OPT}\big(\sigma; [\tau_k, \tau_k + T) \cup [\mu_j, \mu_{j+x} + T)\big)} < \frac{1}{\beta}. \tag{86}$$

There are two cases to consider.

**Case I.** $0 \leq \eta \leq \gamma$. By Corollary 3.2, for each $k = 0, ..., x-1$, the $T$-future-cost at $\tau_{i+k}$ is at least $\gamma$, i.e.,

$$s_{4k+2} + s_{4k+3} + s_{4k+4} \geq \gamma. \tag{87}$$

By Lemma 4.2, we have

$$\begin{cases} s_{-1} \geq \gamma, \\ s_1 + s_2 \geq \gamma - \eta, \\ s_{4k} + s_{4k+1} + s_{4k+2} \geq \gamma - \eta \quad \text{for each } k = 1, ..., x-1, \\ s_{4x} + s_{4x+1} \geq \gamma - \eta. \end{cases} \tag{88}$$

Note that for each $k = 0, ..., x-1$, all the travel requests in the $(4k+2)$-th and the $(4k+4)$-th time intervals are reduced requests of both $\mathsf{PFSUM}$ and $\mathsf{OPT}$. Thus, to maximize the cost ratio in $[\tau_k, \tau_k + T) \cup [\mu_j, \mu_{j+x} + T)$, we should minimize $s_{4k+2}$ and $s_{4k+4}$. If they are greater than $\gamma$, the cost ratio can be increased by decreasing $s_{4k+2}$ or $s_{4k+4}$ to $\gamma$ without violating (87) and (88). Thus, for the purpose of deriving an upper bound on the cost ratio, we can assume that

$$s_{4k+2} \leq \gamma, \tag{89}$$
$$s_{4k+4} \leq \gamma. \tag{90}$$

It follows from (89), (90) and Lemma 4.4 that, for each $k = 0, ..., x-1$,

$$s_{4k+2} + s_{4k+3} + s_{4k+4} \leq 2\gamma + \eta. \tag{91}$$

As a result, we have

$$\frac{\mathsf{PFSUM}\big(\sigma; [\tau_k, \tau_k + T) \cup [\mu_j, \mu_{j+x} + T)\big)}{\mathsf{OPT}\big(\sigma; [\tau_k, \tau_k + T) \cup [\mu_j, \mu_{j+x} + T)\big)}$$

$$= \frac{(x+2)C + \beta\Big[s_{-1} + \sum_{k=0}^{x-1}\big(s_{4k+1} + s_{4k+2} + s_{4k+4}\big) + s_{4x+1}\Big] + \sum_{k=0}^{x-1} s_{4k+3}}{(x+1)C + \beta\Big[s_{-1} + \sum_{k=0}^{x-1}\big(s_{4k+2} + s_{4k+3} + s_{4k+4}\big)\Big] + \sum_{k=0}^{x} s_{4k+1}}$$

$$\leq \frac{(x+2)C + \beta\Big[\gamma + \sum_{k=0}^{x-1}\big(s_{4k+1} + s_{4k+2} + s_{4k+4}\big) + s_{4x+1}\Big] + \sum_{k=0}^{x-1} s_{4k+3}}{(x+1)C + \beta\Big[\gamma + \sum_{k=0}^{x-1}\big(s_{4k+2} + s_{4k+3} + s_{4k+4}\big)\Big] + \sum_{k=0}^{x} s_{4k+1}} \quad \text{(by (88))}$$

$$= \frac{(x+2)C + \beta\gamma + \beta\sum_{k=0}^{x} s_{4k+1} + \sum_{k=0}^{x-1}\left(s_{4k+2} + s_{4k+3} + s_{4k+4}\right) - (1-\beta)\sum_{k=0}^{x-1}\left(s_{4k+2} + s_{4k+4}\right)}{(x+1)C + \beta\left[\gamma + \sum_{k=0}^{x-1}\left(s_{4k+2} + s_{4k+3} + s_{4k+4}\right)\right] + \sum_{k=0}^{x} s_{4k+1}}$$

$$\leq \frac{(x+2)C + \beta\gamma + \beta\sum_{k=0}^{x} s_{4k+1} + x(2\gamma+\eta) - (1-\beta)\sum_{k=0}^{x-1}\left(s_{4k+2} + s_{4k+4}\right)}{(x+1)C + \beta(\gamma + x(2\gamma+\eta)) + \sum_{k=0}^{x} s_{4k+1}} \quad \text{(by (86) and (91))}$$

$$= \frac{(x+2)C + \beta\gamma + \sum_{k=0}^{x} s_{4k+1} + x(2\gamma+\eta) - (1-\beta)\left[\sum_{k=0}^{x-1}\left(s_{4k+1} + s_{4k+2} + s_{4k+4}\right) + s_{4x+1}\right]}{(x+1)C + \beta(\gamma + x(2\gamma+\eta)) + \sum_{k=0}^{x} s_{4k+1}}$$

$$= \frac{(x+2)C + \beta\gamma + \sum_{k=0}^{x} s_{4k+1} + x(2\gamma+\eta)}{(x+1)C + \beta(\gamma + x(2\gamma+\eta)) + \sum_{k=0}^{x} s_{4k+1}}$$

$$\quad - \frac{(1-\beta)\left[(s_1 + s_2) + \sum_{k=1}^{x-1}\left(s_{4k} + s_{4k+1} + s_{4k+2}\right) + (s_{4x} + s_{4x+1})\right]}{(x+1)C + \beta(\gamma + x(2\gamma+\eta)) + \sum_{k=0}^{x} s_{4k+1}}$$

$$\leq \frac{(x+2)C + \beta\gamma + \sum_{k=0}^{x} s_{4k+1} + x(2\gamma+\eta) - (1-\beta)(x+1)(\gamma-\eta)}{(x+1)C + \beta(\gamma + x(2\gamma+\eta)) + \sum_{k=0}^{x} s_{4k+1}} \quad \text{(by (88))}$$

$$\leq \frac{(x+2)C + \beta\gamma + x(2\gamma+\eta) - (1-\beta)(x+1)(\gamma-\eta)}{(x+1)C + \beta(\gamma + x(2\gamma+\eta))} \quad \left(\text{since } \sum_{k=0}^{x} s_{4k+1} \geq 0\right)$$

$$= \frac{x\left(2\gamma + (2-\beta)\eta\right) + \left(\gamma + (1-\beta)\eta\right)}{x\left((1+\beta)\gamma + \beta\eta\right) + \gamma}$$

$$\leq \frac{2\gamma + (2-\beta)\eta}{(1+\beta)\gamma + \beta\eta}. \quad \text{(since } x \geq 1) \tag{92}$$

**Case II.** $\eta > \gamma$. By Lemma 4.3, for each $k = 0, ..., x - 1$, the total regular cost in the $(4k+3)$-th time interval (which is in an off phase and has length at most $T$) is less than $2\gamma + \eta$:

$$s_{4k+3} < 2\gamma + \eta. \tag{93}$$

On the other hand, the total regular cost in any time interval in an on phase is non-negative, except for $s_{-1}$, which is at least $\gamma$:

$$s_{-1} \geq \gamma. \tag{94}$$

As a result, we have

$$\frac{\mathsf{PFSUM}\left(\sigma; [\tau_k, \tau_k + T) \cup [\mu_j, \mu_{j+x} + T)\right)}{\mathsf{OPT}\left(\sigma; [\tau_k, \tau_k + T) \cup [\mu_j, \mu_{j+x} + T))\right)} < \frac{(x+2)C + \beta\gamma + x(2\gamma+\eta)}{(x+1)C + \beta(\gamma + x(2\gamma+\eta))}$$

$$\text{(by (86), (93), and (94))}$$

$$= \frac{x\left(C + 2\gamma + \eta\right) + \left(C + \gamma\right)}{x\left(C + \beta(2\gamma+\eta)\right) + \gamma}$$

$$\leq \frac{(3-\beta)\gamma + \eta}{(1+\beta)\gamma + \beta\eta}. \quad \text{(since } x \geq 1) \tag{95}$$

The result follows from (92) and (95).

### A.7.2 Non-Augmented Pattern VI Combined with Pattern IV

**Proposition Restated.** If $\mathsf{OPT}$ purchases $x + 1$ Bahncards ($x \geq 0$) starting from the off phase of $E_{l-1}$, where (i) the first Bahncard has its purchase time $\tau_k$ falling in the off phase of $E_{l-1}$ and its expiry time $\tau_k + T$ falling in the on phase of $E_l$, (ii) for each $p = 1, ..., x$, the $(p+1)$-th Bahncard has its purchase time $\tau_{k+p}$ falling in the on phase of $E_{l+p-1}$ and its expiry time $\tau_{k+p} + T$ falling in the on phase of $E_{l+p}$, and for some $j \geq l$ and $i \geq k$, $\mathsf{OPT}$ purchases $y$ Bahncards ($y \geq 0$) in successive on phases starting from $E_{j+x+1}$, where for each $p = 0, ..., y - 1$, the $(p+1)$-th Bahncard has its purchasing time $\tau_{i+x+p+1}$ falling in the on phase of $E_{j+x+p+1}$ and its expiry time $\tau_{i+x+p+1} + T$

falling in the on phase of $E_{j+x+p+2}$, and $(v)$ OPT does not purchase any new Bahncard in the on phase of $E_{j+x+y+1}$, then

$$\frac{\mathsf{PFSUM}\big(\sigma;[\tau_k,\mu_{l+x+1})\cup[\mu_{j+x+1},\mu_{j+x+y+1}+T)\big)}{\mathsf{OPT}\big(\sigma;[\tau_k,\mu_{l+x}+T)\cup[\mu_{j+x+1},\mu_{j+x+y+1}+T)\big)}\le\begin{cases}\frac{2\gamma+(2-\beta)\eta}{(1+\beta)\gamma+\beta\eta}&0\le\eta\le\gamma,\\[2mm]\frac{(3-\beta)\gamma+\eta}{(1+\beta)\gamma+\beta\eta}&\eta>\gamma,\end{cases}\tag{96}$$

where the upper bound is tight (achievable) for any $x$.

**Proof.** As shown in Figure 24, we divide $[\tau_k,\mu_{l+x+1})\cup[\mu_{j+x+1},\mu_{j+x+y+1}+T)$ into $4(x+y)+5$ time intervals, where each time intervals starts and ends with the time when OPT or PFSUM purchases a Bahncard or a Bahncard purchased by OPT or PFSUM expires. The first $4x+3$ time intervals, from time $\tau_i$ to $\mu_{j+x}+T$, are concerned time duration of Pattern IV. The $(4x+4)$-th interval is the off phase $[\mu_{l+x}+T,\mu_{l+x+1})$. The total regular costs in these time intervals are denoted by $s_{-1},s_0,s_1,...,s_{4x+1},s_{4x+2}$, respectively. $[\mu_{j+x+1},\mu_{j+x+y+1}+T)$ is the concerned interval of Pattern VI, which is divided into $4y+1$ time intervals. The total regular costs in these intervals are denoted by $q_1,...,q_{4y+1}$, respectively.

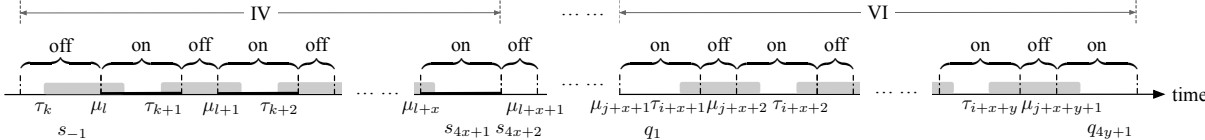

Figure 24: Illustration for Appendix A.7.2. The shaded rectangle is the valid time of a Bahncard purchased by OPT.

According to Proposition 4.10. In IV of (9), the total cost of travel requests in the last on phase of Pattern IV and the following off phase is at least $\gamma$. Thus we have

$$c(\sigma;[\mu_{l+x},\mu_{l+x+1})=s_{4x}+s_{4x+1}+s_{4x+2}\ge\gamma\tag{97}$$

First, we observe that the cost ratio $\mathsf{PFSUM}\big(\sigma;[\tau_k,\mu_{l+x+1})\cup[\mu_{j+x+1},\mu_{j+x+y+1}+T)\big)/\mathsf{OPT}\big(\sigma;[\tau_k,\mu_{l+x+1})\cup[\mu_{j+x+1},\mu_{j+x+y+1}+T)\big)$ is less than $1/\beta$, as shown below:

$$\frac{1}{\beta}-\frac{\mathsf{PFSUM}\big(\sigma;[\tau_k,\mu_{l+x+1})\cup[\mu_{j+x+1},\mu_{j+x+y+1}+T)\big)}{\mathsf{OPT}\big(\sigma;[\tau_k,\mu_{l+x+1})\cup[\mu_{j+x+1},\mu_{j+x+y+1}+T)\big)}$$

$$=\frac{(x+1)C+\beta\Big[s_{-1}+s_0+\sum_{k=0}^{x-1}(s_{4k+2}+s_{4k+3}+s_{4k+4})\Big]+\sum_{k=0}^{x}s_{4k+1}+s_{4x+2}}{\beta\cdot\Big[\mathsf{OPT}\big(\sigma;[\tau_k,\mu_{l+x+1})\cup[\mu_{j+x+1},\mu_{j+x+y+1}+T)\big)\Big]}$$

$$+\frac{yC+\beta\Big[\sum_{k=0}^{y-1}(q_{4k+2}+q_{4k+3}+q_{4k+4})\Big]+\sum_{k=0}^{y}q_{4k+1}}{\beta\cdot\Big[\mathsf{OPT}\big(\sigma;[\tau_k,\mu_{l+x+1})\cup[\mu_{j+x+1},\mu_{j+x+y+1}+T)\big)\Big]}$$

$$-\beta\cdot\frac{(x+1)C+\beta\Big[\sum_{k=0}^{x-1}\big(s_{4k}+s_{4k+1}+s_{4k+2}\big)+\big(s_{4x}+s_{4x+1}\big)\Big]+\sum_{k=-1}^{x-1}s_{4k+3}+s_{4x+2}}{\beta\cdot\Big[\mathsf{OPT}\big(\sigma;[\tau_k,\mu_{l+x+1})\cup[\mu_{j+x+1},\mu_{j+x+y+1}+T)\big)\Big]}$$

$$-\beta\cdot\frac{(y+1)C+\beta\Big[\sum_{k=0}^{y-1}\big(q_{4k+1}+q_{4k+2}+q_{4k+4}\big)+q_{4y+1}\Big]+\sum_{k=0}^{y-1}q_{4k+3}}{\beta\cdot\Big[\mathsf{OPT}\big(\sigma;[\tau_k,\mu_{l+x+1})\cup[\mu_{j+x+1},\mu_{j+x+y+1}+T)\big)\Big]}$$

$$=\frac{(x+1)(1-\beta)C+\beta(1-\beta)(\sum_{k=0}^{x-1}s_{4k+2}+\sum_{k=0}^{x}s_{4k})+(1-\beta^2)\sum_{k=0}^{x}s_{4k+1}+(1-\beta)s_{4x+2}}{\beta\cdot\Big[\mathsf{OPT}\big(\sigma;[\tau_k,\mu_{l+x+1})\cup[\mu_{j+x+1},\mu_{j+x+y+1}+T)\big)\Big]}$$

$$+\frac{\big[y-\beta(y+1)\big]C+\beta(1-\beta)\Big[\sum_{k=0}^{y-1}(\sum_{k=0}^{y-1}(q_{4k+2}+q_{4k+4})\Big]+(1-\beta^2)\sum_{k=0}^{y}q_{4k+1}+(1-\beta)s_{4x+2}}{\beta\cdot\Big[\mathsf{OPT}\big(\sigma;[\tau_k,\mu_{l+x+1})\cup[\mu_{j+x+1},\mu_{j+x+y+1}+T)\big)\Big]}$$

$$\geq \frac{\Big[(x+y)(1-\beta)+(1-2\beta)\Big]C+\beta(1-\beta)s_{4x}+(1-\beta^2)s_{4x+1}+(1-\beta)s_{4x+2}}{\beta\cdot\Big[\mathsf{OPT}\big(\sigma;[\tau_k,\mu_{l+x+1})\cup[\mu_{j+x+1},\mu_{j+x+y+1}+T)\big)\Big]}$$

$$\geq \frac{\Big[(x+y)(1-\beta)+(1-2\beta)\Big]C+\beta(1-\beta)(s_{4x}+s_{4x+1}+s_{4x+2})}{\beta\cdot\Big[\mathsf{OPT}\big(\sigma;[\tau_k,\mu_{l+x+1})\cup[\mu_{j+x+1},\mu_{j+x+y+1}+T)\big)\Big]} \quad \text{(since } 0\leq\beta<1)$$

$$\geq \frac{\Big[(x+y)(1-\beta)+(1-2\beta)\Big]C+\beta(1-\beta)\gamma}{\beta\cdot\Big[\mathsf{OPT}\big(\sigma;[\tau_k,\mu_{l+x+1})\cup[\mu_{j+x+1},\mu_{j+x+y+1}+T)\big)\Big]} \quad \text{(by (97))}$$

$$= \frac{\Big[(x+y)(1-\beta)+(1-\beta)\Big]C}{\beta\cdot\Big[\mathsf{OPT}\big(\sigma;[\tau_k,\mu_{l+x+1})\cup[\mu_{j+x+1},\mu_{j+x+y+1}+T)\big)\Big]}$$

$$> 0. \quad \text{(since } 0\leq\beta<1, C>0)$$

Thus, the following inequality always holds:

$$\frac{\mathsf{PFSUM}\big(\sigma;[\tau_k,\mu_{l+x+1})\cup[\mu_{j+x+1},\mu_{j+x+y+1}+T)\big)}{\mathsf{OPT}\big(\sigma;[\tau_k,\mu_{l+x+1})\cup[\mu_{j+x+1},\mu_{j+x+y+1}+T)\big)} < \frac{1}{\beta} \tag{98}$$

**Case I.** $0\leq\eta\leq\gamma$. By Corollary 3.2, the $T$-future-cost at $\tau_{i+k+1}$ is at least $\gamma$, i.e.,

$$\begin{cases} s_{-1}+s_0\geq\gamma, \\ s_{4k+2}+s_{4k+3}+s_{4k+4}\geq\gamma & \text{for each } k=0,...,x-1, \\ q_{4k+2}+q_{4k+3}+q_{4k+4}\geq\gamma & \text{for each } k=0,...,y-1. \end{cases} \tag{99}$$

By Lemma 4.2, we have

$$\begin{cases} s_{4k}+s_{4k+1}+s_{4k+2}\geq\gamma-\eta & \text{for each } k=0,...,x-1, \\ s_{4x}+s_{4x+1}\geq\gamma-\eta. \\ q_1+q_2\geq\gamma-\eta, \\ q_{4k}+q_{4k+1}+q_{4k+2}\geq\gamma-\eta & \text{for each } k=1,...,y-1, \\ q_{4y}+q_{4y+1}\geq\gamma-\eta. \end{cases} \tag{100}$$

Note that for Pattern IV, all the travel requests in the $(4k+2)$-th and the $(4k+4)$-th (for $k=0,...,x-1$) time intervals are reduced requests of both $\mathsf{PFSUM}$ and $\mathsf{OPT}$, similar for Pattern VI. Thus, to maximize the cost ratio in $[\tau_k,\mu_{l+x}+T)\cup[\mu_{j+x+1},\mu_{j+x+y+1}+T)$, we should minimize these $s_{4k+2}$'s and $s_{4k+4}$'s. If they are greater than $\gamma$, the cost ratio can be increased by decreasing $s_{4k+2}$ or $s_{4k+4}$ to $\gamma$ without violating (99) and (100). Thus, for the purpose of deriving an upper bound on the cost ratio, we can assume that

$$\begin{cases} s_{4k+2}\leq\gamma & \text{for each } k=0,...,x, \\ s_{4k+4}\leq\gamma & \text{for each } k=-1,...,x-1, \\ q_{4k+2}\leq\gamma & \text{for each } k=0,...,y-1, \\ q_{4k+4}\leq\gamma & \text{for each } k=0,...,y-1. \end{cases} \tag{101}$$

It follows from (101) and Lemma 4.4 that,

$$\begin{cases} s_{-1}+s_0\leq 2\gamma+\eta \\ s_{4k+2}+s_{4k+3}+s_{4k+4}\leq 2\gamma+\eta & \text{for each } k=0,...,x-1, \\ q_{4k+2}+q_{4k+3}+q_{4k+4}\leq 2\gamma+\eta & \text{for each } k=0,...,y-1 \end{cases} \tag{102}$$

As a result, we have

$$\frac{\mathsf{PFSUM}\big(\sigma; [\tau_k, \mu_{l+x+1}) \cup [\mu_{j+x+1}, \mu_{j+x+y+1} + T)\big)}{\mathsf{OPT}\big(\sigma; [\tau_k, \mu_{l+x+1}) \cup [\mu_{j+x+1}, \mu_{j+x+y+1} + T)\big)}$$

$$= \frac{(x+1)C + (s_{-1}+s_0) + \sum_{k=0}^{x-1}\theta_k + \beta\sum_{k=0}^{x}s_{4k+1} + s_{4x+2} - (1-\beta)\Big[\sum_{k=0}^{x-1}\big(s_{4k}+s_{4k+2}\big) + s_{4x}\Big]}{(x+1)C + \beta(s_{-1}+s_0+\sum_{k=0}^{x-1}\theta_k) + \sum_{k=0}^{x}s_{4k+1} + s_{4x+2} + yC + \beta\sum_{k=0}^{y-1}\delta_k + \sum_{k=0}^{y}q_{4k+1}}$$

$$+ \frac{(y+1)C + \sum_{k=0}^{y}q_{4k+1} + \sum_{k=0}^{y-1}\delta_k - (1-\beta)\Big[\sum_{k=0}^{y-1}\big(q_{4k+1}+q_{4k+2}+q_{4k+4}\big) + q_{4y+1}\Big]}{(x+1)C + \beta(s_{-1}+s_0+\sum_{k=0}^{x-1}\theta_k) + \sum_{k=0}^{x}s_{4k+1} + yC + \beta\sum_{k=0}^{y-1}\delta_k + \sum_{k=0}^{y}q_{4k+1}}$$

$$\leq \frac{(x+1)C + (s_{-1}+s_0) + \sum_{k=0}^{x-1}\theta_k + \beta\sum_{k=0}^{x}s_{4k+1} - (1-\beta)\Big[\sum_{k=0}^{x-1}\big(s_{4k}+s_{4k+2}\big) + s_{4x}\Big]}{(x+1)C + \beta(s_{-1}+s_0+\sum_{k=0}^{x-1}\theta_k) + \sum_{k=0}^{x}s_{4k+1} + yC + \beta\sum_{k=0}^{y-1}\delta_k + \sum_{k=0}^{y}q_{4k+1}}$$

$$+ \frac{(y+1)C + \sum_{k=0}^{y}q_{4k+1} + \sum_{k=0}^{y-1}\delta_k - (1-\beta)\Big[\sum_{k=0}^{y-1}\big(q_{4k+1}+q_{4k+2}+q_{4k+4}\big) + q_{4y+1}\Big]}{(x+1)C + \beta(s_{-1}+s_0+\sum_{k=0}^{x-1}\theta_k) + \sum_{k=0}^{x}s_{4k+1} + yC + \beta\sum_{k=0}^{y-1}\delta_k + \sum_{k=0}^{y}q_{4k+1}}$$

$$\text{(since } s_{4x+2} \geq 0\text{)}$$

$$\leq \frac{(x+1)C + (x+1)(2\gamma+\eta) + \sum_{k=0}^{x}s_{4k+1} - (1-\beta)\Big[\sum_{k=0}^{x-1}\big(s_{4k}+s_{4k+1}+s_{4k+2}\big) + (s_{4x}+s_{4x+1})\Big]}{(x+1)C + \beta(x+1)(2\gamma+\eta) + \sum_{k=0}^{x}s_{4k+1} + yC + \beta y(2\gamma+\eta) + \sum_{k=0}^{y}q_{4k+1}}$$

$$+ \frac{(y+1)C + \sum_{k=0}^{y}q_{4k+1} + y(2\gamma+\eta) - (1-\beta)\Big[q_1 + q_2 + \sum_{k=1}^{y-1}\big(q_{4k}+q_{4k+1}+q_{4k+2}\big) + q_{4y} + q_{4y+1}\Big]}{(x+1)C + \beta(x+1)(2\gamma+\eta) + \sum_{k=0}^{x}s_{4k+1} + yC + \beta y(2\gamma+\eta) + \sum_{k=0}^{y}q_{4k+1}}$$

$$\text{(by (98) and (102))}$$

$$\leq \frac{(x+1)C + (x+1)(2\gamma+\eta) + \sum_{k=0}^{x}s_{4k+1} - (1-\beta)\Big[x(\gamma-\eta)+\gamma\Big]}{(x+1)C + \beta(x+1)(2\gamma+\eta) + \sum_{k=0}^{x}s_{4k+1} + yC + \beta y(2\gamma+\eta) + \sum_{k=0}^{y}q_{4k+1}}$$

$$+ \frac{(y+1)C + \sum_{k=0}^{y}q_{4k+1} + y(2\gamma+\eta) - (1-\beta)(y+1)(\gamma-\eta)}{(x+1)C + \beta(x+1)(2\gamma+\eta) + \sum_{k=0}^{x}s_{4k+1} + yC + \beta y(2\gamma+\eta) + \sum_{k=0}^{y}q_{4k+1}}$$

$$\text{(by (97) and (100))}$$

$$\leq \frac{(x+y+2)C + (x+y+1)(2\gamma+\eta) - (1-\beta)\Big[(x+y+2)\gamma - (x+y+1)\eta\Big]}{(x+y+1)C + \beta(x+y+1)(2\gamma+\eta)}$$

$$\left(\text{since } \sum_{k=0}^{x}s_{4k+1} \geq 0, \sum_{k=0}^{y}q_{4k+1} \geq 0\right)$$

$$= \frac{2(x+y+1)\gamma + (x+y+1)(2-\beta)\eta}{(x+y+1)\Big[(1+\beta)\gamma + \beta\eta\Big]}$$

$$= \frac{2\gamma + (2-\beta)\eta}{(1+\beta)\gamma + \beta\eta}. \tag{103}$$

**Case II.** $\eta > \gamma$. By Lemma 4.3, for Pattern IV, the total regular cost in the $(4k+3)$-th (for $k = -1, ..., x$) time interval (which is in an off phase and has length at most $T$) is less than $2\gamma + \eta$, similar for Pattern VI. Thus we have

$$\begin{cases} s_{4k+3} < 2\gamma + \eta & \text{for each } k = -1, ..., x-1, \\ q_{4k+3} < 2\gamma + \eta & \text{for each } k = 0, ..., y-1. \end{cases} \tag{104}$$

On the other hand, the total regular cost in any time interval in an on phase is non-negative, so it can be minimized to zero, except the last on phase of Pattern IV. As a result, we have

$$\frac{\mathsf{PFSUM}\big(\sigma; [\tau_k, \mu_{l+x+1}) \cup [\mu_{j+x+1}, \mu_{j+x+y+1} + T)\big)}{\mathsf{OPT}\big(\sigma; [\tau_k, \mu_{l+x+1}) \cup [\mu_{j+x+1}, \mu_{j+x+y+1} + T)\big)}$$

$$\leq \frac{(x+1)C + \sum_{k=-1}^{x-2} s_{4k+3} + s_{4x-1} + \beta s_{4x} + (y+1)C + \sum_{k=0}^{y-1} q_{4k+3}}{(x+1)C + \beta(s_{-1} + \sum_{k=0}^{x-2} s_{4k+3} + s_{4x-1} + s_{4x}) + yC + \beta \sum_{k=0}^{y-1} q_{4k+3}}$$

$$\leq \frac{(x+y+2)C + (x+y)(2\gamma+\eta) + \gamma + \eta + \beta\gamma}{(x+y+1)C + \beta(x+y+1)(2\gamma+\eta)} \quad \text{(by (97) and (104))}$$

$$= \frac{(x+y+2)C + (x+y+1)(2\gamma+\eta) + (\beta-1)\gamma}{(x+y+1)C + \beta(x+y+1)(2\gamma+\eta)}$$

$$= \frac{(x+y+1)C + (x+y+1)(2\gamma+\eta)}{(x+y+1)C + \beta(x+y+1)(2\gamma+\eta)}$$

$$\leq \frac{(3-\beta)\gamma + \eta}{(1+\beta)\gamma + \beta\eta}. \tag{105}$$

The result follows from (103), and (105).

## A.8 More Experimental Results and Discussions

For commuters, Figures 25 to 30 demonstrate the performance of various algorithms under different settings of $\beta$, $C$, and $T$. Additionally, with the practical significance of these parameters in mind, we increase $C$ or decrease $T$ when $\beta$ is reduced. In Figure 26, where $C$ is exceptionally high, PFSUM, $\mathsf{SUM}_w$, and SUM all tend towards not purchasing Bahncards. However, in this scenario, FSUM's competitive ratio escalates due to its complete reliance on predictions with large errors.

Additional results for occasional travelers are presented in Figures 28 to 30. These figures exhibit a consistent pattern, with PFSUM demonstrating its consistency in scenarios of small prediction error. Furthermore, PFSUM proves its robustness in cases of extremely large prediction errors, particularly evident in Figure 30. Here, while all other prediction-incorporated algorithms have competitive ratios of approximately 1.2 or higher, PFSUM consistently maintains a competitive ratio of less than 1.1.

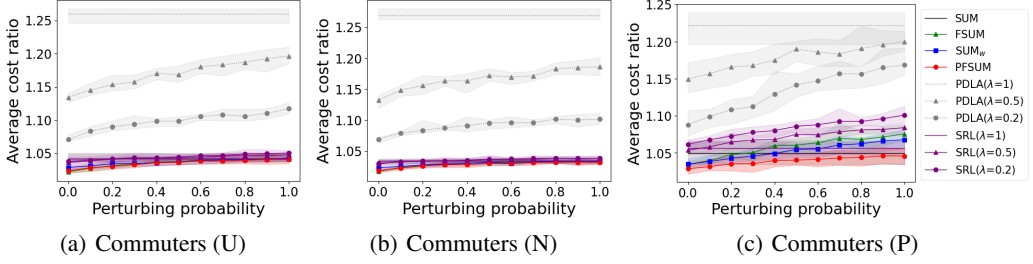

(a) Commuters (U)          (b) Commuters (N)          (c) Commuters (P)

Figure 25: The cost ratios for commuters ($\beta = 0.6$, $T = 10$, $C = 200$). "U", "N" and "P" represents Uniform, Normal and Pareto ticket price distributions respectively.

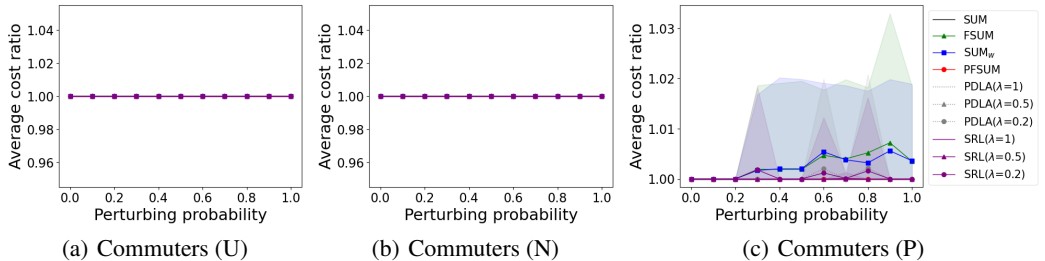

Figure 26: The cost ratios for commuters ($\beta = 0.6$, $T = 10$, $C = 2000$). "U", "N" and "P" represents Uniform, Normal and Pareto ticket price distributions respectively.

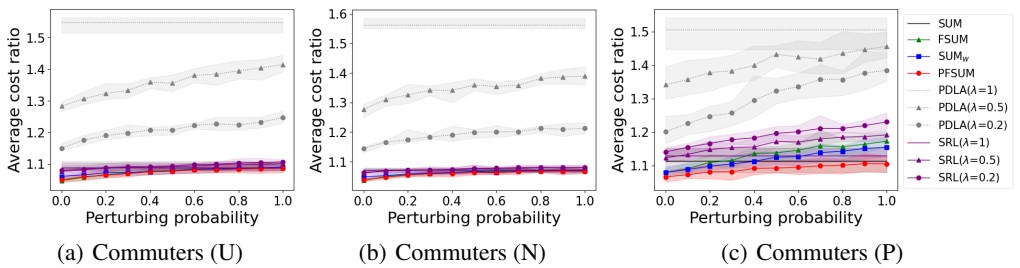

Figure 27: The cost ratios for commuters ($\beta = 0.2$, $T = 10$, $C = 400$). "U", "N" and "P" represents Uniform, Normal and Pareto ticket price distributions respectively.

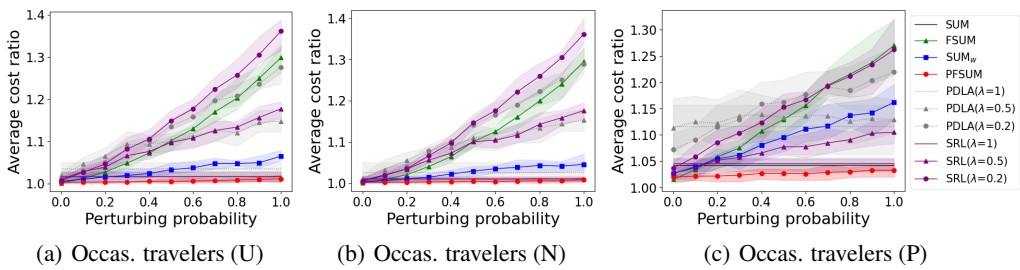

Figure 28: The cost ratios for occasional travelers ($\beta = 0.6$, $T = 10$, $C = 200$). "U", "N" and "P" represents Uniform, Normal and Pareto ticket price distributions respectively.

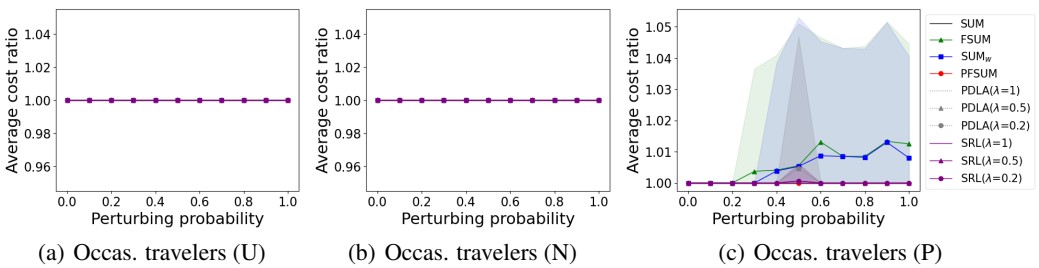

Figure 29: The cost ratios for occasional travelers ($\beta = 0.6$, $T = 10$, $C = 2000$). "U", "N" and "P" represents Uniform, Normal and Pareto ticket price distributions respectively.

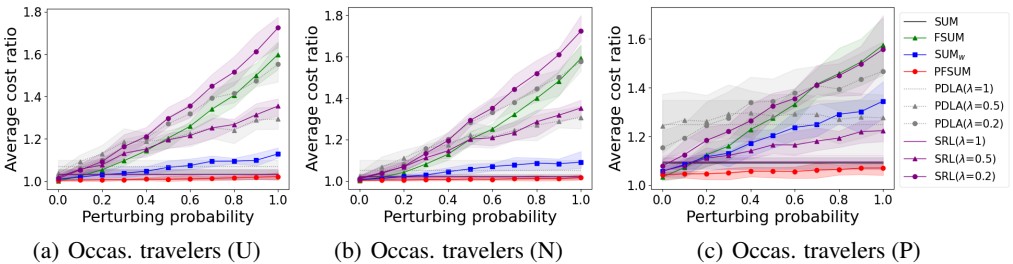

(a) Occas. travelers (U)   (b) Occas. travelers (N)   (c) Occas. travelers (P)

Figure 30: The cost ratios for occasional travelers ($\beta = 0.2$, $T = 10$, $C = 400$). "U", "N" and "P" represents Uniform, Normal and Pareto ticket price distributions respectively.

