# OpenReview forum: "Learning-Augmented Algorithms for the Bahncard Problem"
_NeurIPS.cc/2024/Conference — NeurIPS 2024 poster_

### Official Review · Reviewer_ZK3p · 2024-07-07

**Soundness:** 2
**Presentation:** 3
**Contribution:** 2
**Rating:** 3
**Confidence:** 5

**Summary:**

This work studies the Bahncard Problem and proposes a new learning-augmented algorithm, PFSUM. The writing of the paper is clear and easy to follow. The work also provides detailed mathematical proofs for six patterns followed by the experiments validate the theoretical findings.

**Strengths:**

This paper presents several interesting elements, particularly in its approach to a more generalized version of the Bahncard Problem. The consistency of the proposed algorithm, with its bounded promises, is a noteworthy feature. Unlike traditional approaches, this algorithm does not require predicting the entire future input, which simplifies the decision-making process. Additionally, the paper provides a detailed analysis of the cost-effectiveness across six different travel patterns, offering valuable insights into the performance of the algorithm.

**Weaknesses:**

The contributions of this paper are relatively narrow and ambiguous. One significant issue is that the robustness of the PFSUM algorithm remains unbounded when β converges to 0. This limitation undermines the reliability of the algorithm in scenarios where β is very small. Additionally, the parameter T is critical as it defines the boundaries of both the time interval and the prediction interval. However, the paper does not provide a clear analysis of how variations in T impact the competitive ratio, leaving an essential aspect of the algorithm's performance unexplored.

The related work section fails to engage with recent literature that could provide context for this study's contributions. While [12] is a milestone work from 2020, recent publications exploring different facets of the same problem are omitted. These could include the works like "Online algorithms with costly predictions (2023)", "Learning-Augmented Online Algorithm for Two-Level Ski-Rental Problem (2024)" and others. Including these works would offer a more comprehensive view of the field's current state. The authors could also indicate the principle and mechanism differences, rather than contextual disparities, between their work and [25].

The experimental validation is insufficient, relying on only two benchmarks, which inadequately demonstrates the generalizability and robustness of the proposed algorithm. While the paper claims to measure online algorithm performance using competitive ratios, the experiments primarily employ cost ratios. This discrepancy between theoretical claims and experimental measures undermines the impact of the findings and may lead to misinterpretation of the PFSUM's actual performance improvements.

**Questions:**

1. What are the specific contributions of your work compared to other studies on the same problem, including those not cited in this paper?

2. Have you explored the effects of varying the length of the prediction interval on the performance of PFSUM? Could you provide insights into the trade-off between consistency and robustness in your algorithm?

3. Could you explain why your experimental validation relies on only two benchmarks? What are the reasons behind this choice, and do you believe that including more performance benchmarks could address potential limitations in demonstrating the algorithm's effectiveness?

**Limitations:**

There is no negative societal impact of this work.

---

> ### Author Rebuttal · Authors · 2024-08-07
>
> ### **Weakness 1 & Question 1**
> Our specific contributions are to (1) develop an effective learning-augmented algorithm PFSUM for the Bahncard problem **using predictions on an immediate future only, which is more practical since the predictor is much easier to train**; (2) analyze the competitive ratio of PFSUM **as a function of prediction error** by a divide-and-conquer approach and a holistic analysis of card purchasing patterns and their concatenations; (3) experimentally evaluate PFSUM and compare it with state-of-the-art algorithms in the literature. Moreover, we would like to point out that we focus on a **continuous** time setting of the Bahncard problem where requests can arise and cards can be purchased at any time instant along the time dimension. This is motivated by applications such as cloud instance reservation (in clouds, reserved VM instances can be purchased at any time instant). Our setting is more general than the slotted time setting (where requests are made only at the beginning of discrete time slots and so do card purchases) studied by prior work including reference [12] and *Online algorithms with costly predictions (2023)* you mentioned. **The unbounded robustness $1/\beta$ of PFSUM when $\beta$ goes to 0 arises from the continuous time setting.** The worst-case example is that *there are an infinite number of small-cost travel requests made continuously over time* and *predictions always forecast low total cost in the future interval* (so that the total ticket price is infinite while PFSUM does not purchase cards due to bad predictions). This example does not apply to the slotted time setting because the number of requests in an interval is limited (if the total ticket price goes to infinity, the price of individual requests also goes to infinity, and for any request with price exceeding $\gamma$ at time $t$, PFSUM purchases a card at $t$ by definition in lines 175-176). **In the continuous time setting, $T$ does not affect the competitive ratio as changing $T$ just implies scaling time up or down.** This is akin to the optimal conventional Bahncard algorithm SUM having a competitive ratio $2-\beta$ independent of $T$.
>
> ### **Weakness 2**
> Thank you for pointing us to these papers. We will include them in the related work. *Learning-Augmented Online Algorithm for Two-Level Ski-Rental Problem (2024)* studies an extended version of ski-rental problem, which extends ski-rental **along the item dimension** (from one item to multiple items and introduces a new option of combo purchase), whereas the Bahncard problem extends ski-rental **along the time dimension** (making decisions repeatedly). They have different problem structures. *Online algorithms with costly predictions (2023)* focuses on how many predictions are required to gather enough information to output a near-optimal buying schedule, **assuming all predictions are correct**. The learning-augmented algorithm proposed **takes a suggested sequence of buying times as input** and **assumes a slotted time setting**, thereby sharing similar drawbacks to reference [12]. The paper studies only the consistency and robustness of the proposed algorithm, and **does not** derive the competitive ratio as a function of prediction error. Without the latter, one could not judge how tolerant the algorithm is to errors. If there is no ratio function of the error, the ratio of the algorithm will jump directly from consistent to robust if the prediction is slightly wrong (which usually happens in practice). Moreover, there was no experimental evaluation.
>
> ### **Weakness 3 & Question 3**
> Thank you for your comments and questions. In the experiments, cost ratio refers to the ratio between the cost produced by an online algorithm and the offline optimal cost **for a given input** (i.e., travel requests). In the theoretical analysis, competitive ratio refers to the **worst-case** cost ratio across all possible inputs. So, they are consistent. As with most papers in this field, **the experiments tend to show better performance than the theoretically analyzed bounds due to the use of particular datasets**.
>
> Robustness is an upper bound of the competitive ratio across all possible prediction errors, which means the worst-case scenario typically arising from extreme inputs. Our experiments generate travel requests from distributions that resemble real-world scenarios, which are unlikely to encounter such extreme situations. Hence, it is not surprising that the empirical results do not demonstrate the theoretical bound of robustness. **The main purpose of the experiments is to compare the empirical cost ratios of different algorithms with realistic travel request patterns, rather than demonstrating theoretical worst-case bounds.**
>
> For experimental benchmarks, we have tried to find **all** papers related to the Bahncard problem. The **only** article that conducts experiments for the Bahncard problem is reference [28]. We therefore **refer to their experimental setup** of traveler profiles and include various types of ticket price distributions in our paper. Reference [12] did not conduct experiments for the Bahncard problem. We implement its algorithm (PDLA), and include it in our comparison. Thus, our experiments provide comprehensive empirical evaluation and comparison.
>
> ### **Question 2**
> We appreciate your insight. Section 4.1 of our paper studies a prediction interval shorter than $T$ and shows that it is not good. If the length of the prediction interval is $T + \epsilon$ (for any $\epsilon > 0$), it means at time $t$, we know the predicted total regular cost in $[t, t + T + \epsilon)$. But it may happen that all the cost in $[t, t + T + \epsilon)$ is incurred before $t+ T$ (i.e., in $[t, t + T)$) or all the cost is incurred beyond $t+T$ (i.e., in $[t + T, t+T+\epsilon)$). **Thus, it is not helpful to make the correct purchase decision at time $t$, because a Bahncard purchased at $t$ can reduce the cost in $[t, t + T)$ only.**

---

> ### Comment · Reviewer_ZK3p · 2024-08-13
> **My Concerns Remain**
>
> Thank you for the detailed responses from the authors. I thought through your work and your responses. Although I like your work and enjoy reading the text, a couple of unsolved issues hesitate me to improve my score.
>
> 1) Since your algorithm predicts the near future, the advice complexity could be much larger than those algorithms predicting a long-term future. It will be interesting to explore the cumulative average competitive ratio versus the competitive ratio of a long-term prediction to prove whether the proposed online algorithm is more practical.
>
> 2) Relating the prediction errors to the competitive ratio is interesting but is not new, which can date back to the paper “Improving Online Algorithms via ML Predictions” (2018). The better bounded competitive ratio seems to be more attractive and convincing. Furthermore, if we let β = 0, we can observe that the consistency of PFSUM converges to at least 2. As is known, in the ski rental problem, the worst-case competitive ratio is 2. It means when β = 0, the consistency of PFSUM is worse than the robustness of other online algorithms.
>
> 3) If T is related to the prediction errors, will it affect the competitive ratio?
>
> 4) As mentioned, you only found one work that conducted experiments on the Bahncard problem, you can still employ other online algorithms on the Bahncard problem, as you do with your reference [12]. A single benchmark cannot show your proposed contributions, particularly in ML/AI conferences. Those works mostly conducted extensive experiments on numerous datasets compared across multiple benchmarks. With the current shape of the paper and your arguments, I just feel this problem as the Achilles’s heel of the work. If the theoretical and experimental results of the Bahncard problem are still limited, you can put your algorithm into a ski rental problem with β = 0 and T converging to infinity. Then, you can have more benchmarks even they may not be the perfect one. This issue makes me wonder whether this venue is the right place for this work. In my opinion, the theoretical computing conferences, e.g., STOC, FOCS, SODA, may be a better fit for this work as they pay more attention on theoretical contributions and will have more audiences to read and cite the work.

---

> > ### Author Response · Authors · 2024-08-14
> > **(1/2) Official Comment by Authors**
> >
> > Thank you for your feedback. Below are our responses to your concerns.
> >
> > ### **Question 1**
> >
> > PFSUM predicts the near future only when it encounters a regular travel request, regardless of how long the time horizon is. However, when the time horizon is sufficiently long, time series predictions (those predicting a long-term future) are **intractable** in the real world (i.e., their prediction complexity is $\infty$). Therefore, it is **inaccurate** to assert that our prediction complexity is much larger.
> >
> > For practical verification, our experimental evaluation actually includes the type of effectiveness comparison you mentioned, specifically comparing our proposed algorithm, PFSUM, which predicts the near future, with the PDLA algorithm from reference [12], which predicts the long-term future. The results show that **PFSUM consistently and significantly outperforms PDLA**.
> >
> > Moreover, a once-for-all long-term prediction made at the very beginning struggles to adapt to travel requests whose patterns change dynamically, making it impractical. In contrast, our short-term prediction can overcome this limitation and adapt to such dynamics online.
> >
> > ### **Question 2**
> >
> > The consistency of PFSUM is **better than** the robustness of other online algorithms even when $\beta = 0$. If $\beta = 0$, the consistency of PFSUM is $\frac{2}{1+\beta} = 2$. For ski-rental, the deterministic algorithm proposed in *Improving Online Algorithms via ML Predictions (NeurIPS '18)* has a robustness of $1 + \frac{1}{\lambda}$ (where $0 < \lambda < 1$ is a hyperparameter), which is greater than 2. For Bahncard, the algorithm proposed in *Online Algorithms with Costly Predictions (2023)* also has a robustness of $1 + \frac{1}{\lambda}$ (where $0 < \lambda < 1$), which is greater than 2.
> >
> > ### **Question 3**
> >
> > We have derived the competitive ratio as a function of the prediction error. If the prediction error is related to $T$, then the effect of $T$ is implicitly included in our result. In the Bahncard problem, $T$ is a given parameter, not something created in any proposed algorithm that can be tuned. Studying the relation between the prediction error and $T$, if any, is orthogonal to our work.

---

> > ### Author Response · Authors · 2024-08-14
> > **(2/2) Official Comment by Authors**
> >
> > ### **Question 4**
> >
> > Reference [12] (published in NeurIPS '20) studied several problems, including the Bahncard problem, but experimented only with the TCP acknowledgment problem using three distributions of packet arrivals. *Online Algorithms with Costly Predictions (2023)* was published in AISTATS, but it did not conduct any experimental evaluation. Thus, we disagree that our experimental evaluation is below the expectations of ML/AI conferences. **In fact, to our knowledge, we are the first to conduct extensive experiments for learning-augmented Bahncard algorithms.** Our experimental evaluation covers two types of traveler profiles, each with three types of ticket price distributions, totaling six benchmarks.
> >
> > Furthermore, it doesn't make sense to evaluate our algorithm with datasets (or benchmarks) for ski rental, since in the Bahncard problem, buying decisions need to be made repeatedly over time, which is the most prominent difference from ski rental and this difference guides the algorithm design. Based on your suggestions, however, we have employed another algorithm for comparison. We adapted Algorithm 2 proposed for ski rental in *Improving Online Algorithms via ML Predictions (NeurIPS '18)* to the Bahncard problem and named it SRL.
> >
> > Let $\lambda \in (0, 1)$ be a hyperparameter, SRL purchases a Bahncard at a regular travel request $(t,p)$ if and only if there exists a time instant $t' \in [t - T, t]$ that satisfies one of the following conditions:
> >
> > - The predicted $T$-future-cost at time $t'$ is $\geq \gamma$, and the sum of the costs over the time interval $[t', t]$ is greater than $\lambda \gamma$.
> > - The predicted $T$-future-cost at time $t'$ is $< \gamma$, and the sum of the costs over $[t', t]$ is $> \frac{\gamma}{\lambda}$.
> >
> > We found that PFSUM also consistently outperforms SRL in the experiments, as shown below.
> >
> > - The following table presents the additional results for *Commuters* when $\beta = 0.8$, $T = 10$, $C = 100$, and the price distribution is Pareto.
> >
> >     | average cost ratio/perturbing probability | 0.0 | 0.1 | 0.2 | 0.3 | 0.4 | 0.5 | 0.6 | 0.7 | 0.8 | 0.9 | 1.0 |
> >     |---|---|---|---|---|---|---|---|---|---|---|---|
> >     | SRL ($\lambda = 1$) | 1.027 | 1.027 | 1.027 | 1.027 | 1.027 | 1.027 | 1.027 | 1.027 | 1.027 | 1.027 | 1.027 |
> >     | SRL ($\lambda = 0.6$) | 1.025 | 1.036 | 1.042 | 1.045 | 1.046 | 1.047 | 1.047 | 1.048 | 1.048 | 1.048 | 1.048 |
> >     | SRL ($\lambda = 0.4$) | 1.026 | 1.043 | 1.052 | 1.056 | 1.058 | 1.059 | 1.060 | 1.060 | 1.060 | 1.060 | 1.060 |
> >     | PFSUM | **1.014** | **1.021** | **1.024** | **1.025** | **1.027** | **1.027** | **1.027** | **1.027** | **1.027** | **1.027** | **1.027** |
> >
> > - The following table presents the additional results for *Occasional Travelers* when $\beta = 0.8$, $T = 10$, $C = 100$, and the price distribution is Pareto.
> >
> >     | average cost ratio/perturbing probability | 0.0 | 0.1 | 0.2 | 0.3 | 0.4 | 0.5 | 0.6 | 0.7 | 0.8 | 0.9 | 1.0 |
> >     |---|---|---|---|---|---|---|---|---|---|---|---|
> >     | SRL ($\lambda = 1$)| 1.023 | 1.023 | 1.023 | 1.023 | 1.023 | 1.023 | 1.023 | 1.023 | 1.023 | 1.023 | 1.023 |
> >     | SRL ($\lambda = 0.6$)| 1.015 | 1.04 | 1.046 | 1.052 | 1.054 | 1.055 | 1.055 | 1.055 | 1.055 | 1.055 | 1.055 |
> >     | SRL ($\lambda = 0.4$) | 1.013 | 1.07 | 1.088 | 1.104 | 1.110 | 1.114 | 1.115 | 1.115 | 1.115 | 1.115 | 1.115 |
> >     | PFSUM | **1.010** | **1.017** | **1.022** | **1.022** | **1.010** | **1.017** | **1.022** | **1.022** | **1.010** | **1.017** | **1.022** |
> >
> > ### Concluding Remarks
> >
> > We are not claiming that our work is perfect. But our work does address learning-augmented Bahncard from perspectives different than existing work (including reference [12] and *Online algorithms with costly predictions (2023)*). We also conduct a comprehensive experimental evaluation to compare with existing algorithms. We believe that our work is valuable to the community and can further stimulate new ideas & studies of the learning-augmented Bahncard problem.

---

### Official Review · Reviewer_AGsA · 2024-07-10

**Soundness:** 3
**Presentation:** 4
**Contribution:** 3
**Rating:** 7
**Confidence:** 4

**Summary:**

In this paper, the authors investigate the Bahncard problem in the learning-augmented context. The Bahncard problem is a generalization of ski rental, originating from the railway pass of the German railway company of the same name, where an algorithm must choose between a cheap short-term solution (purchaing railway fare at full price) or an expensive long-term solution (purchasing a Bahncard to receive a discount on all future fares for a set period). Previous works have studied the Bahncard problem extensively in the non-learning-augmented setting, and in the learning-augmented setting by formulating the Bahncard problem as a linear program and applying the primal-dual framework for approximately solving LPs. The authors contribute by proposing an algorithm using the problem structure of the Bahncard problem, independent of the primal-dual framework, that can handle fractional and non-uniform inputs, and utilizes short-term predictions as opposed to an advice on the entire sequence of travels.

Formally, the authors consider problem instances where a series of travel requests arrive at time $t_i$ with ticket price $p_i$. An algorithm can either choose to buy the ticket in full price, or purchase a Bahncard with price $C$, that is valid for a time period of length $T$, and discounts all purchase while valid by a multiplicative factor of $0 \leq \beta \leq 1$. The author's proposed algorithm, PFSUM, is $2/(1+\beta)$-consistent (the competitive ratio when the prediction is accurate) and $1/\beta$-robust (the competitive ratio upper bound with arbitrarily inaccurate prediction).

PFSUM is a very intuitively simple algorithm, that builds upon previous ideas. On a high level, the algorithm examines both the requests that arrived in the past $T$ time period, as well as the predicted requests that will arrive in the next $T$ time period. If the total cost of both the past and the future time period is larger than a certain parameter, the algorithm purchases a Bahncard. The proof strategy divides the algorithm's execution into phases, and bounds the ratio between PFSUM's solution cost and the optimal solution cost for each phase, depending on how the Bahncard purchases overlap between the two solutions.

**Strengths:**

Overall the paper is very well-written and addresses a theoretically and practically interesting problem in the context of learning-augmented algorithms, improving on previous solutions. The results yield significant improvements over prior works on the same problem, by specializing and utilizing the problem structure, and the model (advice, performance) chosen by the author are natural and reasonably practical. The algorithms and high-level ideas presented are simple and intuitive, and while the detailed proofs are quite involved, they are presented in a fashion that is easy to follow and understand, which I believe is an important characteristic of a good paper, especially for a widely attended conference such as NeurIPS.

In particular, I appreciate the author's explanation of various initial attempts based on prior works in Section 4: I can easily follow the development of their algorithm from simple preliminary ideas inspired by existing literature, and understand why these naive algorithms does not provide satisfactory bounds, and what led the authors to make improvements and modifications to derive the eventual PFSUM algorithm. Apart from this, the theorems and technical claims are also introduced in sufficient context for me to understand what role does each claim play in the overall structure of proofs and logic. The various figures used by the authors to illustrate definitions and ideas are also much appreciated.

**Weaknesses:**

I do not have much major complaints about the paper. One particular issue is that I am not certain how the conclusion of Proposition 4.6 on page 6 is trivial: while I agree the numerator is upper bounded by Lemma 4.3, I do not see an obvious lower bound on the cost of the optimal solution in the denominator, for which Lemma 4.3 only implies an upper bound. A quick delve into the appendices seems to provide traces of arguments supporting $OPT(*) \geq (1+\beta) \gamma + \beta \eta$, so I am inclined to trust the soundness of the author's claims, but I believe that this is important enough an argument to be made explicit in the main corpus.

I am recommending a weak accept as an initial score, but can be convinced by clarifications from the authors to raise the score to accept.

**Questions:**

As stated above, I would like to understand the logic behind Proposition 4.6 in more detail, which I believe is important yet non-trivial enough to be explicitly shown in the main corpus.

A few minor questions/suggestions/comments:
- Line 169-172, Section 4.3: The argument behind SUM's success might be more suitable in an earlier section, to help the readers understand both Section 4.1 and 4.2 better.
- Line 188: The definition of the prediction error only concerns the sum of the cost of requests. Is it possible to obtain better bounds and results using a more "fine-grained" error function? This might be an interesting future direction if feasible.
- Line 203, Lemma 4.4: The statement and the figure together slightly confuses me - what if the time interval $[t, t+T)$ does not intersect on phases on both sides? Intuitively the same result should hold with $s_2 = s_4 = 0$, but it is not made explicit by either the lemma statement or the figure.
- Line 221-222, text of Figure 2: "...$x$ can be any non-negative **integers**." It should be **integer**?
- Line 252, Proposition 4.8: The definitions seem to suggest that there should be no regular requests during an on phase, so what is the logic behind the proposition statement? Are the regular cost and the on phase referencing different algorithms?

**Limitations:**

The authors discuss their limitations properly in the introduction and the conclusion sections. No ethical concerns are applicable.

---

> ### Author Rebuttal · Authors · 2024-08-07
>
> Dear Reviewer AGsA, we deeply appreciate your acknowledgment and support of our work! Below, we respond to the weakness and questions 1-5, one by one.
>
> ### **Weakness**
> We greatly appreciate your constructive comments. In particular, we greatly appreciate that you understood our paper in depth. We will rewrite the proof of Proposition 4.6 and hope the new version makes this proposition clearer for readers. The new proof is presented below.
>
> *Proof.* Let $x = c(\sigma; [\tau_i, \tau_i + T))$. Based on the definition of Pattern II,  $\textsf{OPT}(\sigma; [\tau_i, \tau_i + T)) = C + \beta x$ (OPT buys a card at $\tau_i$) and $\textsf{PFSUM}(\sigma; [\tau_i, \tau_i + T)) = x$ (by definition, PFSUM does not buy cards during any off phase). Hence, the cost ratio is $\frac{x}{C + \beta x}$, which increases with $x$ since $\beta < 1$. By Lemma 4.3, $x < 2\gamma + \eta$. Hence, the cost ratio is bounded by $\frac{2\gamma + \eta}{C + \beta (2\gamma + \eta)} = \frac{2\gamma + \eta}{(1-\beta)\gamma + \beta (2\gamma + \eta)} = \frac{2\gamma + \eta}{(1+\beta)\gamma + \beta \eta}$.
>
> ### **Question 1**
> Thank you for your extensive review. We will follow your suggestion to add more description to Section 4.1 to help readers to understand.
>
> ### **Question 2**
> We agree with you. Making more fine-grained predictions for future requests and defining the error function accordingly might help to improve the bounds, which is indeed an interesting direction for future research. Thanks for your valuable suggestion.
>
> ### **Question 3**
> Your insight is reasonable. Actually, Lemma 4.3 describes the case you mentioned, in which the time interval $[t, t+T)$ is fully contained in an off phase, and does not intersect on phases on either sides. Therefore, we omit this case in the figure associated with Lemma 4.4.
>
> ### **Question 4**
> Thanks for pointing out this. You are right, we made a typo here. $x$ should be an integer.
>
> ### **Question 5**
> Thanks for alerting us to this ambiguous description. Firstly, yes, there will be no regular requests during an on phase. An on phase refers to a period during which PFSUM has a valid Bahncard. On the other hand, *the total regular cost* in an on phase refers to **the sum of the original ticket prices** before discounts in this phase (please refer to line 108). Therefore, the regular cost of a given interval is fixed and independent of the algorithm.

---

> > ### Comment · Reviewer_AGsA · 2024-08-08
> >
> > Thank you for your response. Yes, please include the proof (sketch) for Proposition 4.6 in the camera-ready version, if accepted. Personally, I am quite excited to see if there is potential in a more 'fine-grained' advice format, but obviously it is out of scope for this submission.
> >
> > With Proposition 4.6 clarified I am raising my rating from 6 to 7.

---

> > > ### Author Response · Authors · 2024-08-08
> > > **Official Comment by Authors**
> > >
> > > We are very pleased to know that your concerns have been resolved. We also greatly appreciate your high regard for our work's contribution. We will include these discussions in the final paper and try our best to proofread all the details. Once again, we thank you for your constructive suggestions!

---

### Official Review · Reviewer_m14s · 2024-07-12

**Soundness:** 3
**Presentation:** 3
**Contribution:** 3
**Rating:** 6
**Confidence:** 2

**Summary:**

In this paper, the authors provide a learning-augmented approach for solving the Bahncard problem, which is a generalization of the ski-rental problem. The authors provide theoretical guarantees for the consistency and robustness of the proposed algorithm PFSUM, which measures the performance of the proposed method compared to an optimal offline method for different levels of prediction error in learning augmentation. The paper also provides empirical results of the proposed method and a comparison with existing methods, which validate the proposed method.

**Strengths:**

* The paper proposes learning augmentation for the Bahncard problem with theoretical guarantees, which seems novel in the related literature.
* The proposed method is introduced in a methodical way which helps understand the motivation and intuition behind the method.
* The paper provides convincing theoretical and empirical results that validate the proposed method.

**Weaknesses:**

* There seems to be a gap between the theoretical result for robustness and the empirical results in the paper. Specifically, when $\beta$ is very small (i.e. when the discount is larger), the bound for competitive ratio tends to be very large. However, the empirical results show that the cost ratio is close to 1 ( as shown in Figure 29, for example). Is there an intuitive reason for this gap?

**Questions:**

Please see the Weaknesses.

---

> ### Author Rebuttal · Authors · 2024-08-07
>
> Dear Reviewer m14s, we are immensely grateful for your positive comments! Below is our response to the weakness.
>
> ### **Weakness**
> We greatly appreciate your keen observation between the experimental results and the theoretical robustness bound $1 / \beta$. Allow us to start by revisiting the following three definitions, although you might already fully understand them.
> * **Cost ratio:** the ratio between the cost produced by an online algorithm and the cost of the offline optimum for a given input (i.e., travel requests).
> * **Competitive ratio:** the **worst-case** cost ratio across **all possible** inputs, comparing an online algorithm with the offline optimum.
> * **Robustness:** an upper bound of the competitive ratio across **all possible** prediction errors.
> **Therefore, the theoretical bound $1 / \beta$ describes the algorithm’s performance in the worst-case scenario, typically arising from very extreme inputs, under the prediction error being $\infty$.** In contrast, our experiments generate travel requests from distributions that closely resemble real-world scenarios, which are unlikely to encounter such extreme situations. Hence, it is likely that the empirical results do not demonstrate the theoretical bound of robustness.
>
> Furthermore, we want to explain that, by our experimental setup, **the prediction error $\eta$ is not $\infty$ even when the perturbation probability is $1.0$**. Note that the random noise added is sampled from the same distribution used for generating ticket prices. Taking Fig. 29 as an example, when $T = 10$ and the perturbation probability is $1.0$, the expected prediction error $\eta$ is $500$, which is the sum of 10 samples of the noise distribution with a mean of 50 (equal to the ticket price distributions). On the other hand, $\gamma = C / (1 - \beta) = 500$. Let's assume the actual prediction error sampled from distribution is exactly its expectation, i.e., $\eta = 500$. Then, by equation (10), the competitive ratio for the case of perturbation probability = $1.0$ in Fig. 29 is $(4-\beta) / (1+2\beta) = 3.8/1.4 \approx 2.714$, much smaller than $1/\beta = 5$.
>
> In fact, the instance that leads to the theoretical worst-case scenario $1/\beta$ is an **arbitrarily dense** sequence of requests with small prices (the price of each request $\ll \gamma$). However, the experiments consider a discrete timeline (a widely used approach in numerical experiments), which makes it difficult to construct extreme instances. This is one of the reasons why we rarely observe the worst competitive ratio in the experiments.

---

> ### Comment · Reviewer_m14s · 2024-08-12
>
> Thank you for the clarification. In light of comments from other reviewers and your responses, I am inclined to keep my positive score.

---

> > ### Author Response · Authors · 2024-08-13
> >
> > We are really excited to learn that you have recognized our responses. Thanks for your support for our work!

---

### Decision · Program_Chairs · 2024-09-25

**Decision:**

Accept (poster)

**Comment:**

The paper presents a novel, well-explained method with strong theoretical backing and empirical validation, offering significant improvements over previous work. The algorithms and ideas are presented clearly and intuitively, and the paper is well-written overall.  A key strength is that the algorithm simplifies decision-making by not requiring prediction of the entire future input.

Some reviewers deemed the experimental validation insufficient, relying on too few benchmarks. Additionally, the unbounded robustness of the PFSUM algorithm when β approaches 0 and the lack of analysis on the impact of T were highlighted as weaknesses.

Overall, we recommend acceptance as a poster.

To authors: please make sure to address the remaining concerns in the final version.